# Evaluating critical rainfall conditions for large-scale landslides by detecting event times from seismic records

Hsien-Li Kuo[1], Guan-Wei Lin[1,*], Chi-Wen Chen[2], Hitoshi Saito[3], Ching-Weei Lin[1], Hongey Chen[2,4], Wei-An Chao[5]

[1] Department of Earth Sciences, National Cheng Kung University, No. 1, University Road, Tainan City, 70101, Taiwan
[2] National Science and Technology Center for Disaster Reduction, No. 200, Sec. 3, Beixin Road, Xindian District, New Taipei City, 23143, Taiwan
[3] College of Economics, Kanto Gakuin University, 1-50-1 Mutsuura-higashi, Kanazawa-ku, Yokohama, 236-8501, Japan
[4] Department of Geosciences, National Taiwan University, No.1, Section 4, Roosevelt Road, Taipei, 10617, Taiwan
[5] Department of Civil Engineering, National Chiao Tung University, No. 1001, Daxue Rd., Hsinchu, 30010, Taiwan

*Correspondence to*: Guan-Wei Lin (gwlin@mail.ncku.edu.tw)

**Abstract.** One of the purposes of landslide research is to establish an early warning method for rainfall-induced landslides. The insufficient observations of the past, however, have inhibited the analysis of critical rainfall conditions. This dilemma may be resolved by extracting the times of landslide occurrences from the seismic signals recorded by adjacent seismic stations. In this study, the seismic records of the Broadband Array in Taiwan for Seismology were examined to identify the ground motion triggered by large landslides occurring from 2005 to 2014. A total of 62 landslide-induced seismic signals were identified. The seismic signals provided the occurrence times of the landslides for assessment of the rainfall conditions, including rainfall intensity ($I$), duration ($D$), and effective rainfall ($Rt$). Comparison of three common rainfall threshold models ($I$–$D$, $I$–$Rt$, and $Rt$–$D$) revealed duration and effective rainfall to be the crucial factors in developing a forecast warning model. In addition, a critical height of water model combining physical and statistical approaches, $(I-1.5) \cdot D = 430.2$, was established through analysis of rainfall information from the 62 landslides that occurred. The critical height of water model was applied to Typhoon Soudelor of 2015 and successfully issued a large landslide warning for southern Taiwan.

Key words: large landslide, seismic signal, rainfall threshold, forecast

## 1. Introduction

In recent years, the frequency of extreme rainfall events has increased globally, as has the number of large-scale natural disasters (Tu and Chou, 2013; Saito et al., 2014). These large-scale natural disasters (e.g., landslides, floods, etc.) cause both huge economic losses and human casualties. In mountainous areas, large landslides can change the landscape and erosion processes as well. Several previous studies have reported that the characteristics of a large landslide may include (1) extremely rapid mass movement, (2) huge landslide volume, and (3) deep-seated excavation into rock formations (Chigira and Kiho, 1994; Lin et al., 2006). However, the discrimination of large and non-large landslides is still challenging. In practice, the velocity of mass movement and depth of excavation are both difficult to measure, so the landslide area is commonly regarded as an indicator of the scale of a landslide. Although the occurrence frequency of large landslide is lower than that of non-large landslides, known as small landslides, large landslides cause rapid changes in the landscape, and the scale of disasters induced by large landslides is greater than that of small landslides. Therefore, in this study, a landslide that disturbed an area larger than 0.1 km$^2$ is considered a large landslide, while one not meeting this criterion is considered a small landslide. It is well known that rainfall plays a significant role in the occurrence of landslides, so thorough understanding of the influences of different rainfall factors is necessary. To reduce losses, the critical rainfall conditions that trigger large landslides must be identified so that a rainfall threshold can be used as a forecast model to execute disaster prevention and mitigation measures.

In past research, it was difficult to estimate the threshold of precipitation convincingly due to the lack of accurate information on the occurrence times of landslides. Recent studies in geophysics (Kanamori et al., 1984; Suriñach et al., 2005; Lin et al., 2010; Ekström and Stark, 2013; Chao et al., 2016; Chao et al., 2017) have suggested that the mass movement of large landslides may generate ground motion. If such ground motion is recorded by seismic stations, the occurrence times of large landslides can be extracted from the records. In one case study, the rainfall that triggered the Xiaolin landslide, a giant landslide in southern Taiwan that disturbed an area of ~2.6 km² and resulted in more than 400 deaths in August 2009, was examined. In general, if the exact occurrence time of a landslide cannot be investigated, the time point with the maximum hourly rainfall will be conjectured as the occurrence time of the landslide (Chen et al., 2005; Wei et al., 2006; Staley et al., 2013; Yu et al., 2013; Xue et al., 2016). It was found that the time error between the conjectured and exact times would be 13 hours, which would result in an erroneous cumulated rainfall measurement of 513.5 mm (Fig. 1). However, with the assistance of seismic records, the time information for estimating critical rainfall can be acquired.

This study attempts to determine the occurrence times of landslides by identifying landslide-generated seismic signals to construct rainfall thresholds, and to clarify which thresholds are more suitable for triggering different warnings for small and large landslides. By applying various rainfall factors into statistical analysis, a statistical threshold can be built to explore the critical rainfall conditions of landslide occurrences, such as using rainfall intensity and duration to define rainfall threshold curves (Caine, 1980; Guzzetti et al., 2008; Saito et al., 2010; Chen et al., 2015). Those rainfall thresholds provide valuable information for disaster prevention and mitigation. In this study, seismic data recorded by the network of the Broadband Array in Taiwan for Seismology (BATS) (Fig. 2a) and landslide maps generated from satellite images were used to obtain the exact occurrence times and locations of large landslides. From these, the rainfall threshold for large landslides in Taiwan was developed. Moreover, located at the junction of the Eurasian plate and the Philippine Sea plate, Taiwan has frequent tectonic activity (Ho, 1986; Yu et al., 1997; Willett et al., 2003). Fractured rock mass coupled with a warm and humid climate, and an average of 3 to 5 typhoon events per year, contribute to the high frequency of slope failures in mountainous areas in Taiwan (Wang and Ho, 2002; Shieh, 2000; Dadson et al., 2004; Chang and Chiang, 2009; Chen, 2011). The high coverage of the seismic network and rain gauge stations in Taiwan, coupled with the high occurrence frequency of landslides, make the island a suitable area for examining the use of seismic observations to identify landslide occurrence times and thus the rainfall factors contributing to landslide events.

## 2. Study Method

### 2.1 Large landslide mapping

To determine the locations and basic characteristics of large landslides occurring in the years 2005–2014, the landslide areas across the entire island of Taiwan were interpreted using SPOT-4 satellite remote sensing images with a spatial resolution of 10 m in multispectral mode. Images with minimal cloud cover were selected from pre- and post-typhoon and heavy rainfall events. All images were orthorectified to a standard base image and checked manually using fixed visible markers to ensure spatial consistency over time. Figures 2b and 2c show synthetic SPOT images that were used to identify landslides triggered by Typhoon Morakot in 2009. Bare areas are visibly distinguishable in the SPOT images.

The Normalized Difference Vegetation Index (NDVI) was used to conduct a preliminary classification of bare areas (Lin et al., 2004). The exact NDVI thresholds for bare areas differed from one image to another and were determined by tuning the cut-off value based on visible contrasts. After image interpretation, classified areas were clustered based on slope using a digital elevation model with a resolution of 40 m to identify bare areas not associated with landslides (e.g., roads and buildings). The results of the interpretation were compared with a 1:5000 topographic map to exclude areas of interpretation misjudgement,

such as fallow farmland or alluvial fans. Landslides induced specifically by rainstorm events were distinguished by overlaying the pre- and post-event image mosaics. Based on the definition and description of deep-seated gravitational slope deformation (DSGSD) and large landslides (Lin et al., 2013a; Lin et al., 2013b), a large landslide should possess three characteristics: 1) a depth larger than 10 m, 2) a volume greater than 1,000,000 m$^3$, and 3) a high velocity. In practice, it is difficult to confirm these three characteristics without in-situ investigation and geodetic survey. Therefore, a disturbed area of 100,000 m$^2$ was determined as an accommodating indicator to sort large landslides from small landslides. Finally, large and small landslides were distinguished and classified according to the criterion of an affected area of 0.1 km$^2$. In this study, the types and mechanisms of individual landslides were not investigated, but landslide area was used as the main factor for investigating the different rainfall conditions that trigger large and small landslides.

## 2.2 Interpretation of ground motions induced by large landslides

The movement of a landsliding mass has several different motion processes, such as sliding, falling, rotation, saltation, rolling and impacting. These complex motion processes act on the ground surface to generate ground motion (Kanamori et al., 1984; Ekström and Stark, 2013). The seismic wave generated by a landslide can be attributed to the shear force and loading on the ground surface as the mass moves downslope. Many studies have shown that the source mechanism of a landslide is highly complicated, and that its seismic waves mainly consist of surface waves and shear waves, making it difficult to distinguish *P* and *S* waves from station records (Lin et al., 2010; Suwa et al., 2010; Dammeier et al., 2011; Feng, 2011; Hibert et al., 2014). The onset of a landslide seismic signal is generally abrupt. Then the seismic amplitude increases gradually above the ambient noise level to peak ground motion, exhibiting a cigar-shaped envelope. After the peak amplitude, most of the landslide-generated seismic signals have relatively long decay times, on average about 70% of the total signal duration (Norris, 1994; La Rocca et al., 2004; Suriñach et al., 2005; Deparis et al., 2008; Schneider et al., 2010; Dammeier et al., 2011; Allstadt, 2013). In the frequency domain, landslide-induced seismic energy is mainly distributed below 10 Hz, with a triangular signature in a spectrogram, due to an increase over time in high-frequency constituents (Suriñach et al., 2005; Dammeier et al., 2011). The triangular signature in the spectrogram is the distinctive property that readily distinguishes landslide-induced signals from those of earthquakes and other ambient noise.

In this study, a total of nineteen rainstorm events (seventeen typhoon-induced events and two heavy rainfall events) in the years 2005–2014 were selected to examine the seismic records (Table S1). The seismic data during typhoons and heavy rainfall events having cumulated rainfall exceeding 500 mm from 2005 to 2014 were collected, and the seismic signals of local earthquakes, regional earthquakes, and teleseismic earthquakes were excluded based on the earthquake catalogues maintained by the United States Geological Survey and the Central Weather Bureau, Taiwan. After the removal of instrument response, mean, and linear trends, a multitaper method (Percival and Walden, 1993; Burtin et al., 2009) was employed for spectral analysis of the continuous seismic records. A 5-min moving window with 50% overlap of the seismic records provided a good spectrogram in the frequency range of 1–10 Hz. Eventually, landslide-related triangular signatures in the spectrograms were manually identified to find the characteristic signals generated by landslides (Fig. 3a, 3b). To reduce the uncertainty caused by manual identification, events with obvious triangular signatures in the spectrograms (e.g. Fig. S1) were used to examine rainfall statistics in this study.

The detection of the occurrence time of landslide-induced ground motion is a substantial key to this study. In seismology, many methods can be used to detect the appearance of the seismic signals of earthquakes, and one of the most widely used methods is the STA/LTA ratio (Allen, 1978). For landslides, the duration of landslide-induced signals usually ranges from tens to hundreds of seconds (Helmstetter and Garambois, 2005; Chen et al., 2013a). As compared with the current widely-used rainfall data recorded once per hour, the duration of landslide-induced seismic signals is significantly short. Thus, to avoid

misjudgements caused by different signal-detection methods or manual interpretation, this study adopted the time of the maximum amplitude of the envelope of the vertical-component signal recorded in the station closest to the landslide as the occurrence time of the landslide. Considering the transmission speed of seismic waves, a time difference of several seconds to several tens of seconds was negligible with respect to the sampling rate of rainfall records.

To determine which landslides generated ground motion, it was necessary to locate the seismic sources of the signals. However, the arrival times of the *P*- and *S*-waves of landslide-induced ground motion could not be clearly distinguished. As a result, a locating approach proposed by Chen et al. (2013a) and Chao et al. (2016) was adopted in this study to locate the landslide-induced signals. Locations were estimated with a cross-correlation method that could maximize tremor signal coherence among the seismic stations. The criteria of the stations chosen were their geographic distribution and tremor signal-to-noise ratios. The interpreted signals were treated with an envelope function to process cross-correlations analysed from different station pairs. Centroid location estimates were obtained by cross-correlating all station pairs and performing the Monte Carlo grid search method (Wech and Creager, 2008). While traditional methods seek the source location that minimizes the horizontal time difference between predicted travel time and peak lag time, this method seeks to minimize the vertical correlation distance between the peak correlation value and the predicted correlation value.

Finally, the location results of landslide-induced seismic signals were compared with the exact locations of large landslides interpreted from satellite images (Fig. 3c, 3d). If the locations matched, the occurrence times of the landslides could be obtained, and the time information could be applied to rainfall data analysis.

## 2.3 Analysis methods of statistically-based rainfall threshold for landslides

In the study, hourly rainfall data were collected from the records of rain gauge stations (Fig. 2a). The major rainfall events analysed in the study were typhoon events. The distribution of precipitation during typhoon events is usually closely related to the typhoon track and the position of the windward slope, also as known as the orographic effect. In addition, the density and distribution of rainfall stations in mountainous areas directly affect the results of rainfall threshold analysis. If the landslide location and the selected rainfall station are located in different watersheds, the rainfall information is unlikely to represent the rainfall conditions for the landslide. In some cases, however, the diameter of the typhoon was so large that the orographic effects could be ignored (Chen and Chen, 2003; Sanchez-Moreno et al., 2014) Therefore, in this study, the selection criteria for a rainfall station were that the rainfall station must be located within the same watershed as the landslide, and at the shortest straight distance from the landslide; moreover, the watershed must be smaller than 100 km$^2$ in area to ensure that the records at rain gauge stations were sufficient to represent the rainfall at the landslide locations. These criteria were established after testing the influences of distance and topographic effects on rainfall distribution (see supporting information S3). In rainfall analysis, the beginning of a rain event is defined as the time point when hourly rainfall exceeds 4 mm, and the rain event ends when the rainfall intensity remains below 4 mm/h for 6 consecutive hours. The critical rainfall condition for a landslide was calculated from the beginning of a rain event to the occurrence time of the landslide (Jan and Lee, 2004; Lee, 2006). In this way, average rainfall intensity (mm/h), cumulated rainfall (mm), and rainfall duration (h) for each large landslide could be used as the factors in the rainfall threshold analysis. In addition to the three factors mentioned above, the daily rainfall for the seven days preceding the rainstorm was considered as antecedent rainfall (*Ra*). The antecedent rainfall (*Ra*) was calculated with a temporal weighting coefficient of 0.7, with the weight decreasing with days before the event. The formula was $Ra = \sum_{i=1}^{7} 0.7^i \times R_i$, where $R_i$ is the daily rainfall of the i$^{th}$ day before the rainfall event. The sum of antecedent rainfall and principal event rainfall was regarded as the total effective rainfall (*Rt*). This definition of a rain event has been officially adopted in Taiwan (Jan and Lee, 2004). The use of different definitions of a rain event would result in differences in statistical rainfall conditions, but the statistical criteria used in this study ensured the consistency of data processing in the critical rainfall analysis.

Based on different rainfall factors, three common rainfall threshold analysis methods were used in the study. The first method was the *I-D* method, with the power law curve, $I = aD^{-b}$, where *a* is the scaling parameter (the intercept) of the threshold curve and *b* is the slope (the scaling exponent) (Caine, 1980; Wieczorek, 1987; Keefer et al., 1987). In this study, the *I-D* rainfall

threshold curve at 5% exceedance probability was estimated by the method proposed by Brunetti et al. (2010). This threshold was expected to leave 5% of the data points below the threshold line. The second method was the rainfall-based warning model proposed by Jan and Lee (2004), which is based on the *Rt* and *I* product values. With the *I-Rt* method, rainfall intensity and cumulated rainfall were plotted and used to calculate the cumulative probability of the product value of *I* and *Rt* by the Weibull distribution method (Jan and Lee, 2004). The cumulative probability of 5% of *Rt* and *I* product values was taken as the *I-Rt*

rainfall threshold. The third method was the *Rt-D* method (Aoki, 1980; Fan et al., 1999). In the *Rt-D* method, the 5% cumulative probability of the product value of *Rt* and *D* by the Weibull distribution method was taken as the *Rt-D* rainfall threshold.

In addition to that of large landslides, the time information of 193 small landslides such as shallow landslides and debris flows from the years 2006–2014 was collected from the annual reports of debris flows investigated by the Soil and Water

Conservation Bureau (SWCB) of Taiwan, but it was not extracted from seismic records. Most of the 193 small landslides caused disasters and loss of life and property. In some cases, in-situ steel cables or closed-circuit television recorded the time information. This information was applied to the rainfall data analysis and then used to compare the rainfall conditions of the large landslides.

## 2.4 Critical height of water model

Whether a given slope will produce a landslide depends on the balance between the shear strength of the slope material and the downslope component of the gravitational force imposed by the weight of the slope material above a potential slip surface. A critical height of water model proposed by Keefer et al. (1987) was used in this study to construct a rainfall threshold. The model was derived from existing slope stability theory with some simplifying assumptions. The shear strength of the material at a point within a slope is expressed as:

$$s = c\prime + (p - u_w)tan\phi\prime, \tag{1}$$

where $c\prime$ is effective cohesion of material, $p$ is total stress perpendicular to the potential sliding surface, $u_w$ is pore water pressure, and $\phi\prime$ is effective friction angle of slope material. The main cause of a slope failure is the infiltration of rainfall into

the slope and accumulation above the impermeable layer, which increases the pore water pressure of the slope material. As the pore water pressure ($u_w$) increases, the shear strength ($s$) decreases, eventually leading to slope failure. A critical value of pore water pressure $u_{wc}$ exists in each slope, assuming an infinite slope composed of a non-cohesive sliding surface ($c\prime$=0). The pore water pressure threshold can be calculated as:

$$u_{wc} = Z \cdot \gamma_t [1 - (tan\,\theta \,/\, tan\phi\prime\,)], \tag{2}$$

where Z is the vertical depth of the sliding surface, $\gamma_t$ is the unit weight of the slope material, and $\theta$ is the slope angle. Good development of a detachment plane (e.g., sliding surface between sedimentary layers, connected joints, and weathered foliation) has been widely considered as the geological condition under which a large landslide occurs (Agliardi et al., 2001; Tsou et al.,

2011). Therefore, in this study, the *c'* of the detachment plane is simply assumed to be zero to represent the critical situation of slope stability.

As the pore water pressure $u_w$ increases to the pore water pressure threshold $u_{wc}$, a critical height of water $Q_C$ is retained above the sliding surface until the initiation of slope failure. The $Q_C$ is calculated as:

$$Q_C = (u_{wc}/\gamma_w) \cdot n_{ef}, \tag{3}$$

where $u_{wc}$ is the critical value of pore water pressure, $\gamma_w$ is the unit weight of water, and $n_{ef}$ is the effective porosity, which is the residual porosity of the slope material under free gravity drainage. The drainage rate of a saturated zone is represented by the average value $I_0$, the unit of which is mm/h. In a heavy rainfall event, the critical quantity of water for causing a slope failure is defined as:

$$Q_C = (I - I_0) \cdot D, \tag{4}$$

## 3. Results

### 3.1 Topographic features of large landslides

The satellite imagery interpretation showed that, from 2005 to 2014, a total of 686 landslide events with areas greater than 0.1 km$^2$ occurred in mountainous areas of Taiwan (Fig. 4a). Most of these large landslides had areas of 0.12 to 0.15 km$^2$, and their slope angles before the landslides occurred were concentrated between 30° and 40° (Fig. 4b). The number of landslides occurring on slope angles exceeding 40° slightly increased after 2010. Although the increase was quite slight, it was most likely due to the fact that during the extremely heavy rainfall of Typhoon Morakot in 2009, more than 2000 mm precipitated in four days, causing a large number of landslides and exhausting many unstable slopes (Chen et al., 2013b). Consequently, landslides occurred on steeper slopes in the following years. The large landslides were primarily concentrated on slopes with elevations ranging from 500 m to 2000 m (Fig. 4c), but the distributions of the highest and lowest elevations of these large landslides showed that their average vertical displacement was greater than 500 m.

The location information of the 193 small landslides investigated by the SWCB was used to obtain the topographic features of the small landslides as well. The distribution of the slope angles of these landslides was similar to that of the large landslides. However, the distribution of the elevations of the small landslides was quite different from that of the large landslides. Unlike those of the large landslides, a large portion of the elevations of small landslides was concentrated at about 1000 m. Although the difference in elevation distribution between large and small landslides seems to indicate that the topographic features of large landslides were relatively more widespread than those of small landslides, the situation should be attributed to the limited *in-situ* investigations of the SWCB. Currently, the vast majority of landslides still cannot be investigated in the field.

### 3.2 The critical rainfall conditions for triggering large landslides

Comparison of the location solutions of seismic signals and the landslide distribution map revealed that the matched large landslides had deviations in distance of 0 to 20 kilometres. In addition to distance, the resultant traces of two horizontal-component signals could be plotted. The direction of the resultant trace of a given landslide-induced seismic record with the slope aspect in the vicinity of the located point could be compared so as to eliminate the irrelevant landslides, those which had slope aspects different from the signal traces. The ground motion traces of the signals had to be correlated with the directions of movement of the landslides to reconfirm the matched large landslides. In total, 62 large landslides were paired successfully with seismic record locations (Fig. 2a, Table S2). These 62 large landslides were distributed in watersheds with high cumulated rainfall during heavy rainfall events. In addition, the 62 large landslides were verified by satellite images from multiple years

to guarantee that the shapes and positions were highly credible. Subsequently, the occurrence times of these 62 large landslides were obtained from seismic signals.

The time information was used to implement rainfall analysis. About two-thirds (41) of the large landslides occurred when the total effective rainfall exceeded 1000 mm (Fig. 5). The statistical results of rainfall intensities at the times of large landslide occurrences showed that more than half of the large landslides occurred when the rainfall intensity was less than 20 mm/h. Only seven of the large landslides occurred when the rainfall duration was less than 24 hours, and the rainfall durations of these seven events all exceeded 10 hours. The results of single rainfall-factor analysis indicated that the effects of rainfall duration and cumulated rainfall were much more remarkable for large landslides than for small landslides, and that the rainfall intensity at the time of landslide occurrence was not the main factor influencing large landslides. Therefore, the average rainfall intensity was adopted for the following multi-factorial analyses.

## 4. Rainfall thresholds for large landslides

### 4.1 Dual rainfall-factor analysis of *I–D*, *I–Rt,* and *Rt–D* thresholds

The single rainfall-factor analysis indicated that there was no significant correlation between landslides and rainfall intensity at the time of large landslide occurrences. In the dual rainfall-factor analysis, the *I–D* rainfall threshold was assessed by using the average values of rainfall intensity and rainfall duration. The obtained *I–D* rainfall threshold was $I = 71.9D^{-0.47}$ ($D > 24$ h) (Fig. 6a). The rainfall information obtained from small landslides that were reported by the SWCB from 2006 to 2014 was also compared, and the *I–D* rainfall threshold curve for large landslides also fit the lower boundary of the rainfall conditions of small landslides. In addition, the distribution of the rainfall durations indicated that the small landslides were distributed evenly from 3 to 70 hours, while the large landslides were mostly distributed above 20 hours. The rainfall intensity, however, could not be used effectively to distinguish these two kinds of slope failures. Even under the same rainfall duration, the rainfall intensities of many small landslides were higher than those of large landslides. This result sufficiently demonstrated that rainfall intensity could not be used to distinguish between small landslides and large landslides. Therefore, the *I–D* rainfall threshold may not allow assessment of the landslide scale. It was also found that most of the large landslides with larger areas were concentrated in rainfall durations of more than 50 hours, but the average rainfall intensity was not well-correlated with landslide area. The average rainfall intensity of the small landslides was very high for short durations, but the average duration of the small landslides was much lower than that of large landslides. Therefore, continuous high-intensity rainfall incurs a high likelihood of large landslide occurrence.

The *I–D* rainfall thresholds obtained in the study were also compared with those of previous studies that focused on shallow landslides or debris flows. This comparison revealed that the *I–D* threshold curve for large landslides was much higher than the threshold curves for shallow landslides or debris flows.

Based on the analysis of the relationship between total effective rainfall (*Rt*) and rainfall duration (*D*), the product of *Rt* and *D* for large landslides with a cumulative probability of 5% was 12,773 mm·h (Fig. 6b), and the rainfall threshold was also much higher than the 5% cumulative probability of small landslides (487 mm·h). Total effective rainfall differed considerably between large and small landslides. Most small landslides had a total effective rainfall below 500 mm, and only a few occurred when total effective rainfall exceeded 1000 mm. The landslide size groups shifted from small landslides for relatively short duration and low effective rainfall to large landslides for long duration and very large effective rainfall. As a result of the disparity in the *Rt–D* threshold curves for large and small landslides, it was determined that *Rt–D* analysis could be used effectively to distinguish small landslides from large landslides.

The analysis of the relationship between average rainfall intensity ($I$) and total effective rainfall ($Rt$) revealed that the product value of both factors for 5% cumulative probability was 5,640 $mm^2/h$ (Fig. 6c). The $Rt$–$I$ threshold curve for large landslides was not much higher than that for small landslides (1,541 $mm^2/h$). Combining the results of the three kinds of dual-factor rainfall threshold analyses revealed that the critical rainfall conditions for small landslides included high average rainfall intensity but relatively low effective rainfall, while those for large landslides included long rainfall duration and high effective cumulated rainfall. These results corresponded well with the former theoretic expectation (Van Asch et al., 1999; Iverson, 2000).

The main mechanism of shallow landslides is heavy rainfall along with rapid infiltration, causing soil saturation and a temporary increase in pore-water pressure. However, prolonged rainfall also plays an important role in slow saturation, which in turn influences the groundwater level and soil moisture, and causes large landslides. These facts have been recognized in many studies around the world (Wieczorek and Glade, 2005; Van Asch et al., 1999; Iverson, 2000), but they have been analysed in only a few locations (e.g., a mountainous debris torrent, a shallow landslide event, and an individual rainfall event). Using the regional dataset of landslides and the times information, this study identified the critical rainfall conditions for large and small landslides in Taiwan.

## 4.2 The critical height of water model for large landslides

The critical height of water, $Q_C$, on a sliding surface for each large landslide was estimated based on its slope gradient, depth (estimated by the equation $Z = 26.14A^{0.4}$; $Z$: depth in m; $A$: disturbed area in $m^2$), and the geological material parameters of the study area (Table 1). The $Q_C$ value was inserted into $Q_C = (I - I_0) \cdot D$ to obtain an $I_0$ value for each large landslide. For the 62 detected landslides, the cumulative probability of 5% of the $Q_C$ and $I_0$ values was taken as the critical value. The critical value of $I_0$ was 1.5, the critical $Q_C$ was 430.2, which is more suitable for large than for small landslides, and the threshold curve was rewritten as $(I - 1.5) \cdot D = 430.2$. The application of this threshold curve to average rainfall intensity and rainfall duration showed that almost all the large landslides could have been forecasted. This application demonstrated a good function as a large landslide forecast model (Fig. 6d). In addition, the threshold curve can be used to distinguish large landslides and small landslides clearly. This advantage can prevent or reduce false forecasts. The critical height of water model combines statistical and deterministic approaches for the assessment of critical rainfall. Therefore, the parameters used to calculate $Q_C$ can be adjusted based on regional geologic and topographic environments within a specific area. The $Q_C$ model illustrates the importance of the cumulative volume of water and rainfall duration to large landslides and takes into account the effects of both infiltration of water and average rainfall intensity. The critical hydrological conditions for large landslides, a long duration and a high amount of cumulated rainfall, can be determined as well.

In general, physically-based models are easy to understand and have high predictive capabilities (Wilson and Wieczorek, 1995: Salciarini and Tamagni, 2013; Papa et al., 2013; Alvioli et al., 2014). However, they depend on the spatial distribution of various geotechnical data (e.g., cohesion, friction coefficient, and permeability coefficient), which are very difficult to obtain. Statistically-based methods can include conditioning factors that influence slope stability, which are unsuitable for physically-based models. Statistically-based models rely on good landslide inventories and rainfall information. In this study, the $Q_C$ threshold for a large landslide was estimated based on a mixture of physically- and statistically-based methods. Unlike other physically-based $I$-$D$ thresholds, which are commonly constructed based on artificial rainfall information for shallow landslides (Salciarini et al., 2012; Chen et al., 2013c; Napolitano et al., 2016) (Table S3), the $Q_C$ threshold proposed in this study seemed to be higher and more suitable for large landslides (Fig. 6d).

Although the geological and rainfall conditions in Taiwan and in other countries are not the same, seismic records can be used to obtain the time information of landslide occurrences for rainfall threshold analysis in other countries. For countries with geological and rainfall conditions similar to those of Taiwan (e.g., Japan and the Philippines) (Saito and Oguchi, 2005; Yoshimatsu and Abe, 2006; Evans et al., 2007; Yumul et al., 2011), the results of this study may serve as a useful reference for the development of a forecast model for rainfall-triggered landslides.

## 5. Discussion

### 5.1 Application of rainfall thresholds

To verify the usability of the rainfall thresholds proposed in this study, Typhoon Soudelor of 2015 was chosen to demonstrate the early warning performance. Typhoon Soudelor was one of the most powerful storms on record. It generated 1400 mm of rainfall in northeastern Taiwan and almost 1000 mm of rainfall in the southern mountainous area of Taiwan (Wei, 2017; Su et al., 2016). After the seismic signal analytical procedure, the occurrence time, 2015/8/8 18:59:50 (UTC), of a large landslide (named the Putanpunas Landslide) located in southern Taiwan was obtained (Fig. 7). The seismic signal generated by the Putanpunas Landslide was also detected by Chao et al. (2017). The seismic signals generated by this large landslide could be identified from six BATS stations, and the distance error was less than 6 km. The rainfall records of rain gauge station C1V190, which was situated in the same watershed and 14.6 km away from the large landslide, were collected for rainfall analysis. Typhoon Soudelor made landfall in Taiwan on August 7, 2015, and dropped a cumulated rainfall of 546 mm and had a maximum rainfall intensity of 39 mm/h on August 8 at rain gauge station C1V190 (Fig. 8). The rainfall event began at 22:00 August 7 and last for 26 hours, and the Putanpunas Landslide initiated at the 22nd hour. This landslide occurred when the rainfall intensity was on the decline.

Regarding landslide early warning using rainfall thresholds, once the rainfall conditions at a given rainfall station exceed the rainfall thresholds for triggering landslides, the slopes located within the region of the rainfall station will have high potential for failure. Based on the statistically-based $I$-$D$ threshold for small landslides, a small-landslide warning would have been issued at the sixth hour of the rainfall event (Fig. 8). The long interval of sixteen hours between the warning and the occurrence time of the Putanpunas Landslide could have reduced the reliability of the warning or even caused the warning to be considered a false alarm. Therefore, it is essential to establish different thresholds for landslides of different scales. Using the $I$-$Rt$ threshold (i.e., $Rt·I$= 5,640), a large-landslide warning would have been issued at the ninth hour of the rainfall event (i.e., thirteen hours before the Putanpunas Landslide occurred). According to the statistically-based $I$-$D$ threshold for large landslides, a landslide warning would have been issued at the same hour as the $I$-$Rt$ threshold. In addition, a warning based on the $Rt$-$D$ threshold (i.e., $Rt·D$ = 12,773) would have been issued three hours after the occurrence time of the Putanpunas Landslide. According to the rainfall records and the critical height of water model (i.e. $(I$-1.5$)·D$=430.2), a landslide warning would have been issued at 16:00 on August 8, three hours before the occurrence time of the Putanpunas Landslide. Compared to the statistically-based $I$-$D$ threshold, the $I$-$Rt$ threshold, and the $Rt$-$D$ threshold, the critical height of water model had a better early-warning performance for the 2015 Putanpunas Landslide.

### 5.2 Limitation of seismic detection for large landslides

The number of large landslides detected from seismic records, 62, comprised only nine percent of the total large landslides in 2005–2014 in Taiwan. This low percentage indicates that the vast majority of large landslides were not well identified from seismic records. If this limitation can be surmounted, more time information on large landslide occurrences can be used to develop rainfall thresholds. The average interstation spacing of the Broadband Array in Taiwan for Seismology is around 30

km. A higher density of seismic stations would improve the detection function. In addition, to determine the limitation of large landslide detection distance as a function of large landslide-disturbed area, the most distant seismic station where large landslide signals were visible was selected. Some previous studies have applied similar approaches to probe the detection limit (Dammeier et al., 2011; Chen et al., 2013a). The relationship between the maximum distance of detection and the large landslide-disturbed area shows a limitation of the detection distance due to the large landslide's magnitude (Fig. 9). In Figure 9, each data point represents the distance between a landslide location and the most distant seismic station detecting it, as well as the landslide-disturbed area. In other words, when the distance between a seismic station and a landslide that has the same given landslide-disturbed area as the data is shorter than the value of the data, seismic signals induced by the landslide can be interpreted from the records of the seismic station. Therefore, a lower boundary of these data can be determined to demarcate an effective detectable region. As a large landslide's area increases, the maximum distance between the large landslide location and seismic detection increases. A detection limit can be described by

$$\log(distance) \ = \ 0.5069 \times \log(area) - 1.3443, \tag{5}$$

The boundary of detection was determined empirically based on the two lowest values of the farthest distance of detection (i.e., 31.0 km and 37.6 km) having disturbed areas of $1.6 \times 10^5$ and $1.2 \times 10^5$ m$^2$. For a given large landslide, if a station is located below the upper detection limit, the seismic signal should be detectable. However, not all the stations located in detectable regions recorded clear large landslide-induced seismic signals. One of the possible reasons is that the environmental background noise affected the signal to noise ratio of the seismic records during heavy rainfall events. Therefore, the detection limit may also depend on the signal quality at each station.

## 6. Conclusion

In this study, seismic signals recorded by a broadband seismic network were used to determine the exact times of occurrence of large landslides, and the rainfall threshold for large landslides was assessed statistically based on the time information. Based on the rainfall information of 62 large landslides that occurred from 2005 to 2014 in Taiwan, the rainfall conditions for triggering large landslides include total effective rainfall of more than 1000 mm and rainfall duration of more than 24 hours. After the rainfall thresholds were analysed by the *I-D*, *Rt-D*, and *I-Rt* methods, the rainfall thresholds based on different dual factors for triggering large landslides were obtained. Furthermore, a critical water model combining statistical and deterministic approaches was developed to figure out a three-factor threshold for large landslides. The rainfall information and geologic/topographic parameters finally were applied to obtain the threshold curve, $(I-1.5) \cdot D = 430.2$, where average rainfall intensity *I* is in mm/h and rainfall duration *D* is in h. This new critical model can be used to improve the forecasting of large landslides and will not lead to confusion between small landslides and large landslides. The influences of extreme rainstorm events and rock types on the rainfall threshold were also investigated. However, the changes in the rainfall thresholds for large landslides either before or after an extreme event or in different rock types were not notable.

### Acknowledgement

The authors gratefully acknowledged the financial support of the Ministry of Science and Technology of Taiwan and the Soil and Water Conservation Bureau, Council of Agriculture, Executive Yuan of Taiwan. The source of all seismic and rainfall information included in this paper was the Institute of Earth Sciences, Academia Sinica of Taiwan, and the Seismology Center, Central Weather Bureau (CWB), Taiwan.

**Supplementary Material**

The supplementary material contains five sections (S1–S5), including three supplementary figures (Fig. S1–S3) and three supplementary tables (Tables S1–S3). Nineteen selected rainfall events occurring in the years 2005–2014 are listed in Table S1. A sequence of spectrograms of seismic signals induced by the ID 1 landslide of 2005 is displayed in Fig. S1. The validation of the rainfall data used in the study is explained in section S3, which includes Fig. S2 and S3. Detailed information on the 62 detected landslides is shown in Table S2. The equations of three physically-based I-D thresholds reported in previous studies are listed in Table S3.

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

**Table 1. Parameters for calculating critical height of water $Q_C$**

| Parameters | Value | Reference |
|---|---|---|
| unit weight of slope material, $\gamma_t$ | 2.65 t/m$^2$ | |
| effective friction angle, $\phi'$ | 37 $^\circ$ | Handin et al. (1957, 1963) |
| effective porosity, $n_{ef}$ | 0.1 | West (1995) |

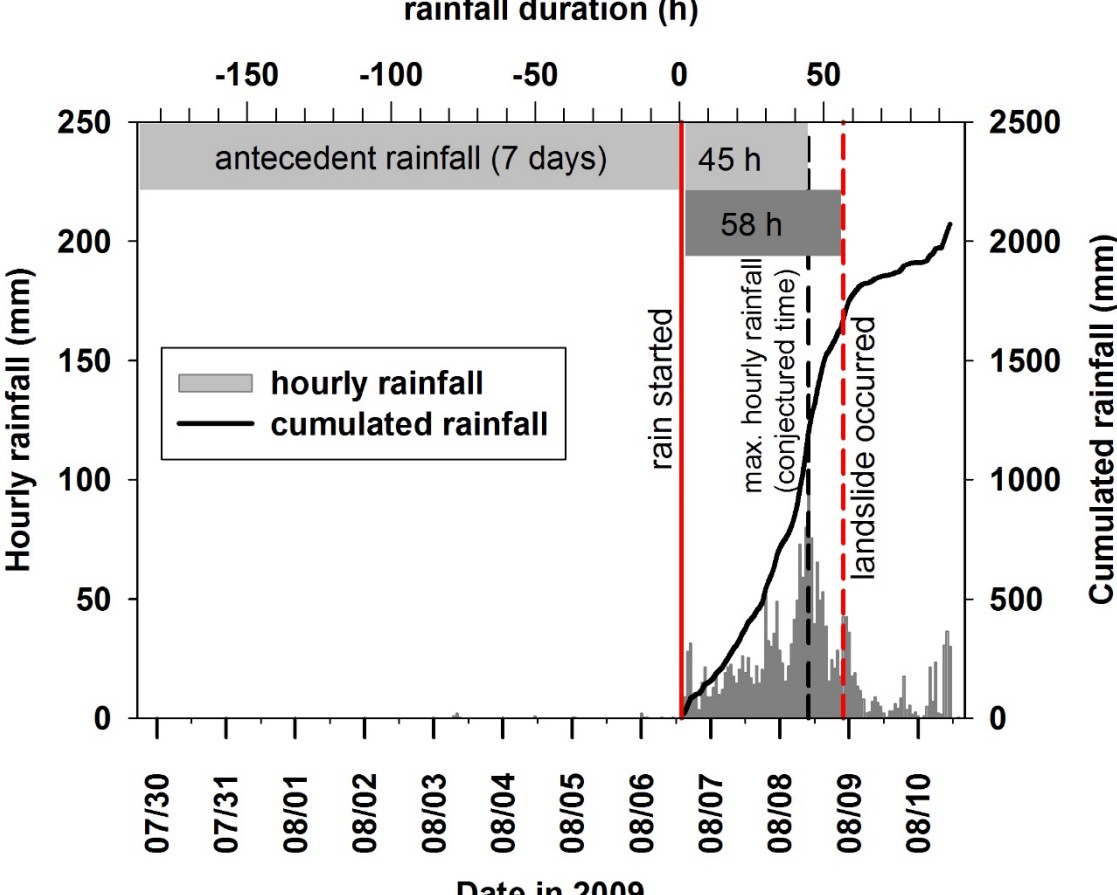

Figure 1: Time series of hourly rainfall and cumulative rainfall from July 29 to August 10, 2009. Rainfall data were collected from the CWB C0V250 rainfall gauge station, which is 12 km from the Xiaolin landslide. The Xiaolin landslide occurred at UTC 22:16 on August 8, 2009. The rainfall event induced by Typhoon Morakot in 2009 started at UTC 14:00 on August 6, when hourly rainfall exceeded 4 mm. The maximum hourly rainfall was at UTC 10:00 on August 8. In general, if the exact time of landslide occurrence cannot be investigated, the time point with the maximum hourly rainfall will be conjectured as the occurrence time of the landslide (Chen et al., 2005).

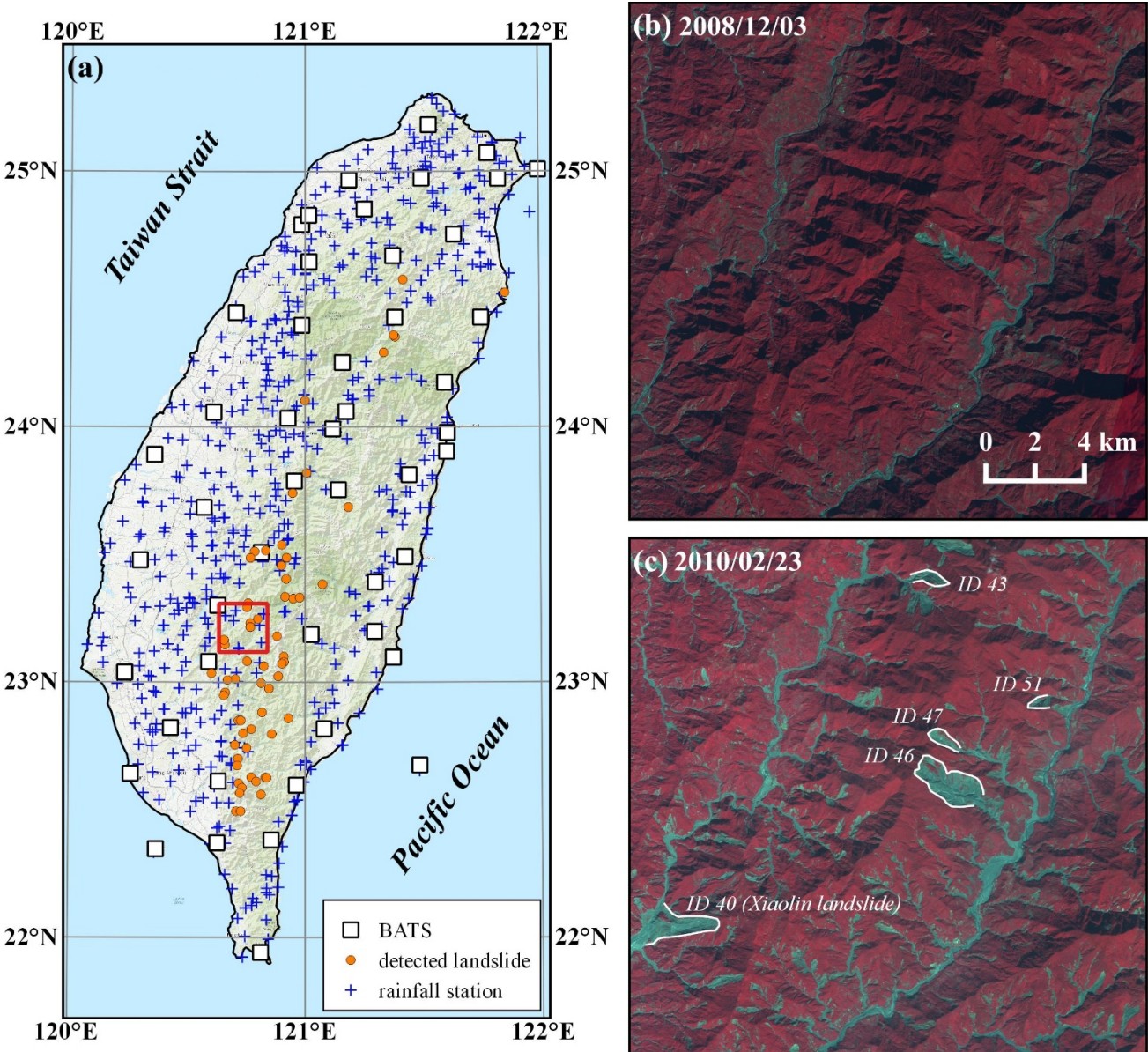

**Figure 2: Comparison of satellite images pre- and post-Typhoon Morakot. (a) Overview map of Taiwan and distribution of rainfall gauge stations. The red frame denotes the areas displayed in (b) and (c). (b) SPOT image taken on December 3, 2008. (c) SPOT image taken on February 23, 2010.**

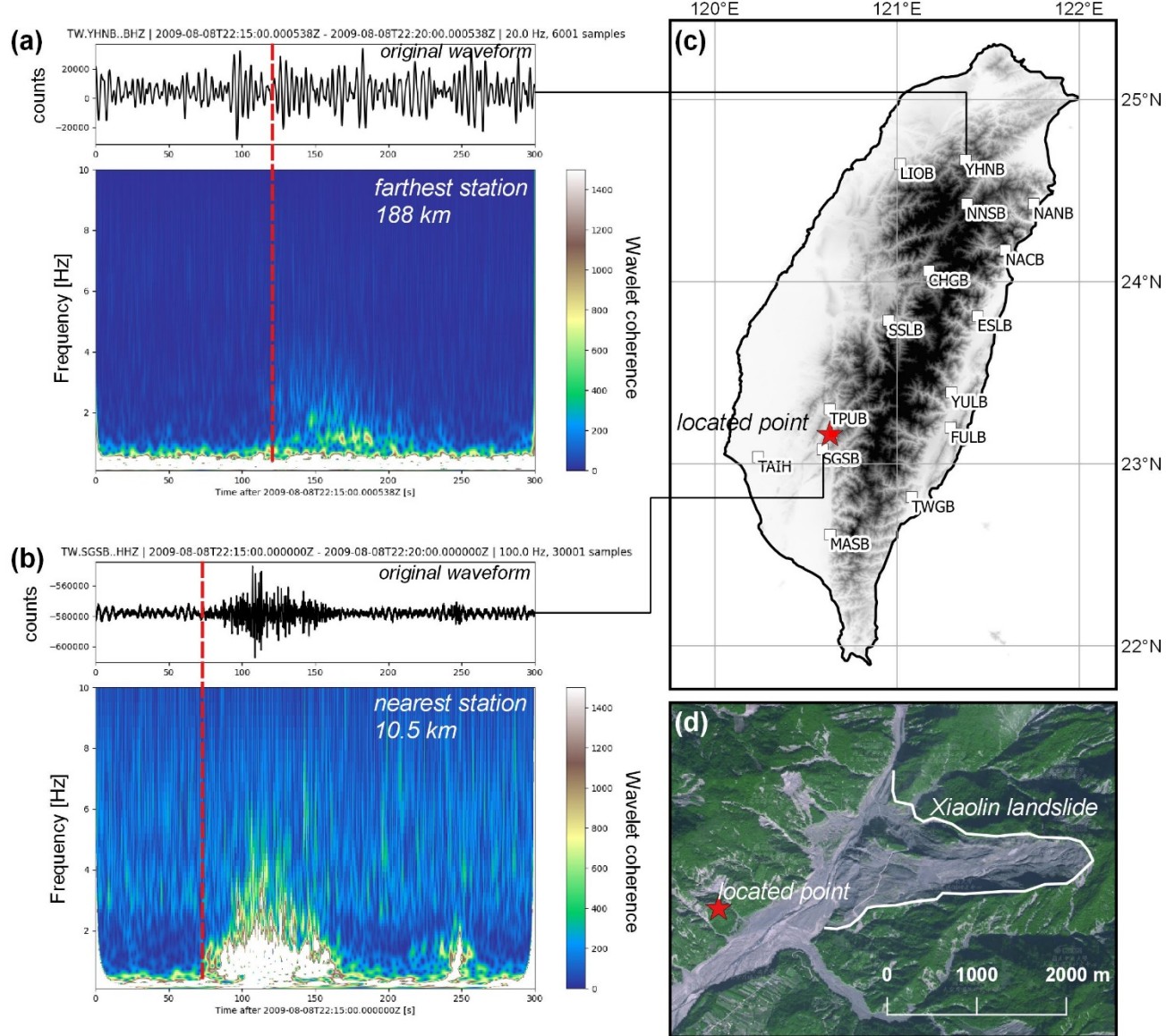

**Figure 3: Characteristic triangle signature visible in a spectrogram within a time window starting at UTC 22:15 and ending at UTC 22:20 on August 8, 2009. (a) Original waveform and spectrogram of the vertical component at station YHNB. (b) Original waveform and spectrogram of the vertical component at station SGSB. (c) Distribution of 15 detections of ground motion induced by the Xiaolin landslide and the location result. (d) The located point and the location of the Xiaolin landslide. The location error between the location result and the landslide site is about 1.5 km.**

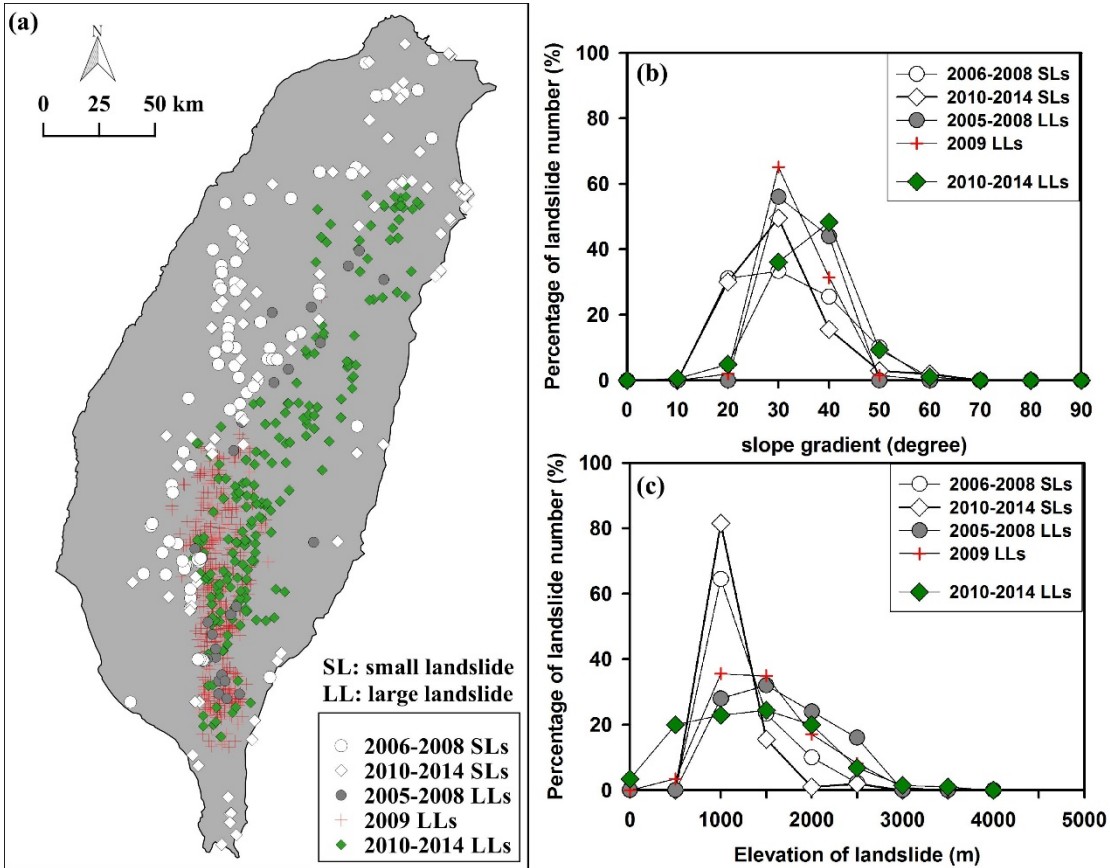

Figure 4: (a) Distribution map of large landslides from 2005 to 2014 and small landslides from 2006 to 2014. (b) The numerical distribution of slope gradients of large and small landslides, presented in percentages. (c) The numerical distribution of elevations of large and small landslides, presented in percentages.

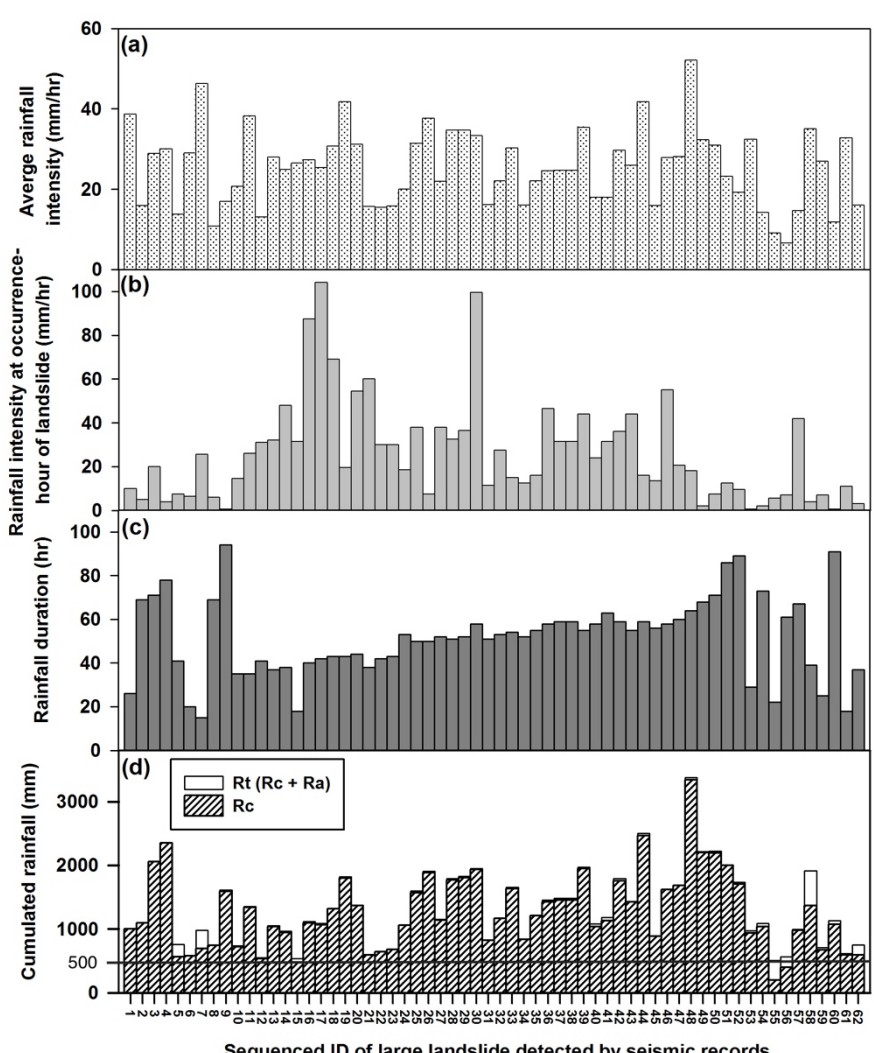

**Figure 5: Single-factor rainfall analysis. Each large landslide is assigned an ID number in the figure. The ID numbers of large landslides are displayed in chronological order. ID 1–4 are the large landslides occurring in 2005; ID 5 is a large landslide occurring in 2006; ID 6–9 are the large landslides occurring in 2008; ID 10–52 are the large landslides occurring in 2009; ID 53 is a large landslide occurring in 2010; ID 54–56 are the large landslides occurring in 2011; ID 57–60 are the large landslides occurring in 2012; ID 61–62 are the large landslides occurring in 2013. No large landslides occurring in 2007 or 2014 were successfully paired with seismic signal results. Most large landslides occurred when rainfall duration exceeded 24 hours, cumulative rainfall exceeded 1000 mm, and rainfall intensity was less than 20 mm/h.**

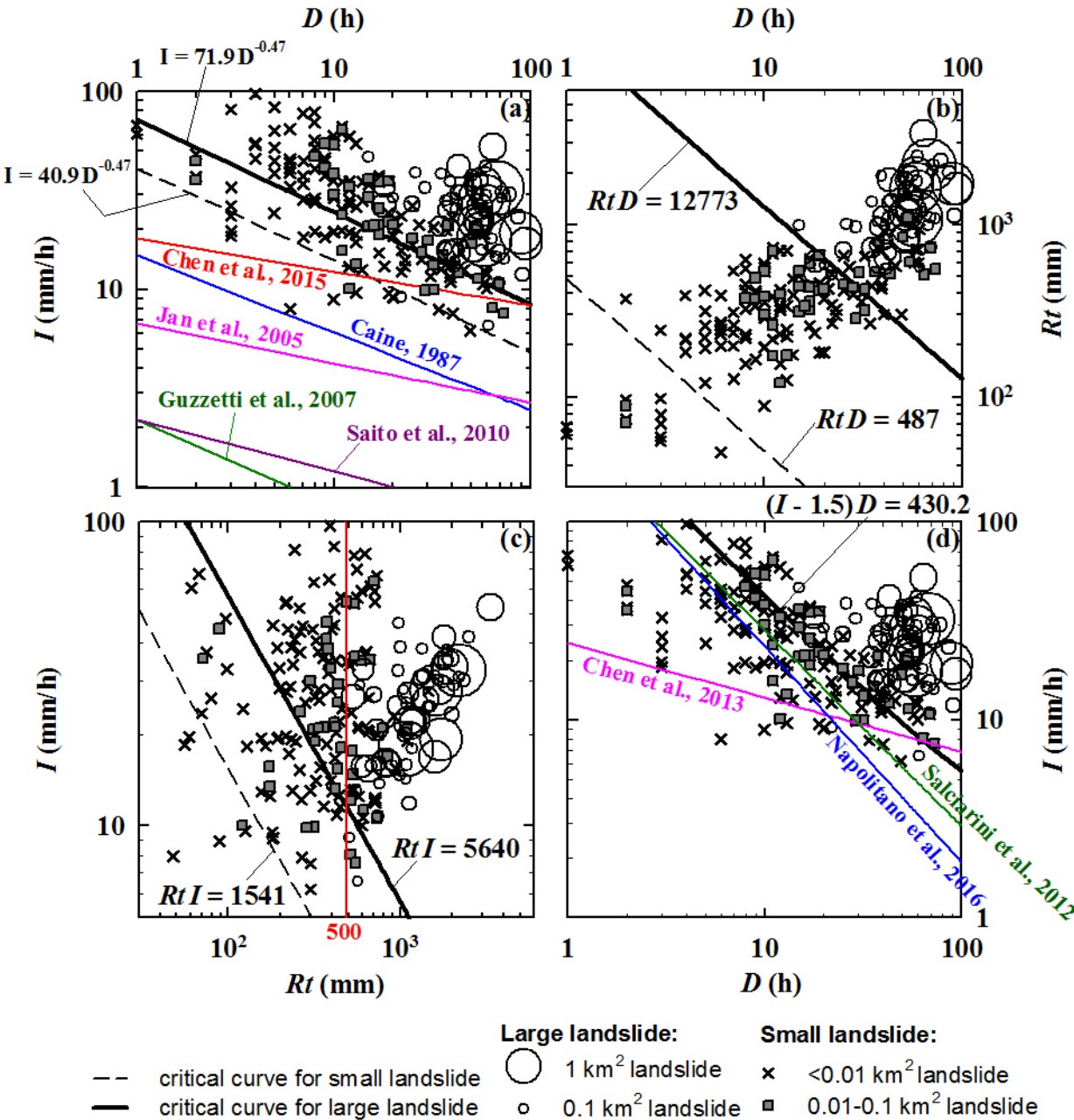

**Figure 6:** (a) I-D rainfall threshold. (b) Rt-D method rainfall threshold. (c) I-Rt method rainfall threshold. (d) Threshold of the critical height of water model, (I-1.5)D=430.2.

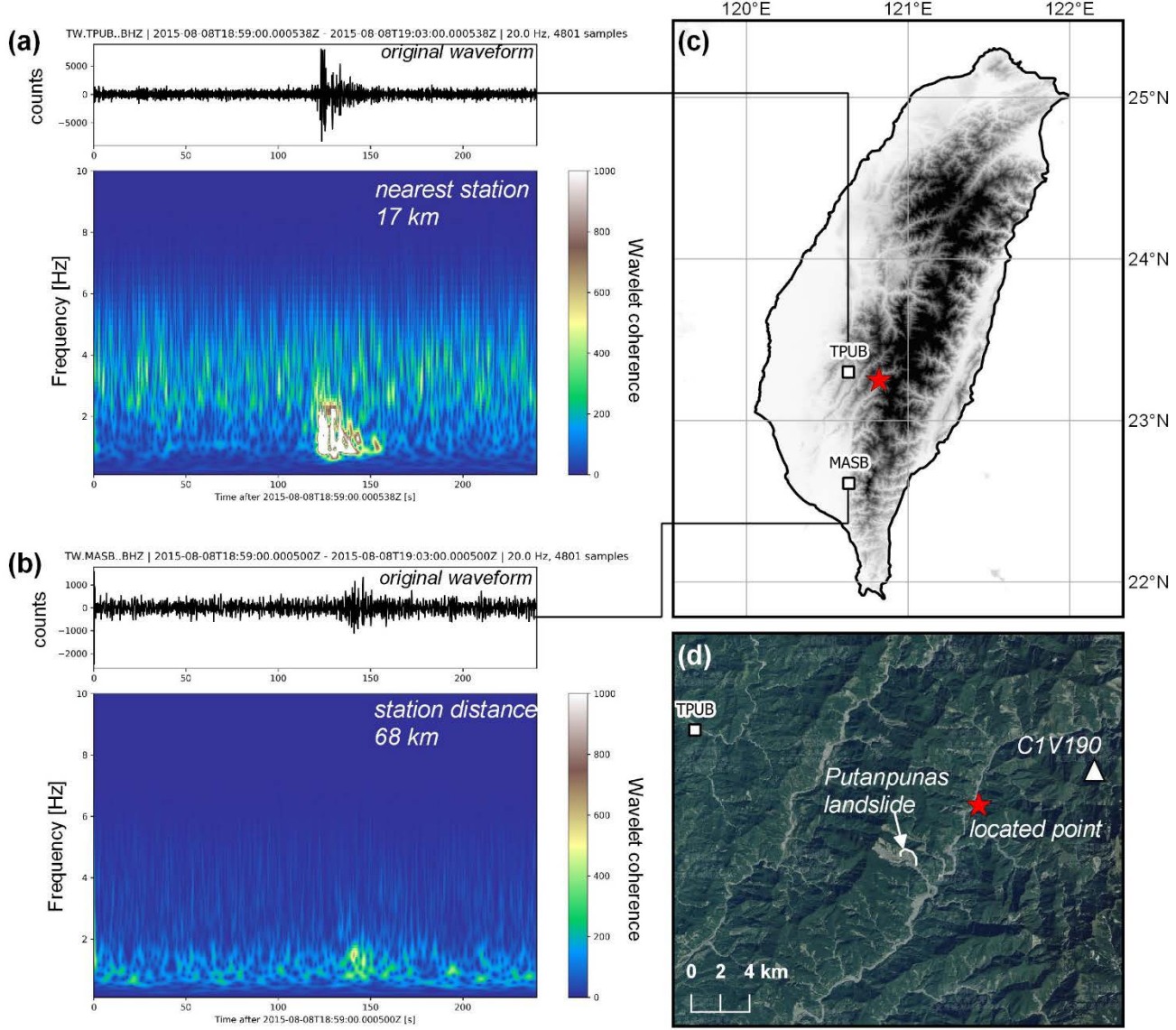

**Figure 7: Characteristic triangle signature visible in a spectrogram within a time window starting at UTC 18:59 and ending at UTC 19:03 on August 8, 2015. (a) Original waveform and spectrogram of the vertical component at station TPUB. (b) Original waveform and spectrogram of the vertical component at station MASB. (c) Distribution of located point (red star) and these two seismic stations. (d) The located point and the landslide site. The distance error between the location result and the landslide site is 3.7 km.**

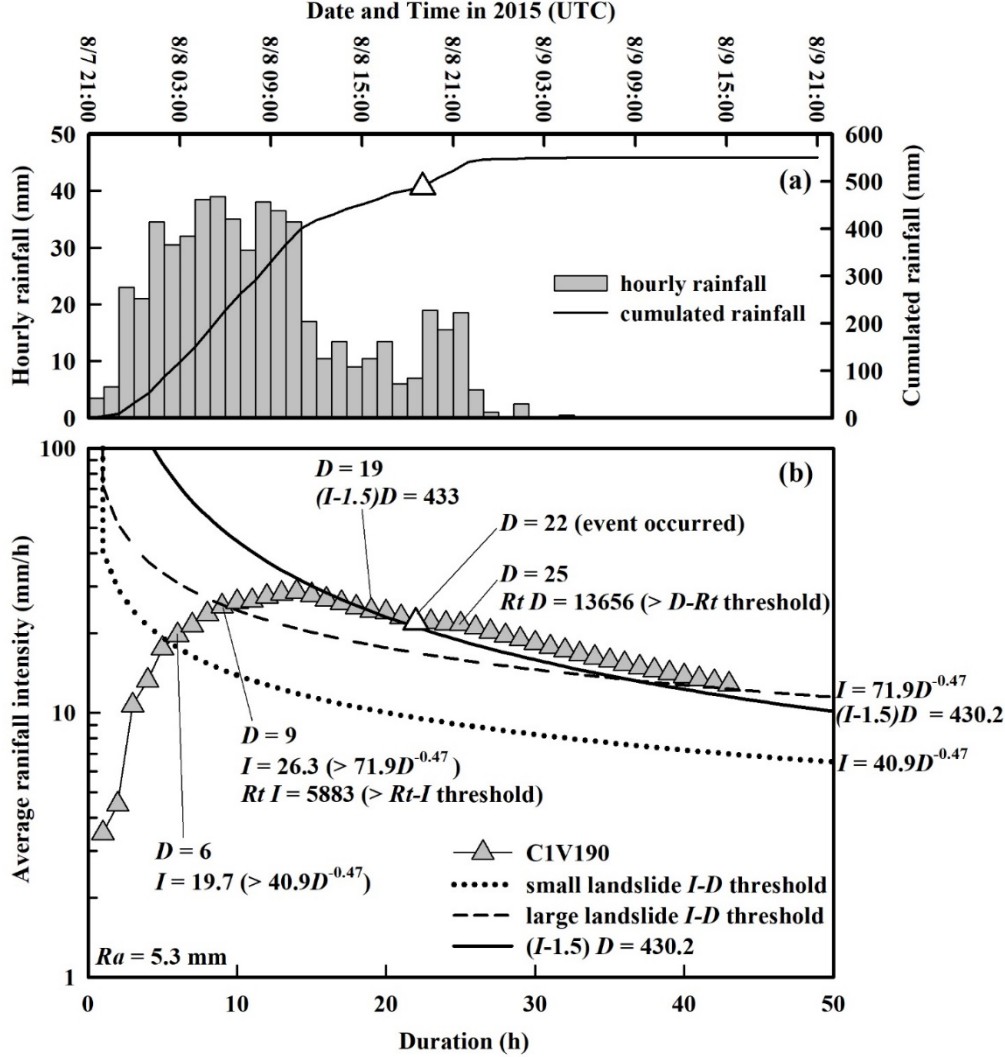

**Figure 8: (a) Hourly and cumulated rainfall record by rainfall station C1V190. The white triangle showed the occurrence time of the large landslide occurring in 2015. (b) The rainfall threshold of the critical height of water model issued the early warning three hours before the landslide initiated (white triangle).**

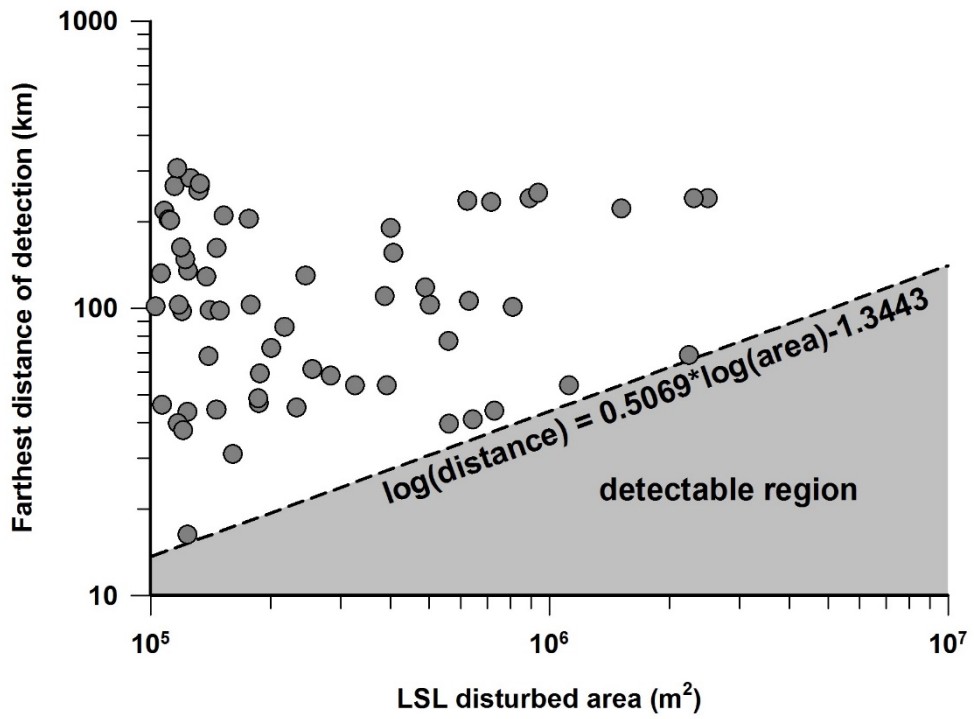

**Figure 9: Maximum distance of landslide-signal detection as a function of landslide-disturbed area. For a given large landslide, the seismic signal should be visible at all stations plotted beneath the curve.**