# Peer review of "Evaluating critical rainfall conditions for large-scale landslides by detecting event times from seismic records"

_Natural Hazards and Earth System Sciences, 2018_

## Referee Comment (RC1) · O. Marc (Referee) · 17 Jul 2018

Dear Editor,

I have read carefully the manuscript from Kuo et al.

Overall, the authors present an interesting, novel dataset (although relatively modest) and do a series of classic (rainfall threshold) and less classic (physically based threshold) analysis that can be worth publishing, but I think the discussion and some of the analysis need to be improved before that.

My detailed review with a summary, major and detailed comments are attached.

[Figure]

An important note is that the linear axis of the plots in FIg 6 does not allow to interpret easily the threshold in relation to the data. I would urge the authors to present the same data in a log log format as soon as possible so that I and other reviewer or interested reader can better judge and understand their result, and so that comments on this figure can be adequately made.

Please also note the supplement to this comment:
https://www.nat-hazards-earth-syst-sci-discuss.net/nhess-2018-126/nhess-2018-126-RC1-supplement.pdf

[Figure]

**Supplement:**

Review of Kuo et al., Evaluating critical rainfall conditios for large scale landslides

Kuo et al., present a landslide catalogue in Taiwan, obtained by rmote sensing, from which they extract 62 large landslides that can be accurately timed thanks to seismic detection, and compared to local rainfall gaging data. Then they assess which type of rainfall threshold could be derived for this dataset, including a threshold guided by physical considerations, and compare it to a dataset of smaller landslides in Taiwan. The paper ends with a rather unconvincing or unclear discussion on potential variabiliy of the thresholds and on issues sith seismic detection.

Overall, the authors present an interesting, novel dataset (although relatively modest) and do a series of classic (rainfall threshold) and less classic (physically based threshold) analysis that can be worth publishing, but the discussion and some of the analysis need to be improved before that.

**Major comment:**
1/ Timing is an issue but rainfall estimation as well. Notably because rain gage may be far from the landslides and not experiencing similar rainfall especially due to orographic effects. The author explain they only associate landslide with rainfall measured within 100km2. I think this is a good start but in the analysis it would be good to indicate ( by a color coding ?) the horizontal distance from the landslide, as well as to discuss difference in elevation between station and landslide median elevation for example. This would allow the authors to discuss uncertainty and the degree of reliability of rainfall estimates for the landslides.

2/ I think the attempt of the authors to define a threshold based on physical considerations is worth, but insufficient in th present form : the assumption and limit of the model lack validation/discussion, and the practical utility/validity of the model copared to purey empirical ones is poorly demonstrated. I give detailed proposition to test and refine the model, but in any case a more quantitative comparison of the validity of the different threshold seems important if the author want to underline the physical model has a path forward. I think also this part may benefit from being put in perspective compared to other work on physically based threshold. For example :

**-- Salciarini and Tamagni 2013, Physically based rainfall thresholds for shallow landslide initiation at regional scales.**
-- Papa et al., 2013, Derivation of critical rainfall thresholds for shallow landslides as a tool for debris flow early warning systems
-- Alvioli et al., 2014, scaling properties of rainfall induced landslides predicted by a physically based model.

3/ I think the discussion needs to be revised sigificantly.
The authors seek to discuss effects oncritical threshold that cannot really be assessed with the data they have, while several points are not really discussed : For example 1/ uncertainty on rainfall parameters, 2/ the added value of seismic dating of landslide and its limit (size of landslide distance from stations (currently section 5.3 needs significant clarification) , 3/ The value of the critical rainfall volume : how better compare with other, how to determine or constrain I0 etc

4/ Last, I strongly suggest the authors to define variable names for antecedent rainfall (e.g. Ra), cumulated rainfall (e.g. Rc) to later compare with Rt (Rt = Rc + Ra) and to be consistent in text and figure when they talk about rainfall amount.

**Line by Line comments :**

P2 L 5 : LSL / SSL : this is heavy and makes the draft harder to read. Why not simply use small and large landslide and indicating the boundary is at 0.1km2 ?

 P2 L21 : State in the text how was estimated the occurrence time. Based on peak rainfall correct ?
In Fig 1 Caption you say that in general peak rainfall intensity is used.This may go int the main text, with one or two references. Indeed, simple groundwater modelling ( e.g. Wilson and Wieczorek 1995) could estimate soil moisture based on the rainfall data and find a maximal pore pressure after the peak rainfall. Other simple modelling approach or assumption may give different estimation times.

P2 L34 : Fractural geological conditions >> Fratcured rock mass
P2 L35 : slope disasters >> I would suggest slope failures , more general (here and at other place in the text)

P3 L21 : By a rainstorm (which one ?) or by the Morakot typhoon ? Please clarify.

P3 L25 : end of the setnece unclear. Main factor to separate SSL from LSL or to relate to rainfall triggerring ? If so how ?

P3 L 30 : Ok the triangular signature is typical, but could you cite and discuss what are other typical properties ?  I know there are quite some papers discussing how to detect and classify landslides based on various properties of the spectrogram or of the waveform.

P4 L 3 : Only now we learn that the landslide mapping was done between 2009 and 2014. Please indicate it at the start of the mapping section.

P4 L 35 : Could you give an estimate of how often the location point and landslide maps matched ? And what was the maximal acceptable offset from a mapped landslide ?

P5 L 4 : Need some reference for that : the track does not necessarily say so much given the size of the diameter of typhoons are some times similar to Taiwan island size... And the windward slope is not obvious. If you refer to orographic effects say it clearly, but this also occur ar large scale not a fine scale.

P5 L5-10 : Very true indeed. Another important point may be the altittude of the gauging station and of the upper part of the landslide. If the gage is near the river at the outlet of the 100km² catchment possibly 500m or more below slopes where landslide happen the rainfall may be quite different.

P5  L 14 : Say if this is your definition ( we define the beginining of a rain event) or a general one (then cite other studies.

P5 L18-20 : I understand it is hard to choose objectively which time should be considered for antecedent rainfall, but an arbitrary threshold without temporal weighting seems disingenuous... It is fair to use the official definition but what about testing a coupd other antecedent rainfall conditions : for example the cumulated rain over 3 or 5 days. Or a weighted sum over the 10 preceding days ( with weight decreaseing with time before the event).

P6 L 4-7 : How was the occurrence time obtained for SSL ? Not by seismic means ? SO how accurate are these times ? Are we back to the same uncertainties as shown in Fig 1 ? Authors should clarify that.

P6 : Subsection 2.4 : missing "l", >> water model ?

P6 EQ 1  and 2 : ok but the assumption C' = 0 maybe quite a big one , especially for large bedrock landslides... Need to be discussed at some point, because Qc would be larger with non zero C.

P6 EQ 4 : Qc is actually the height of saturated regolith above the failure plane, in mm. Maybe clearer than calling it a critical volume. Note that in EQ 3 it is a critical height. But in EQ 4 it is simply a height assuming I0 is correctly estimated.
Another key issue is that this equation does not account for the antecedent rainfall. As I and D are for the triggering storm only, correct ?
Finally, I do not see why the authors assume a linear drainage. Most hydrological simple model of soil drainage (backed up by theory and observations) show a non linear drainage rate, where drainage increase with the amount of water in the soil (e.g., Wilson and wieczorek, 1995). I think the uthors should discuss this choice here or in discussion. Ths model is very easy to implement and use to obtain soil water level, only requiring the hourly estimate of rainfall and an assumed drainage parameters. I think it may be a interesting addtion to the paper to really make the authors model physical.
I note that a number of recent attempt to model physically landslide threshold (cf major comments) should be mentionned and discussed here and/or in discussion these models and how they compare to the author proposition.

P7 L5 : "their slope angles"
Do you mean the mean slope within the landslide body ?

P7 L7 : " This increase was most likely due to the fact that during the extremely heavy rainfall of Typhoon Morakot in 2009, more than 2,000 mm precipitated in four days, causing numerous landslides on lower slopes and reducing the stability of the steeper slopes in the following years."

>> I do not think this claim is supported by the data of Fig 4 : First in 2009 Morakot did not seem to be so different from 2005-2008 in terms of slope distribution. 2nd it only affected the southern half of the distribution of 2010-2014. If the hypothesis of the author is true, comparing only pre 2009 amd post 2009 in the Morakot area only (i.e. southern half of the dataset) would yield an even more pronounced shift, while the northenr half should have no shift. I invite the authors to check and show this to support their claim.
Alternatively they should try to check that statistical uncertainties may not be responsible for shift, and it would be interesting to compute a confidence interval on each histogram.
Last point, either if morakot did perturb the slope distribution the author need to clarify their argument, as it is not obvious how failing gentle slope would weaken steeper slopes ( as a start the author could try to demonstrate that failing slopes in 2010-2014 are spatially related to 2009 failures)

P7 L18-20: Yes probably.

P7 L 24-26 : Not clear. To clarify.

P7 L 26 : Could you explain with some more details how these 62 LSL were obtained ? Is it the combination of neary gages and seismic signal quality ? Anything else ? One sentence for recalling the reader of thr criteria used would be helpful.

P8L25 : Interesting, but size is not the only difference with these other thresholds.
The fact you focussed on large landslides, requiring higher total rainfall, and thus higher I-D lines is likely contributing. However, how much of the difference could be due to seismic dating ? To the regional characteristics of the landslide ( as some threshold are global, other taiwanese or japanese). I think these should be menionned here or in discussion, because your threshold for SSL is also much larger than most other threshold, and these SSL are more siilar in size  to past study.

P9 L 1-2 : it was determined that Rt–D analysis could be used effectively to distinguish SSLs from LSLs.
>> I think it is very interesting to see in Fig 5B that the landslide size groups shift from small for relatively short duration and low rainfal amount to large landslides for long and very large cumulative rainfall.

P9 L8 : "conditions for SSLs included high average rainfall intensity but relatively low cumulated rainfall"
>> You plot Rt that is the total effective rainfall in Fig 5. So do SSL have low cumulated rainfall or low Rt or both ( if Ra is low...)

In any case this plot is also quite interesting, as it matches well the theoretical expectations (Van asch 1999, Iverson 2000) stating that very large landslides will require high cumulated rainfall (unlikely to accumulate over short timescales) while small landslides may be caused by  transient pulse of water accumulation in the shallow regolith relating to very high intensity, but that do not need to cumulate large amount of water.

P9 L14-15 : Not only Wieczorek and Glade could cited here. Van asch 1999, Iverson 2000 discussed that earlier.

P9 L20 : This seems like a very crude approach. I would strongly encourage the author to have a Compute Qc based on an actual estimation of the landslide slope and the  landslide depth : Using Larsen 2010 or a local Area Depth relation from Taiwanese dataset (Chen 2013 ) the authors could use A to derive Z and thus obtain a more realistic extimate of Qc as a function of Z and the mean slope. The effect of small variations in porosity or friction angle could also be computed and shown.

I understand you want a single average threshold to compare to a population. Nevertheless you can make an almost individual prediction of each large landslide ( with Depth and Slope) and compare it to uniquely constrain rainfall information, thanks to your seismic dating. I think it would be worth checking the validity of the model this way, and potentially refining the drainage model that seems critical to really obtain a physically based threshold.

P9 L 23 -25: Is this curve allowing to better predict the LSL compared to the other plots in Fig 5 ( Especialy I – Rt or I-D?) Same question for the separation from SSL/LSL . The authors should provide some statistices confirming that this model is better than a Rt -I for example. Log Logplot is absolutely necessarry for all plot.
Further, the very low drainage found by the authors, mean their threshold is almost ID ~452 or R~452. And indeed a vertical line in  the I -Rt graph at about 500 may be as good...

P10L14 : If so you should observe a larger fraction of the LSL in 2010-2014 neighborih a 2005-2009 landslide, compare to LSL in 2005-2009 bein the reactivation of older landslides. Given the small dataset (62?), I encourage the authors to check each LSL and report the proportion of reactivation

before and after 2009. Then they can support and discuss this hypothesis.

P10 Section 5.1 and 5.2 Strange writing: the authors oscillate between presenting new result about shift between threshold for different subset and then concluding that they are insignificant. Based onFig 7 and 8 I do think the dataset of the author is insufficient to discuss these two topics and I would strongly suggest the author to remove these two sections ( or just mention rapidly that sub dividing the the dataset does not give clear difference and send Fig 7 and 8 in Supplement.) ans give more space to discussing other points, like their critical rainfall model, or the uncertainties on rainfall.

P11 section 5.3 : maye interesting but Fig 9 is too confusing. So I suspect text and Fig 9 should be clarified a bit.
P11 Eq 5 :  to discuss validity and limits of EQ 5 it should be made clearer how ( empirically?) and with which dataset/environment  this relationship was obtained.

Fig 3 : closest station is MASB (in the caption) or SGSB (in the map) ? It means 90% of the landslide and seismic signal

Fig 5 : The last panel is not very clear : Cumulated rainfall is the total rainfall in the triggering storm. Antecedent rainfall has no reason to be compared directly with landslide occurrence, but only when summed with the cumulated rainfall. So why not show Rt the total effective rainfall together with Rc the cumulative rainfall ( Given that Rt>= Rc it should be easy to visualize).

Fig 6 : Log Log scale is needed on all panel. Right now we do not see clearly the position of the different datapoints.

Fig  7 and 8 : I do not believe any of the subset can be significantly distinguished. What is driving the (small) difference in threshold curve is only 1 or 2 points out of each subset ( that seems to be15-25 points). These low points shift the threshold while the bulk of each popuation do not seem different in any way. I am convinced this can only be due to chance and not to a shift of the whole population.
I am even surprised that the curve are so low because if they are the 5% exceedance probability ~ 1point should be left out in subset of ~20...

Fig 9 : I really tried, but did not understand it... I got that the line, is an empirical estimation of the distance at which station should be able to detect a landslide  of a given size.
What are the points ? The 62 LSL ? If yes why are they all above the line ? Does that mean only distant station detect the slides? I can believe for some but not the whole dataset, and this seems contradictory with Fig 3

References not used in the manuscript

-- Wilson and Wieczorek 1995,  **Rainfall** thresholds for the initiation of debris flows at La Honda, California
-- Iverson, 2000, Landslide triggering by rain infiltration
-- Van asch et al., 1999, A view on some hydrological triggering systems in **landslides**
Larsen et al., 2010, **Landslide** erosion controlled by hillslope material

---

## Referee Comment (RC2) · Anonymous Referee #2 · 30 Jul 2018

**General comments**

The paper "Evaluating critical rainfall conditions for large-scale landslides by detecting event times from seismic records" is a very interesting paper with original approach. The combination of the tools and methods to define rainfall threshold to landsliding is interesting and the several steps of the analysis are presented. However, the reader can be lost in the used databases, in particular between what concerns the 2009 typhoon analysis and the rest of the chronical. The results can be discussed (detection of only 62 landslides, thresholds between 500/300mm…), or justified by figures completed (see below comments on the figures).

**Specific comments**

P2 L21: the event of 2009 is the only one mentioned, for the moment we can think that the research only focus on this event.

P3 L8: Date of the images? Number? Mapping only for the 2009 event.

P3 L23: Why 0.1km² Is it the limit of the automatic detection based on SPOT images? How many landslides were detected?

P3 L26: How we consider the progressive instability and the signal before the main failure?

P4 L3: Now we don't care about 2009 event. Why 2005-2014? What was the aim of 2009?

P4 L11: "only events with obvious signature", do you mean the 62 landslides in the fig.1? can you develop the characteristics of the signal that you can highlight with these 62 events?

P4 L19: how can you consider the lag time between rainfall / soil saturation… and failure?

P4 L29: I think the chosen method can be shortly developed here.

P4 L35: there is only Xiaolin landslide in this figure. So you focus on 2009 events? do you compared all the landslides detected by remote sensing (fig.1 is it landslides detected by seismic signal of remote sensing: to clarify) (how many by remote sensing?) with the seismic signal? an example of signal related to a smaller event than Xiaolin would be interesting (fig.3).

Do the SSLs have a significant signal also?

P5 L3: Now you are studying events between 2005-2014? It is a little bit confusing. How many typhoon events? don't you consider previous smaller rainfall events that could affect the mechanical properties of the slopes?

P5 L6: How can you consider the topographic, orographic effects?

P5 L10: 100km² is already large catchment.

P5 L2: rain event = typhoon?

P6 L4: 193 small landslides for which period?

P6 L15: EQ1 cohesion here is only considered for a discontinuity (C = 0)? Or for the specific material?

P7 L9/22: lower slopes VS steeper slopes… upslope VS downslope? Can you explain it? Is it a regressive erosion of the slope?

P7 L9/24 & 26: landslides for 2009 event? Or the detection of 62 landslides grounded on seismic signal among 686 inventoried landslides? What is the landslide seismic magnitude?

P8 L4: what about SSL?

**Discussion:** The discussion is interesting because it puts the results in perspective. Nevertheless, some points have to be clarify.

5.1. The authors highlight the fact that critical rainfall to trigger landslides has decreased since 2010 (500mm to 300mm) according to the results fig. 7. How many events the threshold is based on? The figure 7 is not so evident.

To explain these results, the authors question the Morakot typhoon. Was it an exceptional hydro-climatic event? The other solution is that instabilities induced by the 2009 typhoon are responsible of recent landslides. This idea should be developed here, and maybe associated to a map of the landslides scars (delineation of the departure areas) and differentiated according to the periods of the triggering…

5.2. The authors mention the fact that landslides occurred several types of rocks with different geotechnical behaviors, but the chosen geotechnical parameters (table 1) are identical. Why?

Effective rainfall, and rainfall duration thresholds according to the rock types are not clear in the figure 8, could another statistical analysis put the conclusion of the authors in obvious fact?

**Figure**

Fig. 2. Add legend for the detected landslide: Is it detected by seismic signal analysis? Is the point, the centroid of the landslide? Why not the delineation of the landslide body? Dates of the both satellite images here.

Fig. 3. Location of the detected landslides in 2009? Is there other spectrogram for previous landslides? Or after 2009? or associated to another landslide triggered in 2009: X spectrogram for 1 landslide. The star is the location defined with which seismic station?

Fig. 4. Maybe with the topography visible on the map?

Fig. 7. A) Is there only 1 event for the lowest limit?

---

## Author Comment (AC1) · 24 Aug 2018

RC #1: Kuo et al., present a landslide catalogue in Taiwan, obtained by remote sensing, from which they extract 62 large landslides that can be accurately timed thanks to seismic detection, and Compared to local rainfall gaging data. Then they assess which type of rainfall threshold could be derived for this dataset, including a threshold guided by physical considerations, and compare it to a dataset of smaller landslides in Taiwan. The paper ends with a rather unconvincing or unclear discussion on potential variability of the thresholds and on issues sith seismic detection. Overall, the authors present an interesting, novel dataset (although relatively modest) and do a series of classic

(rainfall threshold) and less classic (physically based threshold) analysis that can be worth publishing, but the discussion and some of the analysis need to be improved before that.

R: The authors very much appreciate the constructive feedback of the reviewer – it has certainly helped the authors improve this manuscript.

*Figure and Line by line reply have been provided with the supplementary file. Please see the attached material.

Major comment

1. Timing is an issue but rainfall estimation as well. Notably because rain gage may be far from the landslides and not experiencing similar rainfall especially due to orographic effects. The author explain they only associate landslide with rainfall measured within 100km2. I think this is a good start but in the analysis it would be good to indicate ( by a color coding ?) the horizontal distance from the landslide, as well as to discuss difference in elevation between station and landslide median elevation for example. This would allow the authors to discuss uncertainty and the degree of reliability of rainfall estimates for the landslides.

R: The authors appreciate the reviewer's constructive suggestion. The spatial information (distance and elevation) of each used rain gauge station will be added to supplementary materials as Table S1. The effect of rain gauge distribution over the accuracy of rainfall has been assessed using gauge observation in a 35 km $\times$ 50 km region of south Taiwan (Fig. S1). The amounts of daily rainfall during 2009 Typhoon Morakot (8/6-8/11) recorded at 19 rain gauge stations were selected to validate the accuracy of rainfall. At first, the amounts of daily rainfall were interpolated to 01V040 station using IDW methods. The errors between measurements and interpolated data were smaller than 15 %. It indicates IDW method can be used to interpolate rainfall to a selected location in our study area. Secondly, the amounts of daily rainfall at the central point of the 35 km $\times$ 50 km region were estimated. The errors of daily rainfall between

the central point and the nearest rain gauge station (01V040) were smaller than 10 % (0.5%-10% at different date). Besides, the correlation coefficients would keep at 90% as a distance between the central point and rain gauge stations less than 20 km, and even keep at 98% as a distance less than 10 km (Fig. S2). Therefore, in the study, an upper limit of basin area smaller than 100 km2 (10 km × 10 km was adopted to avoid a significant decrease of the accuracy of rainfall. The influence of topography on rainfall variability has been analyzed in the same 35 × 50 km region of south Taiwan. The highest station elevation is 1792 m a.s.l. at C1V270, and the lowest station elevation is 105 m a.s.l. at C10830. The standard deviation of station elevation is 561 m. The values of standard deviation of daily rainfall at the 19 stations were calculated, and less than 13% except for a high standard deviation, 45%, on sixth August (average daily rainfall less than 2 mm). The results demonstrated that high and even extreme rainfall are less influenced by elevation, while low and medium rainfall events are significantly influenced by elevation variation, with most of the rainfall appearing on high elevations. Similar results have also been reported by some previous studies (Sanchez-Moreno et al., 2014; Ge et al., 2017). Because the study only considered the rainfall events with total cumulated rainfall greater than 500 m, the elevation effect was ignored as selecting rain station.

2. I think the attempt of the authors to define a threshold based on physical considerations is worth, but insufficient in the present form: the assumption and limit of the model lack validation/discussion, and the practical utility/validity of the model compared to pure empirical ones is poorly demonstrated. I give detailed proposition to test and refine the model, but in any case a more quantitative comparison of the validity of the different threshold seems important if the author want to underline the physical model has a path forward. I think also this part may benefit from being put in perspective compared to other work on physically based threshold. For example: Salciarini and Tamagni 2013, Physically based rainfall thresholds for shallow landslide initiation at regional scales Papa et al., 2013, Derivation of critical rainfall thresholds for shallow landslides as a tool for debris flow early warning systems Alvioli et al., 2014, scaling

properties of rainfall induced landslides predicted by a physically based model

R: The authors appreciate the reviewer's suggestions and agree that the comparison of physically-based and statistically-based thresholds is needed. The study focused on rainfall conditions for triggering landslides in a wide (national scale) study area, a purely physical model may be not suitable. We would like to call it a mixed physically- and statistically-based model. The rainfall threshold using a mixed physically- and statistically-based model in the study will be compared with others using physically-based models. The relative discussion will be added to the text as below. "In general, physically-based models are easy to understand and have high predictive capabilities. However, they depend on the spatial distribution of various geotechnical data (cohesion, friction coefficient, permeability coefficient, etc.) which are very difficult to obtain. Statistically-based methods can include conditioning factors that influence slope stability which is unsuitable for physically based models. Statistically-based models rely on good landslide inventories and rainfall information. In the study, the Qc threshold for large landslides is estimated based on mixing physically- and statistically-based methods. Comparing to other physically-based I-D thresholds which were constructed based on artificial rainfall information for shallow landslides (Table S3), the Qc threshold proposed by the study seemed to be higher and more suitable for large landslides (Fig. 6).

In order to verify the application of the rainfall early warning model, we chose the typhoon Soudelor for demonstrating the forecasting performance. Typhoon Soudelor was one of the strongest storms in the world during 2015. It generated 1400 mm of rainfall in northeastern Taiwan and almost 1000 mm of rainfall in the southern mountainous area of Taiwan (Wei, 2017; Su et al., 2016). After completed the seismic signal analytical procedure, we obtained the occurrence time, 2015/8/8 18:59:50 (UTC), of a large landslide events located in southern Taiwan (Fig.7). This event was also detected by Chao et al. (2017) using a seismicity-based method. This event could be interpreted by six BATS stations and the location error was less than 6 km. We chose

the C1V190 rain station which situated in the same watershed and was 14.6 km away from the large landslide event. The typhoon Soudelor landfall in Taiwan on August 7, 2015, and dropped a cumulated rainfall and a maximum rainfall intensity of 546 mm and 39 mm/h on August 8 at the rain gauge station C1V190 (Fig. 8). The rainfall event began at 22:00 August 7 and last 26 hours while the landslide initiate at the 22th hour. Regarding this event, the average rainfall intensities exceeded the threshold were considered to be unstable, while those lower than the threshold were stable. According to the records, the landslide warning could be issued at 5:00 which was 4 hours earlier than the landslide initiated (Fig. 8). Then, comparing to the application of the I-D threshold which would issue the warning alert 12 hours before the landslide occurred, the Qc method seemed to be more suitable for large landslide early warning model.

3. I think the discussion needs to be revised significantly. The authors seek to discuss effects on critical threshold that cannot really be assessed with the data they have, while several points are not really discussed: For example 1/ uncertainty on rainfall parameters, 2/ the added value of seismic dating of landslide and its limit (size of landslide distance from stations (currently section 5.3 needs significant clarification) , 3/ The value of the critical rainfall volume : how better compare with other, how to determine or constrain I0 etc

R: The authors appreciate the reviewer's constructive comment. The section of discussion has been revised significantly. The revision includes: 1) The authors agree that uncertainty on rainfall parameters will influence the distribution of statistically-based rainfall data. In order to constrain the indeterminate variation of rainfall threshold analyses, a consistent process of calculating rainfall data with a standard of station selection has to be constructed. In the study, we tested the accuracy of rainfall data and used a consistent calculation method for rainfall parameters carefully. Therefore, the variation of rainfall parameters (I, D, and Rt) could be under control. The further discussion will be added to text. 2) The statement of detection limitation will be modified to make the point clear. 3) The quantity of critical water volume (Qc) was estimated

using the physically-based model proposed by Keefer. Subsequently, the threshold equation, (I-I0)*D=Qc, was adopted to fixed the lower boundary of rainfall data in the I-D plot. The value of I0 was estimated using the same statistically-based method with I-Rt threshold. The value of 1.5 was obtained as the exceeding probability of 5%. We would like to call it a mixed physically- and statistically-based model. The mixed model could recover the limitation while we just used a purely physically-based model or a purely statistically-based model. The modified illustration will be added to the test.

4. Last, I strongly suggest the authors to define variable names for antecedent rainfall (e.g. Ra), cumulated rainfall (e.g. Rc) to later compare with Rt (Rt = Rc + Ra) and to be consistent in text and figure when they talk about rainfall amount.

R: Thanks for the suggestion. The variable names have been modified according to the suggestions.

Please also note the supplement to this comment:
https://www.nat-hazards-earth-syst-sci-discuss.net/nhess-2018-126/nhess-2018-126-AC1-supplement.pdf

**Supplement:**

**RC #1: Kuo et al., present a landslide catalogue in Taiwan, obtained by remote sensing, from which they extract 62 large landslides that can be accurately timed thanks to seismic detection, and compared to local rainfall gaging data. Then they assess which type of rainfall threshold could be derived for this dataset, including a threshold guided by physical considerations, and compare it to a dataset of smaller landslides in Taiwan. The paper ends with a rather unconvincing or unclear discussion on potential variabiliy of the thresholds and on issues sith seismic detection.**

**Overall, the authors present an interesting, novel dataset (although relatively modest) and do a series of classic (rainfall threshold) and less classic (physically based threshold) analysis that can be worth publishing, but the discussion and some of the analysis need to be improved before that.**

R: The authors very much appreciate the constructive feedback of the reviewer – it has certainly helped the authors improve this manuscript.

**(1) Major comment**

**1. Timing is an issue but rainfall estimation as well. Notably because rain gage may be far from the landslides and not experiencing similar rainfall especially due to orographic effects. The author explain they only associate landslide with rainfall measured within 100km². I think this is a good start but in the analysis it would be good to indicate ( by a color coding ?) the horizontal distance from the landslide, as well as to discuss difference in elevation between station and landslide median elevation for example. This would allow the authors to discuss uncertainty and the degree of reliability of rainfall estimates for the landslides.**

R: The authors appreciate the reviewer's constructive suggestion. The spatial information (distance and elevation) of each used rain gauge station will be added to supplementary materials as Table S1.

The effect of rain gauge distribution over the accuracy of rainfall has been assessed using gauge observation in a 35 km × 50 km region of south Taiwan (Fig. S1). The amounts of daily rainfall during 2009 Typhoon Morakot (8/6-8/11) recorded at 19 rain gauge stations were selected to validate the accuracy of rainfall. At first, the amounts of daily rainfall were interpolated to 01V040 station using IDW methods. The errors between measurements and interpolated data were smaller than 15 %. It indicates IDW method can be used to interpolate rainfall to a selected location in our study area.

Secondly, the amounts of daily rainfall at the central point of the 35 km × 50 km region were estimated. The errors of daily rainfall between the central point and the nearest rain gauge station (01V040) were smaller than 10 % (0.5%-10% at different date). Besides, the correlation coefficients would keep at 90% as a distance between the

central point and rain gauge stations less than 20 km, and even keep at 98% as a distance less than 10 km (Fig. S2). Therefore, in the study, an upper limit of basin area smaller than 100 km$^2$ (10 km × 10 km was adopted to avoid a significant decrease of the accuracy of rainfall.

The influence of topography on rainfall variability has been analyzed in the same 35 km × 50 km region of south Taiwan. The highest station elevation is 1792 m a.s.l. at C1V270, and the lowest station elevation is 105 m a.s.l. at C10830. The standard deviation of station elevation is 561 m. The values of standard deviation of daily rainfall at the 19 stations were calculated, and less than 13% except a high standard deviation, 45%, on sixth August (average daily rainfall less than 2 mm). The results demonstrated that high and even extreme rainfall are less influenced by elevation, while low and medium rainfall events are significantly influenced by elevation variation, with most of the rainfall appearing on high elevations. Similar results have also been reported by some previous studies (Sanchez-Moreno et al., 2014; Ge et al., 2017). Because the study only considered the rainfall events with total cumulated rainfall greater than 500 m, the elevation effect was ignored as selecting rain station.

[Figure]

Fig. S1. The distribution of rain gauge stations and the location of the central point of the testing area for validating the influence of the distance between rain gauge and a given point.

[Figure]

Fig. S2. Variation of correlation coefficient

Reference

Mishra, A.K. (2013) Effect of rain gauge density over the accuracy of rainfall: a case study over Bangalore, India. SpringerPlus, 2, 311.

Sanchez-Moreno, J.F., Mannaerts, C.M., and Jetten, V. (2014) Influence of topography on rainfall variability in Santiago Island, Cape Verde. International Journal of Climatology, 34, 1081-1097.

Ge, G., Shi, Z., Yang, X., Hao, Y., Guo, H., Kossi, F., Xin, Z., Wei, W., Zhang, Z., Zhang, X., Liu, Y., and Liu, J. (2017) Analysis of Precipitation Extremes in the Qinghai-Tibetan Plateau, China: Spatio-Temporal Characteristics and Topography Effects. Atmosphere, 8(7), 127, doi:10.3390/atmos8070127.

**2. I think the attempt of the authors to define a threshold based on physical considerations is worth, but insufficient in the present form: the assumption and limit of the model lack validation/discussion, and the practical utility/validity of the model compared to pure empirical ones is poorly demonstrated. I give detailed proposition to test and refine the model, but in any case a more quantitative comparison of the validity of the different threshold seems important if the author want to underline the physical model has a path forward. I think also this part may benefit from being put in perspective compared to other work on physically based threshold. For example:**

**Salciarini and Tamagni 2013, Physically based rainfall thresholds for shallow landslide initiation at regional scales**

**Papa et al., 2013, Derivation of critical rainfall thresholds for shallow landslides as a tool for debris flow early warning systems**

**Alvioli et al., 2014, scaling properties of rainfall induced landslides predicted by a physically based model.**

R: The authors appreciate the reviewer's suggestions and agree that the comparison of physically-based and statistically-based thresholds is needed. The study focused on rainfall conditions for triggering landslides in a wide (national scale) study area, a purely physical model may be not suitable. We would like to call it a mixed physically- and statistically-based model. The rainfall threshold using a mixed mixed physically- and statistically-based model in the study will be compare with others using physically-based models. The relative discussion will be added to the text as below.

"In general, physically-based models are easy to understand and have high predictive capabilities. However, they depend on the spatial distribution of various geotechnical data (cohesion, friction coefficient, permeability coefficient, etc.) which are very difficult to obtain. Statistically-based methods can include conditioning factors that influence slope stability which are unsuitable for physically based models. Statistically-based models rely on good landslide inventories and rainfall information. In the study, the $Q_c$ threshold for large landslides is estimated based on mixing physically- and statistically-based methods. Comparing to other physically-based *I-D* threshold which were constructed based on artificial rainfall information for shallow landslides (Table S3), the $Q_c$ threshold proposed by the study seemed to be higher and more suitable for large landslides (Fig. 6).

In order to verify the application of the rainfall early warning model, we chose the typhoon Soudelor for demonstrate the forecasting performance. Typhoon Soudelor was one of the strongest storms in the world during 2015. It generated 1400 mm of rainfall in northeastern Taiwan and almost 1000 mm of rainfall in southern mountainous area of Taiwan (Wei, 2017; Su et al., 2016). After completed the seismic signal analytical procedure, we obtained the occurrence time, 2015/8/8 18:59:50 (UTC), of a large landslide events located in southern Taiwan (Fig.7). This event was also detected by Chao et al. (2017) using a seismicity-based method. This event could be interpreted by six BATS stations and the location error was less than 6 km. We chose the C1V190 rain station which situated in the same watershed and was 14.6 km away from the large landslide event. The typhoon

Soudelor landfall in Taiwan on August 7, 2015 and dropped a cumulated rainfall and a maximum rainfall intensity of 546 mm and 39 mm/h on August 8 at the rain gauge station C1V190 (Fig. 8). The rainfall event began at 22:00 August 7 and last 26 hours while the landslide initiate at the 22th hour. Regarding this event, the average rainfall intensities exceeded the threshold were considered to be unstable, while those lower than the threshold were stable. According to the records, the landslide warning could be issued at 15:00 which was 4 hours earlier than the landslide initiated (Fig. 8). Then, comparing to the application of *I-D* threshold which would issue the warning alert 12 hours before the landslide occurred, the $Q_c$ method seemed to be more suitable for large landslide early warning model.

[Figure]

Fig. 6

[Figure]

Fig. 7

[Figure]

Fig. 8

Table S3. Physically-based I-D threshold considered in the study

|   | Reference | Equation | Study area |
|---|-----------|----------|------------|
| 1 | Salciarini et al. (2012) | $I = 276.2D^{-0.99}$ | Model |
| 2 | Chen et al. (2013) | $I = 24.4D^{-0.28}$ | Taiwan |
| 3 | Napolitano et al. (2016) | $I = 287.8D^{-1.09}$ | southern Italy |

Reference:

Chen, Y. H., Tan, C. H., Chen, M. M., and Su, T. W. (2013) Estimation of rainfall threshold for regional shallow landslides in a watershed. Journal of Chinese Soil and Water Conservation, 44(1), 87-96.

Salciarini, D., Tamagnini, C., Conversini, P., Rapinesi, S. (2012) Spatially distributed rainfall thresholds for the initiation of shallow landslides. Nat. Hazards 61, 229–245.

Napolitano, E., Fusco, F., Baum, R. L., Godt, J. W., and De Vita, P. (2016) Effect of antecedent-hydrological conditions on rainfall triggering of debris flows in ash-fall pyroclastic mantled slopes of Campania (southern Italy). Landslides, 13, 967–983.

Chao, W. A., Wu, Y. M., Zhao, L., Chen, H., Chen, Y. G., Chang, J. M., & Lin, C. M. (2017). A first near real-time seismology-based landquake monitoring system. Scientific Reports, 7, 43510.

Su, Y.F., Chen, W.B., Fu, H. S., Jang, J. H., Chang, C. H. (2016). Application of Rainfall Forecasting to Flood Management --A Case Study of Typhoon Soudelor. Journal of Disaster Management, Vol.5, No.2, pp. 1-17 (in Chinese)

Wei, C. C. (2017). Examining El Niño–Southern Oscillation effects in the subtropical zone to forecast long-distance total rainfall from typhoons: A case study in Taiwan. Journal of Atmospheric and Oceanic Technology, 34(10), 2141-2161.

**3. I think the discussion needs to be revised significantly. The authors seek to discuss effects on critical threshold that cannot really be assessed with the data they have, while several points are not really discussed: For example 1/ uncertainty on rainfall parameters, 2/ the added value of seismic dating of landslide and its limit (size of landslide distance from stations (currently section 5.3 needs significant clarification) , 3/ The value of the critical rainfall volume : how better compare with other, how to determine or constrain I0 etc**

R: The authors appreciate the reviewer's constructive comment. The section of discussion has been revised significantly. The revision includes:

1) The authors agree that uncertainty on rainfall parameters will influence on the distribution of statistically-based rainfall data. In order to constrain the indeterminate variation of rainfall threshold analyses, a consistent process of calculating rainfall data with a standard of station selection has to be constructed. In the study, we tested the accuracy of rainfall data and used a consistent calculation method for rainfall parameters carefully. Therefore, the variation of rainfall parameters ($I, D$, and $Rt$) could be under control. The further discussion will be added to text.

2) The statement of detection limitation will be modified to make the point clear.

3) The quantity of critical water volume (Qc) was estimated using the physically-based model proposed by Keefer. Subsequently, the threshold equation, $(I-I_0) \times D = Qc$, was adopted to fixed the lower boundary of rainfall data in the $I$-$D$ plot. The value of $I_0$ was estimated using the same statistically-based method with $I$-$Rt$ threshold. The

value of 1.5 was obtained as the exceeding probability of 5%. We would like to call it a mixed physically- and statistically-based model. The mixed model could recover the limitation while we just used a purely physically-based model or a purely statistically-based model. The modified illustration will be added to the test.

4. **Last, I strongly suggest the authors to define variable names for antecedent rainfall (e.g. Ra), cumulated rainfall (e.g. Rc) to later compare with Rt (Rt = Rc + Ra) and to be consistent in text and figure when they talk about rainfall amount.**

R: Thanks for the suggestion. The variable names have been modified according to the suggestions.

**(2) Line by Line comments:**

1. **P2 L 5: LSL / SSL : this is heavy and makes the draft harder to read. Why not simply use small and large landslide and indicating the boundary is at 0.1km$^2$ ?**

R: Thanks for the suggestion. The origin term, large-scale landslide and small-scale landslide, have both replaced with "large landslide" and "small landslide", respectively.

2. **P2 L21: State in the text how was estimated the occurrence time. Based on peak rainfall correct? In Fig 1 Caption you say that in general peak rainfall intensity is used. This may go int the main text, with one or two references. Indeed, simple groundwater modelling (e.g. Wilson and Wieczorek, 1995) could estimate soil moisture based on the rainfall data and find a maximal pore pressure after the peak rainfall. Other simple modelling approach or assumption may give different estimation times.**

**R:** Thanks for the suggestions. The authors agree that more and more useful approaches have been developed to get the exact time information of landslide initiation. However, the approaches all depended on in-situ monitoring or other assumptions. So far, the most common and convenient way to assess a factor of rainfall intensity is still based on the peak rainfall intensity. The statement on peak rainfall intensity will be added to text with some references (i.g. Chen et al., 2005; Wei et al., 2006; Staley et al., 2013; Yu et al., 2013; Xue et al., 2016). This time recording standard is a method selected by relevant concerned department of Taiwan, because the lack of clear time about slope failure or debris flow. Therefore, this study uses this method to explain the misjudgment result caused by the lack of clear occurrence time (Chen et al., 2005).

Reference:

Chen, C. Y., Chen, T. C., Yu, F. C., Yu, W. H., and Tseng, C. C. (2005) Rainfall duration and debris-flow initiated studies for real-time monitoring. Environ Geol, Vol. 47, 715–724.

Staley, D., Kean, J. W., Cannon, S. H., Schmidt, K. M., and Laber, J. L. (2013) Objective definition of rainfall intensity–duration thresholds for the initiation of post-fire debris flows in southern California. Landslides, Vol. 10(5), 547–562.

Wei, F., Gao, K., Cui, P., Hu, K., Xu, J., Zhang, G., and Bi, B. (2006) Method of debris flow prediction based on a numerical weather forecast and its application. WIT Transactions on Ecology and the Environment, Vol. 90, 37-46.

Xue, X., and Huang, J. (2016) A rainfall and pore pressure thresholds for debris-flow early warning: The Wenjiagou gully case study. Nat. Hazards Earth Syst. Sci. Discuss., doi:10.5194/nhess-2016-149.

Yu, B., Li, L., Wu, Y., and Chu, S. (2013) A formation model for debris flows in the Chenyulan River Watershed, Taiwan. Natural Hazards, Vol. 68(2), 745–762.

**3. P2 L34: Fractural geological conditions >> Fratcured rock mass**

**R:** Thanks for suggestion. The sentence will be revised based on the suggestion.

**4. P2 L35: slope disasters >> I would suggest slope failures , more general (here and at other place in the text)**

**R:** Thanks for suggestion. The sentence will be revised based on the suggestion

**5. P3 L21: By a rainstorm (which one ?) or by the Morakot typhoon ? Please clarify.**

**R:** Here refers to landslides caused by heavy rain events, not only by a specific event, we will modify the statement to avoid confuse.

"…Landslides induced specifically by rainstorm events were distinguished by overlaying the pre- and post-event image mosaics…."

**6. P3 L25: end of the sentence unclear. Main factor to separate SSL from LSL or to relate to rainfall triggering? If so how?**

**R:** In the study, the landslide types were divided into large landslide and small landslide based on the size of landslide-disturbed area. The rainfall factors of each landslide were assessed after classifying. The main purpose in the study is to find the difference of rainfall thresholds between large and small landslides, but not to classify these two types of landslides by rainfall factor or rainfall pattern. The relative sentence will be revised to avoid confuse.

**7. P3 L 30: Ok the triangular signature is typical, but could you cite and discuss what are other typical properties ? I know there are quite some papers discussing how to detect and classify landslides based on various properties of the spectrogram or of the waveform.**

**R:** The authors thank the reviewer's suggestions. More deeply description on the features of landslide-induced seismic signals will be added to the text as bellows:

"…The seismic wave generated by landslide can be attributed to the shear force and loading on the ground surface as the mass moving downslope. Many studies have shown that the source mechanism of a landslide is highly complicated, and their seismic wave mainly consist of surface wave and shear wave, making it

difficult to distinguish *P* wave and *S* wave from station records (Lin et al., 2010; Suwa et al., 2010; Dammeier et al., 2011; Feng, 2011; Hibert et al., 2014). The onset of landslide seismic signal is generally emergent. Then, the seismic amplitude increases gradually above ambient noise level to peak ground motion, exhibiting a 'cigar' shape envelope. After the peak amplitude, most of landslide-generated seismic signals have relatively long decay time, on average about 70% of total signal duration (Norris, 1994; La Rocca et al., 2004; Surinach et al., 2005; Deparis et al., 2008; Schneider et al., 2010; Dammeier et al., 2011; Allstadt, 2013). In frequency domain, landslide-induced seismic energy was mainly distributed below 10 Hz, with a triangular shaped signature in spectrogram, due to an increase over time in high-frequency constituents (Surinach et al., 2005; Dammeier et al., 2011)."

Reference:

Allstadt, K. (2013). Extracting source characteristics and dynamics of the August 2010 Mount Meager landslide from broadband seismograms. Journal of Geophysical Research: Earth Surface, 118(3), 1472-1490. doi:10.1002/jgrf.20110.

Dammeier, F., Moore, J. R., Haslinger, F., and Loew, S. (2011). Characterization of alpine rockslides using statistical analysis of seismic signals. Journal of Geophysical Research, 116(F4). doi:10.1029/2011jf002037

Deparis, J., Jongmans, D., Cotton, F., Baillet, L., Thouvenot, F., and Hantz, D. (2008). Analysis of rock-fall and rock-fall avalanche seismograms in the French Alps. Bulletin of the Seismological Society of America, 98(4), 1781-1796. doi:10.1785/0120070082.

Feng, Z. (2011). The seismic signatures of the 2009 Shiaolin landslide in Taiwan. Natural Hazards and Earth System Science, 11(5), 1559-1569. doi:10.5194/nhess-11-1559-2011

Hibert, C., Ekström, G., and Stark, C. P. (2014). Dynamics of the Bingham Canyon Mine landslides from seismic signal analysis. Geophysical Research Letters, 41(13), 4535-4541. doi:10.1002/2014gl060592

La Rocca, M., Galluzzo, D., Saccorotti, G., Tinti, S., Cimini, G. B., and Del Pezzo, E. (2004). Seismic signals associated with landslides and with a tsunami at Stromboli volcano, Italy. Bulletin of the Seismological Society of America, 94(5), 1850-1867. doi:10.1785/012003238.

Lin, C. H., Kumagai, H., Ando, M., and Shin, T. C. (2010). Detection of landslides and submarine slumps using broadband seismic networks. Geophysical Research Letters, 37(22), n/a-n/a. doi:10.1029/2010gl044685

Norris, R. D. (1994). Seismicity of rockfalls and avalanches at 3 Cascade Range

volcanos - Implications for seismic detection of hazardous mass movements. Bulletin of the Seismological Society of America, 84(6), 1925-1939.

Schneider, D., Bartelt, P., Caplan-Auerbach, J., Christen, M., Huggel, C., and McArdell, B. W. (2010). Insights into rock-ice avalanche dynamics by combined analysis of seismic recordings and a numerical avalanche model. Journal of Geophysical Research, 115(F4). doi:10.1029/2010jf001734.

Surinach, E., Vilajosana, I., Khazaradze, G., Biescas, B., Furdada, G., and Vilaplana, J. M. (2005). Seismic detection and characterization of landslides and other mass movements. Natural Hazards and Earth System Sciences, 5, 791-798.

Suwa, H., Mizuno, T., and Ishii, T. (2010). Prediction of a landslide and analysis of slide motion with reference to the 2004 Ohto slide in Nara, Japan. Geomorphology, 124(3-4), 157-163. doi:10.1016/j.geomorph.2010.05.003.

**8. P4 L3: Only now we learn that the landslide mapping was done between 2009 and 2014. Please indicate it at the start of the mapping section.**

**R:** Thanks for the suggestion. The illustration will be revised to the text as below"

"To determine the locations and basic characteristics of large landslides occurring during 2005-2014, the landslide areas across the entire island of Taiwan were interpreted using SPOT-4 satellite…"

**9. P4 L35: Could you give an estimate of how often the location point and landslide maps matched? And what was the maximal acceptable offset from a mapped landslide?**

**R:** Once the seismic signals had the characteristics of landslide-induced ground-motions and were located in mountainous area, exceeding 90% of the signals could be paired with the landslides which were located in the vicinity of seismically-locating points, and the slope aspect were consistent with the direction of the trajectories of seismic signals. The average location error, or the distance between the actual and estimated location, was 10.9 km. The best location estimate was for the No. 40 event with an error of 0.5 km, while the worst location estimate was for No. 35 event with an error of 49.3 km.

**10. P5 L 4: Need some reference for that: the track does not necessarily say so much given the size of the diameter of typhoons are sometimes similar to Taiwan island size... And the windward slope is not obvious. If you refer to orographic effects say it clearly, but this also occur at large scale not a fine scale.**

**R:** The authors appreciate the kind suggestions. The statement will be modified based on the suggestions. Some useful reference will be added to text as the below:

Chen, C. S., and Chen, Y. L. (2003). The rainfall characteristics of Taiwan. Monthly Weather Review, 131(7), 1323-1341.

Sanchez-Moreno, J.F., Mannaerts, C.M., and Jetten, V. (2014) Influence of topography on rainfall variability in Santiago Island, Cape Verde. International Journal of Climatology, 34, 1081-1097.

**11. P5 L5-10: Very true indeed. Another important point may be the altitude of the gauging station and of the upper part of the landslide. If the gage is near the river at the outlet of the 100km² catchment possibly 500m or more below slopes where landslide happen the rainfall may be quite different.**

**R:** Thanks for comments. The reply has been addressed as the Q1 of major comment.

**12. P5 L 14: Say if this is your definition ( we define the beginning of a rain event) or a general one (then cite other studies.)**

**R:** Thanks for comments. The sentence will be revised as follows:

"… is generally defined as the time point when hourly rainfall exceeds 4 mm, and the rain event ends when the rainfall intensity remains below 4 mm/h for 6 consecutive hours. The critical rainfall condition for a landslide was calculated from the beginning…. (Jan and Lee, 2004; Lee, 2006)."

Reference:

Jan, C. D., and Lee, M. H. (2004). A debris-flow rainfall-based warning model. J Chin Soil Water Conserv, 35(3), 275-285.

Lee, M. H. (2006). The Rainfall threshold and analysis of Debris flows, Doctoral dissertation, National Cheng Kung University, Taiwan, ROC (in Chinese).

**13. P5 L18-20: I understand it is hard to choose objectively which time should be considered for antecedent rainfall, but an arbitrary threshold without temporal weighting seems disingenuous... It is fair to use the official definition but what about testing a coupd other antecedent rainfall conditions: for example, the cumulated rain over 3 or 5 days. Or a weighted sum over the 10 preceding days (with weight decreaseing with time before the event).**

**R:** Thanks for your suggestion. In this study we used a temporal weighting coefficient of 0.7 with weight decreasing with days before the event (Jan and Lee, 2004). The formula can be written as:

$$Ra = \sum_{i=1}^{7} 0.7^i * R_i$$

We will attach this in a later version.

**14. P6 L 4-7: How was the occurrence time obtained for SSL ? Not by seismic means? SO how accurate are these times? Are we back to the same uncertainties as shown in Fig 1? Authors should clarify that.**

**R:** The time records of the small landslides used in the study were reported by the disaster investigation report of the Soil and Water Conservation Bureau (SWCB) in Taiwan, but not obtained from seismic records. Most of the small landslides caused disasters and loss of life and property. In some cases, in-situ river steel cable or CCTV could record the time information. The clear illustration on the data source of small landslides will be added to the later version.

**15. P6: Subsection 2.4 : missing "l", >> water model ?**

**R:** Thanks for careful reviewing. The mistake will be revised in the text.

**16. P6 EQ 1 and 2: ok but the assumption C' = 0 maybe quite a big one , especially for large bedrock landslides... Need to be discussed at some point, because Qc would be larger with none zero C.**

**R:** We thanks reviewer's recommendation. Well development of detachment plane (e.g., sliding surface between sedimentary layers, connected joints, weathered foliation, etc.) have been widely considered as the geological conditions to occur a large landslide (Agliardi et al., 2001; Tsou et al., 2011). Therefore, in the study, the C' of the detachment plane is simply assumed as the value of zero to behave the critical situation of slope stability. The illustration of C' will be modified to the text.

> "…Well development of detachment plane (e.g., sliding surface between sedimentary layers, connected joints, weathered foliation, etc.) have been widely considered as the geological conditions to occur a large landslide (Agliardi et al., 2001; Tsou et al., 2011)."

Reference:

Tsou, C. Y., Feng, Z. Y., & Chigira, M. (2011). Catastrophic landslide induced by typhoon Morakot, Shiaolin, Taiwan. Geomorphology, 127(3-4), 166-178.

Agliardi, F., Crosta, G., & Zanchi, A. (2001). Structural constraints on deep-seated slope deformation kinematics. Engineering Geology, 59(1-2), 83-102.

**17. P6 EQ 4: Qc is actually the height of saturated regolith above the failure plane, in mm. Maybe clearer than calling it a critical volume. Note that in EQ 3 it is a critical height. But in EQ 4 it is simply a height assuming I0 is correctly estimated.**

**Another key issue is that this equation does not account for the antecedent rainfall. As I and D are for the triggering storm only, correct? Finally, I do not see why the authors assume a linear drainage. Most hydrological simple model of soil drainage (backed up by theory and observations) show a non linear drainage rate, where drainage increase with the amount of water in the soil (e.g., Wilson and wieczorek, 1995). I think the authors should discuss this choice here or in discussion. This model is very easy to implement and use to obtain soil water level, only requiring the hourly estimate of rainfall and an assumed drainage parameter. I think it may be an interesting addition to the paper to really make the authors model physical. I note that a number of recent attempt to model physically landslide threshold (cf major comments) should be mentioned and discussed here and/or in discussion these models and how they compare to the author proposition.**

**R:** Thanks for the valuable suggestions. The original naming of Qc in the manuscript is followed the Keefer (1984). We will revise the naming of Qc to critical water height. Practically, antecedent rainfall is not considered in the empirical/statistically-based *I-D* method.

**18. P7 L5: "their slope angles". Do you mean the mean slope within the landslide body?**

**R:** The slope angles mentioned in the study indicate the mean slope gradient before landslides occurred. The values of slope gradient were utilized to calculate Qc, therefore they should not be affected by landsliding. The slope angles were estimated with a 40 m digital elevation model (DEM) which was created before 2004. The illustration on slope angles will be modified as below.

"…., and their slope angles before the landslides occurred were …."

**19. P7 L7 : " This increase was most likely due to the fact that during the extremely heavy rainfall of Typhoon Morakot in 2009, more than 2,000 mm precipitated in four days, causing numerous landslides on lower slopes and reducing the**

stability of the steeper slopes in the following years."

I do not think this claim is supported by the data of Fig 4 : First in 2009 Morakot did not seem to be so different from 2005-2008 in terms of slope distribution. 2nd it only affected the southern half of the distribution of 2010-2014. If the hypothesis of the author is true, comparing only pre 2009 amd post 2009 in the Morakot area only (i.e. southern half of the dataset) would yield an even more pronounced shift, while the northern half should have no shift. I invite the authors to check and show this to support their claim.

Alternatively they should try to check that statistical uncertainties may not be responsible for shift, and it would be interesting to compute a confidence interval on each histogram. Last point, either if Morakot did perturb the slope distribution the author need to clarify their argument, as it is not obvious how failing gentle slope would weaken steeper slopes (as a start the author could try to demonstrate that failing slopes in 2010-2014 are spatially related to 2009 failures)

**R:** The authors appreciate reviewer's valuable comments. We agree the original sentence was unclear. The sentence will be revised as below"

"The number of landslides occurring on slope angles exceeding 40° slightly increased after 2010. Although the increase was quite slight, this increase was most likely due to the fact that during the extremely heavy rainfall of Typhoon Morakot in 2009, more than 2,000 mm precipitated in four days, causing a large number of landslides, and exhausting a lot of unstable slopes. Consequently, landslides transferred to occur on steeper slopes in the following years."

**20. P7 L 24-26: Not clear. To clarify.**

**R:** The resultant trace of two horizontal-component signals could be plotted. Comparing the direction of the resultant trace of a given landslide-induced seismic record with the slope aspect in the vicinity of locating point, we could eliminate the irrelevant landslides which have different slope aspects with the signal trajectory. The paragraph will be modified as follows:

"In addition to distance, the resultant trace of two horizontal-component signals could be plotted. Comparing the direction of the resultant trace of a given landslide-induced seismic record with the slope aspect in the vicinity of locating point, we could eliminate the irrelevant landslides which have different slope aspects with the signal traces. The ground motion traces of the signals have to be

correlated with the directions of movement of the landslides to reconfirm the matched large landslides."

21. **P7 L26: Could you explain with some more details how these 62 LSL were obtained? Is it the combination of near gages and seismic signal quality? Anything else? One sentence for recalling the reader of the criteria used would be helpful**

**R:** After obtaining the signal at the time of the landslide events, we use the locating method proposed by Chen et al. (2013) to locate the vibration source. Once a landslide is close to a locating point of seismic records and the slope aspect of the landslide is consistent with the direction of signal trajectory, the landslide can be considered as the source of the seismic signals.

22. **P8 L25: Interesting, but size is not the only difference with these other thresholds. The fact you focused on large landslides, requiring higher total rainfall, and thus higher I-D lines is likely contributing. However, how much of the difference could be due to seismic dating? To the regional characteristics of the landslide (as some threshold are global, other Taiwanese or Japanese). I think these should be mentioned here or in discussion, because your threshold for SSL is also much larger than most other threshold, and these SSL are more similar in size to past study.**

**R:** Thanks for suggestions. The study aims to use a seismicity method to get landslide timing for constructing rainfall thresholds, and to discuss which threshold is more suitable to give different warning for small and large landslides. Clearer illustration of the purpose of the study will be added to the introduction section as below:

"The study attempts to get the occurrence times of landslides by identifying landslide-generated seismic signals for constructing rainfall thresholds, and to clarify which threshold is more suitable to give different warning for small and large landslides."

23. **P9 L 1-2: it was determined that Rt–D analysis could be used effectively to distinguish SSLs from LSLs. I think it is very interesting to see in Fig 5B that the landslide size groups shift from small for relatively short duration and low rainfall amount to large landslides for long and very large cumulative rainfall.**

**R:** Thanks for comments. We will add the illustration to the modified manuscript.

24. **P9 L8: "conditions for SSLs included high average rainfall intensity but relatively low cumulated rainfall". You plot Rt that is the total effective rainfall in Fig 5. So do SSL have low cumulated rainfall or low Rt or both ( if Ra is low...)**

**In any case this plot is also quite interesting, as it matches well the theoretical expectations (Van asch 1999, Iverson 2000) stating that very large landslides will require high cumulated rainfall (unlikely to accumulate over short timescales) while small landslides may be caused by transient pulse of water accumulation in the shallow regolith relating to very high intensity, but that do not need to cumulate large amount of water.**

**R:** We thanks for comments. The statement will be revised as below:

"…conditions for small landslides included high average rainfall intensity but relatively low effective rainfall."

25. **P9 L14-15: Not only Wieczorek and Glade could cited here. Van asch 1999, Iverson 2000 discussed that earlier.**

**R:** Thank you for your suggestion. The reference, Van asch (1999) and Iverson (2000), will be cited and added to the modified manuscript.

Reference:

Van Asch, T. W., Buma, J., and Van Beek, L. P. H. (1999). A view on some hydrological triggering systems in landslides. Geomorphology, 30(1-2), 25-32.

Iverson, R. M. (2000). Landslide triggering by rain infiltration. Water resources research, 36(7), 1897-1910.

26. **P9 L20: This seems like a very crude approach. I would strongly encourage the author to have a Compute Qc based on an actual estimation of the landslide slope and the landslide depth: Using Larsen 2010 or a local Area Depth relation from Taiwanese dataset (Chen 2013) the authors could use**

**A to derive Z and thus obtain a more realistic estimate of Qc as a function of Z and the mean slope. The effect of small variations in porosity or friction angle could also be computed and shown.**

**I understand you want a single average threshold to compare to a population. Nevertheless, you can make an almost individual prediction of each large landslide (with Depth and Slope) and compare it to uniquely constrain rainfall information, thanks to your seismic dating. I think it would be worth checking**

**the validity of the model this way, and potentially refining the drainage model that seems critical to really obtain a physically based threshold.**

**R:** Thanks for valuable suggestions. The authors have tried the suggested approaches to recalculate Qc and to estimate the rainfall threshold. The revised rainfall threshold is $(I - 1.5) D = 430.2$. The relative paragraph and Fig. 6 will be modified in the later version of manuscript.

"The critical volume of water, $Q_C$, on sliding surface for each large landslide was estimated based on its slope gradient, depth (estimated by the equation: $Z = 26.14A^{0.4}$; Z: depth in m; A: disturbed area in m$^2$), and the geological material parameters of the study area (Table 1). The $Q_C$ value was inserted into $Q_C = (I - I_0) \cdot D$ to obtain an $I_0$ value for each large landslide. For the 62 detected landslide, the cumulative probability of 5% of Qc and $I_0$ values was taken as the critical values. The critical value of $I_0$ was 1.5, and the critical $Q_C$ was 430.2, which is more suitable for LSLs than for SSLs, and the threshold curve was rewritten as $(I - 1.5) \cdot D = 430.2$."

27. **P9 L 23 -25: Is this curve allowing to better predict the LSL compared to the other plots in Fig 5 (Especially I – Rt or I-D?) Same question for the separation from SSL/LSL. The authors should provide some statistics confirming that this model is better than a Rt -I for example. Log Log plot is absolutely necessary for all plot. Further, the very low drainage found by the authors, mean their threshold is almost ID ~452 or R~452. And indeed a vertical line in the I -Rt graph at about 500 may be as good...**

**R:** Thank you for your suggestion. The log-log figure will be modified based on suggestions. And a vertical line will be added in the *I-Rt* graph at about 500m

28. **P10 L14: If so you should observe a larger fraction of the LSL in 2010-2014 neighbored a 2005-2009 landslide, compare to LSL in 2005-2009 being the reactivation of older landslides. Given the small dataset (62?), I encourage the authors to check each LSL and report the proportion of reactivation before and after 2009. Then they can support and discuss this hypothesis**

**R:** Thank you for the valuable comments. The information on reactivity of 2010-2014 landslides will be provided in table S1. Once a 2010-2014 landslide was on the location of a 2005-2009 landslide, the landslides was classified as a reactivated landslide (marked as "R" in table S1).

29. **P10 Section 5.1 and 5.2 Strange writing: the authors oscillate between presenting new result about shift between threshold for different subset and then concluding that they are insignificant. Based on Fig 7 and 8 I do think the dataset of the author is insufficient to discuss these two topics and I would strongly suggest the author to remove these two sections (or just mention rapidly that sub dividing the dataset does not give clear difference and send Fig 7 and 8 in Supplement.) and give more space to discussing other points, like their critical rainfall model, or the uncertainties on rainfall.**

**R:** Thanks for comments. The authors agree the conceptual view of reviewer. The discussion needs more solid information and field investigation. Therefore, the part about the influence of rock types and an extreme event will be removed and replaced with in-deep comparison of different rainfall threshold.

30. **P11 section 5.3: maybe interesting but Fig 9 is too confusing. So I suspect text and Fig 9 should be clarified a bit.**

**R:** To determine the limits of large landslide detection distance as a function of event volume, we selected the farthest seismic station at which each event was detectable. An event was deemed detectable when we had selected the station for the distance-dependency analysis. The remaining results are shown on a plot of distance versus disturbed area (Fig. 9), where we can observe an upper detection limit described by equation 5. If, for a given event, a station plots in the lower right area below the dashed line (equation (5)), the seismic signal should be detectable. The detection limit also depends on the station signal quality; if the noise level is high, the signal may be obscured, even though a station farther away with a lower noise level will still record it clearly. Similar studies had been reported by Dammeier et al. (2011) and Chen et al. (2013).

31. **P11 Eq 5: to discuss validity and limits of EQ 5 it should be made clearer how (empirically?) and with which dataset/environment this relationship was obtained.**

R: The authors appreciate reviewer's recommendation. In the study, the lower boundary of detection was determined empirically based on two lowest values of the farthest distance of detection (i.g. 31.0 km and 37.6 km) having the disturbed areas of $1.6 \times 10^5$ and $1.2 \times 10^5$ m$^2$ . Dammeier et al. (2011) used a similar way to get their equation of the lower boundary of detection. The modification will be added to text.

32. **Fig 3: closest station is MASB (in the caption) or SGSB (in the map) ? It means 90% of the landslide and seismic signal**

**R:** We deeply thanks for careful inspection. The closest station should be SGSB. The mistake will be revised in the later version of manuscript.

**33. Fig 5: The last panel is not very clear: Cumulated rainfall is the total rainfall in the triggering storm. Antecedent rainfall has no reason to be compared directly with landslide occurrence, but only when summed with the cumulated rainfall. So why not show Rt the total effective rainfall together with Rc the cumulative rainfall (Given that Rt>= Rc it should be easy to visualize).**

**R:** Thanks for the constructive suggestion. The last panel in Fig. 5 has been revised to display Rt and Rc. The revised Fig. 5 is as below:

[Figure]

Fig. 5.

**34. Fig 6: Log Log scale is needed on all panel. Right now we do not see clearly the position of the different data points.**

**R:** Thanks for the constructive suggestion. All figures in Fig. 6 have been transferred to log-log scale as follows:

[Figure]

Fig. 6.

**35. Fig 7 and 8: I do not believe any of the subset can be significantly distinguished. What is driving the (small) difference in threshold curve is only 1 or 2 points out of each subset (that seems to be15-25 points). These low points shift the threshold while the bulk of each population do not seem different in any way. I am convinced this can only be due to chance and not to a shift of the whole population. I am even surprised that the curves are so low because if they are the 5% exceedance probability ~1 point should be left out in subset of ~20...**

**R:** We appreciate the valuable comments, and decided to remove this section. Meanwhile, we added a section to illustrate the validation of the rainfall thresholds mentioned in the study and other previous studies.

**36. Fig 9: I really tried, but did not understand it... I got that the line, is an empirical estimation of the distance at which station should be able to detect a landslide of a given size. What are the points? The 62 LSL? If yes, why are they all above the line? Does that mean only distant station detect the slides? I can believe for some but not the whole dataset, and this seems contradictory with Fig 3**

**R:** This detection limit line is estimated empirically based on distribution of data which represented the farthest distance that landslide signal was detected. This result indicate that the distance between a station and a landside below this line must be detected, but if it exceeds this line, it would probably be missed.

To determine the limits of large landslide detection distance as a function of event volume, we selected the farthest seismic station at which each event was detectable. An event was deemed detectable when we had selected the station for the distance-dependency analysis. The remaining results are shown on a plot of distance versus disturbed area (Fig. 9), where we can observe an upper detection limit described by equation 5. If, for a given event, a station plots in the lower right area below the dashed line (equation (5)), the seismic signal should be detectable. The detection limit also depends on the station signal quality; if the noise level is high, the signal may be obscured, even though a station farther away with a lower noise level will still record it clearly.

Totally 62 data points in the Fig.9. Each data point represents the distance between landslide location and the farthest detectable station as well as landslide-disturbed area. In the order words, it indicate that landslide signals can be detectable as the distance between landslide and seismic station shorter than the value of data. Therefore, to determine a lower boundary of these data can demarcate an effectively detectable region.

**37. References not used in the manuscript**
**-- Wilson and Wieczorek 1995, Rainfall thresholds for the initiation of debris flows at La Honda, California**
**-- Iverson, 2000, Landslide triggering by rain infiltration**
**-- Van asch et al., 1999, A view on some hydrological triggering systems in landslides**
**Larsen et al., 2010, Landslide erosion controlled by hillslope material**
**R:** Thanks for reviewing. The reference list has been overhauled completely before resubmitted.

Wilson, R. C., and Wieczorek, G. F. (1995). Rainfall thresholds for the initiation of debris flows at La Honda, California. Environmental & Engineering Geoscience, 1(1), 11-27.

Iverson, R. M. (2000). Landslide triggering by rain infiltration. Water resources research, 36(7), 1897-1910.

Van Asch, T. W., Buma, J., and Van Beek, L. P. H. (1999). A view on some hydrological triggering systems in landslides. Geomorphology, 30(1-2), 25-32.

Larsen, I. J., Montgomery, D. R., and Korup, O. (2010). Landslide erosion controlled by hillslope material. Nature Geoscience, 3(4), 247.

Table S1

| | Date and time (UTC) | Longitude | Latitude | Disturbed area (km$^2$) | Elev. of Landslide (m) | Rain Station | Distance (km) | Elev. of Rain station (m) | Reactive Landslide | Image Date |
|---|---|---|---|---|---|---|---|---|---|---|
| 1 | 2005/07/18 19:42 | 120.74 | 22.80 | 0.13 | 1388.6 | C1R120 | 5.3 | 820 | N | 2005/07/25 |
| 2 | 2005/07/20 18:15 | 120.75 | 22.74 | 0.13 | 813.7 | C1R130 | 0.9 | 1040 | N | 2005/07/25 |
| 3 | 2005/07/20 21:55 | 120.82 | 22.88 | 0.12 | 1535.0 | 01P260 | 10.8 | 458 | N | 2005/07/25 |
| 4 | 2005/07/21 06:33 | 120.72 | 22.85 | 0.18 | 950.1 | C0R100 | 3.9 | 1006 | R | 2006/07/29 |
| 5 | 2006/06/09 16:53 | 121.33 | 24.29 | 0.11 | 2304.6 | C1H860 | 24.1 | 1840 | R | 2006/07/19 |
| 6 | 2008/07/18 21:30 | 121.01 | 23.82 | 0.10 | 1093.8 | C1I040 | 1.6 | 1693 | R | 2008/11/20 |
| 7 | 2008/07/18 23:55 | 120.66 | 23.15 | 0.12 | 749.0 | C1V230 | 6.0 | 760 | N | 2008/11/25 |
| 8 | 2008/09/15 02:45 | 121.38 | 24.35 | 0.14 | 2236.4 | 41U090 | 5.7 | 1930 | R | 2008/12/03 |
| 9 | 2008/09/17 18:50 | 121.00 | 24.10 | 0.89 | 1104.3 | 01F100 | 2.6 | 1600 | N | 2008/11/15 |
| 10 | 2009/08/08 00:04 | 120.72 | 22.57 | 0.39 | 1207.1 | C1R240 | 10.9 | 74 | R | 2010/04/10 |
| 11 | 2009/08/08 00:35 | 120.73 | 22.49 | 0.12 | 950.2 | 01Q350 | 6.2 | 700 | N | 2010/04/10 |
| 12 | 2009/08/08 01:20 | 120.77 | 23.49 | 0.14 | 1411.2 | H1M240 | 5.2 | 1850 | N | 2010/02/23 |
| 13 | 2009/08/08 02:20 | 120.85 | 22.98 | 0.11 | 2167.4 | 01P260 | 15.2 | 458 | R | 2010/04/10 |
| 14 | 2009/08/08 03:55 | 120.75 | 23.08 | 0.33 | 923.5 | C1V270 | 5.4 | 1792 | R | 2010/02/23 |
| 15 | 2009/08/08 05:35 | 120.83 | 23.52 | 0.50 | 1903.4 | C0H9A0 | 2.4 | 1595 | N | 2010/02/23 |
| 16 | 2009/08/08 06:25 | 120.82 | 23.06 | 0.39 | 1726.3 | 01V040 | 20.8 | 265 | N | 2010/02/23 |
| 17 | 2009/08/08 06:28 | 120.67 | 23.01 | 0.15 | 517.2 | 01V040 | 4.0 | 265 | N | 2010/02/23 |
| 18 | 2009/08/08 07:15 | 120.70 | 23.01 | 0.23 | 903.9 | C1V300 | 1.5 | 1637 | N | 2010/02/23 |
| 19 | 2009/08/08 07:35 | 120.70 | 22.75 | 0.49 | 647.8 | C1R120 | 1.0 | 820 | R | 2010/02/23 |
| 20 | 2009/08/08 08:10 | 120.81 | 23.00 | 0.19 | 2007.5 | C1V300 | 10.1 | 1637 | R | 2010/02/23 |
| 21 | 2009/08/08 08:20 | 120.91 | 23.33 | 0.41 | 1703.8 | 01V070 | 6.3 | 2230 | N | 2010/01/11 |
| 22 | 2009/08/08 09:01 | 120.79 | 22.61 | 0.62 | 1222.0 | 01Q910 | 13.4 | 1158 | R | 2010/04/10 |

| 23 | 2009/08/08 10:40 | 120.86 | 22.80 | 0.16 | 1424.2 | 01Q910 | 12.8 | 1158 | R | 2010/04/10 |
|----|------------------|--------|-------|------|--------|--------|------|------|---|------------|
| 24 | 2009/08/08 11:35 | 120.95 | 23.33 | 0.22 | 2029.6 | C1V170 | 15.1 | 3340 | N | 2010/01/11 |
| 25 | 2009/08/08 13:56 | 120.66 | 22.96 | 0.11 | 407.3 | 01V040 | 5.0 | 265 | N | 2010/02/23 |
| 26 | 2009/08/08 16:15 | 120.88 | 23.18 | 0.14 | 2162.1 | C1V220 | 7.4 | 1781 | R | 2010/01/11 |
| 27 | 2009/08/08 17:05 | 120.71 | 22.49 | 0.94 | 1168.5 | C1R240 | 9.3 | 74 | N | 2010/04/10 |
| 28 | 2009/08/08 17:21 | 120.90 | 23.07 | 0.28 | 2606.3 | C1V270 | 9.8 | 1792 | R | 2010/04/10 |
| 29 | 2009/08/08 17:53 | 120.91 | 23.08 | 0.19 | 2459.5 | C1V270 | 10.7 | 1792 | R | 2010/04/10 |
| 30 | 2009/08/08 18:11 | 120.79 | 23.51 | 1.12 | 1763.8 | 467530 | 2.8 | 2413.4 | N | 2010/02/23 |
| 31 | 2009/08/08 18:16 | 120.83 | 22.63 | 0.72 | 1055.7 | 01Q910 | 13.5 | 1158 | R | 2010/04/10 |
| 32 | 2009/08/08 18:19 | 120.72 | 22.70 | 0.56 | 603.4 | 01Q910 | 5.3 | 1158 | R | 2010/03/06 |
| 33 | 2009/08/08 18:28 | 120.66 | 22.95 | 0.12 | 554.4 | 01V040 | 5.8 | 265 | R | 2010/02/23 |
| 34 | 2009/08/08 19:19 | 120.71 | 22.67 | 0.64 | 705.7 | 01Q910 | 7.6 | 1158 | R | 2010/03/06 |
| 35 | 2009/08/08 20:15 | 120.73 | 22.59 | 0.73 | 1509.7 | C1R240 | 12.8 | 74 | N | 2010/04/10 |
| 36 | 2009/08/08 20:27 | 120.92 | 23.40 | 0.12 | 2278.6 | C1V460 | 4.5 | 1949 | N | 2010/01/11 |
| 37 | 2009/08/08 21:11 | 120.90 | 23.46 | 0.15 | 1904.4 | C1V460 | 2.3 | 1949 | R | 2010/02/23 |
| 38 | 2009/08/08 21:30 | 120.92 | 23.49 | 0.12 | 2450.4 | C1V460 | 6.4 | 1949 | N | 2010/02/23 |
| 39 | 2009/08/08 21:42 | 120.91 | 23.10 | 0.25 | 2274.1 | C1V460 | 37.3 | 1949 | R | 2010/04/10 |
| 40 | 2009/08/08 22:16 | 120.66 | 23.17 | 2.50 | 681.3 | C1R880 | 6.4 | 223 | R | 2010/02/23 |
| 41 | 2009/08/08 22:52 | 120.90 | 23.54 | 0.12 | 1936.8 | C1I340 | 4.5 | 897 | R | 2010/02/23 |
| 42 | 2009/08/08 23:02 | 120.60 | 23.03 | 0.13 | 747.9 | C0V250 | 5.2 | 298 | N | 2010/04/10 |
| 43 | 2009/08/08 23:14 | 120.75 | 23.29 | 0.56 | 1525.1 | C1V200 | 7.7 | 860 | R | 2010/02/23 |
| 44 | 2009/08/08 23:15 | 120.77 | 22.63 | 0.15 | 2309.0 | 01Q250 | 8.2 | 950 | R | 2010/04/10 |
| 45 | 2009/08/08 23:41 | 120.84 | 22.63 | 0.12 | 825.0 | 01Q910 | 13.9 | 1158 | R | 2010/04/10 |
| 46 | 2009/08/09 00:34 | 120.77 | 23.22 | 2.24 | 1352.5 | C1V210 | 4.0 | 700 | R | 2010/02/23 |
| 47 | 2009/08/09 02:52 | 120.77 | 23.23 | 0.81 | 1559.3 | C1V210 | 4.1 | 700 | R | 2010/02/23 |
| 48 | 2009/08/09 03:55 | 120.72 | 22.60 | 0.63 | 923.5 | 01Q250 | 2.5 | 950 | N | 2010/04/10 |
| 49 | 2009/08/09 09:37 | 120.81 | 22.56 | 2.31 | 1144.3 | 01Q350 | 14.3 | 250 | N | 2010/04/10 |

| 50 | 2009/08/09 11:00 | 120.77 | 22.82 | 0.13 | 1669.4 | C1R120 | 9.0 | 820 | R | 2010/04/10 |
| 51 | 2009/08/10 03:54 | 120.80 | 23.25 | 0.20 | 1227.6 | C1V210 | 2.8 | 700 | R | 2010/02/23 |
| 52 | 2009/08/10 04:22 | 120.76 | 23.31 | 1.52 | 1387.2 | C1V160 | 6.3 | 1040 | R | 2010/02/23 |
| 53 | 2010/09/19 23:24 | 120.73 | 22.85 | 0.15 | 1135.0 | 01Q910 | 13.9 | 1158 | R | 2011/04/16 |
| 54 | 2011/08/30 07:10 | 120.93 | 22.86 | 0.12 | 849.8 | 01Q350 | 44.9 | 1275 | R | 2012/02/27 |
| 55 | 2011/08/30 09:13 | 121.18 | 23.69 | 0.11 | 1811.5 | C1T940 | 19.6 | 1570 | R | 2012/02/27 |
| 56 | 2011/08/31 09:37 | 120.98 | 23.33 | 0.11 | 2714.0 | 01V070 | 8.5 | 2230 | R | 2012/02/27 |
| 57 | 2012/08/01 18:40 | 121.42 | 24.58 | 0.12 | 1512.0 | 01U050 | 8.1 | 400 | R | 2013/07/11 |
| 58 | 2012/08/02 10:00 | 121.85 | 24.52 | 0.12 | 83.3 | C0U710 | 33.3 | 1810 | N | 2013/06/28 |
| 59 | 2012/08/02 19:00 | 120.95 | 23.74 | 0.25 | 1677.9 | C1I310 | 6.6 | 1001 | N | 2013/06/03 |
| 60 | 2012/08/03 01:02 | 121.38 | 24.36 | 0.19 | 2356.6 | 41U090 | 4.7 | 1930 | N | 2013/07/11 |
| 61 | 2013/07/13 14:27 | 120.89 | 23.02 | 0.40 | 2604.8 | C1V270 | 10.1 | 1792 | R | 2014/07/13 |
| 62 | 2013/08/22 19:05 | 121.07 | 23.38 | 0.18 | 2114.6 | C1I140 | 41.2 | 1700 | R | 2014/07/13 |

---

## Author Comment (AC2) · 24 Aug 2018

RC #2: General comments

The paper "Evaluating critical rainfall conditions for large-scale landslides by detecting event times from seismic records" is a very interesting paper with original approach. The combination of the tools and methods to define rainfall threshold to landsliding is interesting and the several steps of the analysis are presented. However, the reader can be lost in the used databases, in particular between what concerns the 2009 typhoon analysis and the rest of the chronical. The results can be discussed (detection of only 62 landslides, thresholds between 500/300mm. . .), or justified by figures completed (see below comments on the figures).

R: The authors appreciate the constructive feedback of the reviewer – it has certainly helped the authors improve this manuscript. The reply is summarized as below:

1) Some confusing statements (e.g. landslide number, topic event, study period, etc.) will be modified in the revised manuscript.

2) The authors will provide and modify the description of data sources, quality, and accuracy (including rainfall information, satellite image, and seismic records).

3) More in-deep discussion on results will be added in the modified version.

4) The suggested modification of methods and figures will be done in the manuscript.

*Figure and Line by line reply have been provided with the supplementary file. Please see the attached material.

Please also note the supplement to this comment: https://www.nat-hazards-earth-syst-sci-discuss.net/nhess-2018-126/nhess-2018-126-AC2-supplement.pdf

—————————————————

[Figure]

**Supplement:**

RC #2: **General comments**

**The paper "Evaluating critical rainfall conditions for large-scale landslides by detecting event times from seismic records" is a very interesting paper with original approach. The combination of the tools and methods to define rainfall threshold to landsliding is interesting and the several steps of the analysis are presented. However, the reader can be lost in the used databases, in particular between what concerns the 2009 typhoon analysis and the rest of the chronical. The results can be discussed (detection of only 62 landslides, thresholds between 500/300mm…), or justified by figures completed (see below comments on the figures).**

R: The authors appreciate the constructive feedback of the reviewer – it has certainly helped the authors improve this manuscript. The reply is summarized as below:

1) Some confusing statements (e.g. landslide number, topic event, study period, etc.) will be modified in the revised manuscript.

2) The authors will provide and modify the description of data sources, quality, and accuracy (including rainfall information, satellite image, and seismic records).

3) More in-deep discussion on results will be added in the modified version.

4) The suggested modification of methods and figures will be done in the manuscript.

**Specific comments:**

1. **P2 L21: the event of 2009 is the only one mentioned, for the moment we can think that the research only focus on this event.**

R: The authors appreciate the reminding. We agree that the current manuscript may confuse readers due to many examples belonging to Typhoon Morakot. In the study, totally nineteen rainstorm events (seventeen typhoon-induced events and two heavy rainfall events) in the period of 2005-2014 were selected to examine the seismic records, but not only one event. The modified manuscript will clearly descript the targets in the section of introduction and study materials. The list of selected typhoons and heavy rainfall events will be added to the modified version (Table S2).

In the original manuscript, Typhoon Morakot was mentioned many times because it was one of the most tragic event in Taiwan in the past 20 years (more than 20,000 landslides, and more than four hundred large landslides with the disturbed area larger than 0.1 $km^2$), and therefore many good examples can be shown. In the modified version, the examples of other events will be added.

Table S2. List of selected nineteen rainstorm events (seventeen typhoons and two heavy rainfall events) to examine seismic records for identifying landslide-induced signals

|   | Event | Date (year/month/date) |
|---|-------|------------------------|
| 1 | Haitang | 2005/07/16-07/20 |
| 2 | Talim | 2005/08/30-09/01 |
| 3 | 0609 Rain | 2006/06/09 |
| 4 | Bilis | 2006/07/12-07/15 |
| 5 | 0604 Rain | 2007/06/04 |
| 6 | Kalmaegi | 2008/07/16-07/18 |
| 7 | Fung-Wong | 2008/07/26-07/29 |
| 8 | Sinlaku | 2008/09/11-09/16 |
| 9 | Morakot | 2009/08/05-08/10 |
| 10 | Fanapi | 2010/09/17-09/20 |
| 11 | Megi | 2010/10/21-10/23 |
| 12 | Nanmadol | 2011/08/27-08/31 |
| 13 | Talim | 2012/06/19-06/21 |
| 14 | Saola | 2012/07/31-08/03 |
| 15 | Tembin | 2012/08/21-08/25 |
| 16 | Soulik | 2013/07/11-07/13 |
| 17 | Trami | 2013/08/20-08/22 |
| 18 | Matmo | 2014/07/21-07/23 |
| 19 | Fung-Wong | 2014/09/19-09/22 |

**2. P3 L8: Date of the images? Number? Mapping only for the 2009 event.**

R: Thanks for comments. The date and number of used SPOT-4 satellite images will be listed in Table S1. Landslide mapping was conducted for nineteen rainstorm events (seventeen typhoon-induced events and two heavy rainfall events).

**3. P3 L23: Why 0.1km² Is it the limit of the automatic detection based on SPOT images? How many landslides were detected?**

R: Based on the definition and characteristic of deep-seated gravitational slope deformation (DSGSD) and description of large-scale landslides (Lin et al., 2013; Lin et al., 2013), a large-scale landslide should have three characteristics, including 1) a depth larger than 10 m, 2) a volume greater than 1000,000 $m^3$, and 3) a speedy movement velocity. In practice, it is difficult to get these three characteristics without in-situ investigation and geodetic survey. Therefore, we chose the disturbed area of 100,000 $m^2$ (0.1 $km^2$, volume/depth) as the indicator to sort large-scale landslides from other types of slope failure. Landslide interpretation with satellite imagery is the fastest way to classify large-scale landslides. Actually, more than three hundred seismic signals

having the seismic characteristics of landslide-induced ground motions, however, only 62 signals having clear landslide-signal signatures were detected by at least three seismic stations. Therefore, we just could locate the possible locations of these 62 signals and paired the locating points with landslides. Although the successful detection and locating rate may be less than 20 % in the period of 2005-2014, we believe that the 62 landslides still provide many valuable time information.

Reference:

Lin, C. W., Tseng, C. M., Tseng, Y. H., Fei, L. Y., Hsieh, Y. C., and Tarolli, P. (2013). Recognition of large scale deep-seated landslides in forest areas of Taiwan using high resolution topography. Journal of Asian Earth Sciences, 62, 389-400.

Lin, M. L., Chen, T. W., Lin, C. W., Ho, D. J., Cheng, K. P., Yin, H. Y., and Chen, M. C. (2013). Detecting large-scale landslides using LiDar data and aerial photos in the Namasha-Liuoguey area, Taiwan. Remote Sensing, 6(1), 42-63.

**4. P3 L26: How we consider the progressive instability and the signal before the main failure?**

R: The authors agree that investigation of progressive instability is quite important. We believe that even slight displacement of materials on slope can stir energy transfer and induce seismic signals. However, the seismic signals generated by the processes of progressive deformation/displacement of material on slope do not contain enough energy to be recorded by remote seismic stations. Therefore, we did not try to monitor creeping processes by seismicity-approaches.

**5. P4 L3: Now we don't care about 2009 event. Why 2005-2014? What was the aim of 2009?**

R: Due to a large number of large landslides in 2009, the seismicity method for landslide-generated signals was successfully used. The study attempt to use the seismicity method for a longer period. Besides, the quality of seismic records of Taiwan's broadband seismic network had been significantly enhanced after 2005. The study period was decided to begin in 2005. In addition, identification of landslide-induced signals was conducted manually in the study. Therefore, identification cost a lot of time, and so far we finished the identification until 2014. So the study period during 2005-2004 was determined. It can be expected that more landslide signals will be found in the future.

**6. P4 L11: "only events with obvious signature", do you mean the 62 landslides in the fig.1? can you develop the characteristics of the signal that you can**

**highlight with these 62 events?**

R: The sentence maybe not clear. The revision is as below:

"To reduce the uncertainty caused by the manual identification, the events with very obvious triangle-shape signatures in the spectrograms were used to examine rainfall statistics in this study."

The in-deep description of the characteristics of landslide-induced seismic signals will be added to text as follows:

"…The seismic wave generated by landslide can be attributed to the shear force and loading on the ground surface as the mass moving downslope. Many studies have shown that the source mechanism of a landslide is highly complicated, and their seismic wave mainly consist of surface wave and shear wave, making it difficult to distinguish P wave and S wave from station records (Lin et al., 2010; Suwa et al., 2010; Dammeier et al., 2011; Feng, 2011; Hibert et al., 2014). The onset of landslide seismic signal is generally emergent. Then, the seismic amplitude increases gradually above ambient noise level to peak ground motion, exhibiting a 'cigar' shape envelope. After the peak amplitude, most of landslide-generated seismic signals have relatively long decay time, on average about 70% of total signal duration (Norris, 1994; La Rocca et al., 2004; Surinach et al., 2005; Deparis et al., 2008; Schneider et al., 2010; Dammeier et al., 2011; Allstadt, 2013). In frequency domain, landslide-induced seismic energy was mainly distributed below 10 Hz, with a triangular shaped signature in spectrogram, due to an increase over time in high-frequency constituents (Surinach et al., 2005; Dammeier et al., 2011)."

Reference:

Allstadt, K. (2013). Extracting source characteristics and dynamics of the August 2010 Mount Meager landslide from broadband seismograms. Journal of Geophysical Research: Earth Surface, 118(3), 1472-1490. doi:10.1002/jgrf.20110.

Dammeier, F., Moore, J. R., Haslinger, F., and Loew, S. (2011). Characterization of alpine rockslides using statistical analysis of seismic signals. Journal of Geophysical Research, 116(F4). doi:10.1029/2011jf002037

Deparis, J., Jongmans, D., Cotton, F., Baillet, L., Thouvenot, F., and Hantz, D. (2008). Analysis of rock-fall and rock-fall avalanche seismograms in the French Alps. Bulletin of the Seismological Society of America, 98(4), 1781-1796.

Doi:10.1785/0120070082.

Feng, Z. (2011). The seismic signatures of the 2009 Shiaolin landslide in Taiwan. Natural Hazards and Earth System Science, 11(5), 1559-1569. Doi:10.5194/nhess-11-1559-2011

Hibert, C., Ekström, G., and Stark, C. P. (2014). Dynamics of the Bingham Canyon Mine landslides from seismic signal analysis. Geophysical Research Letters, 41(13), 4535-4541. Doi:10.1002/2014gl060592

La Rocca, M., Galluzzo, D., Saccorotti, G., Tinti, S., Cimini, G. B., and Del Pezzo, E. (2004). Seismic signals associated with landslides and with a tsunami at Stromboli volcano, Italy. Bulletin of the Seismological Society of America, 94(5), 1850-1867. Doi:10.1785/012003238.

Lin, C. H., Kumagai, H., Ando, M., and Shin, T. C. (2010). Detection of landslides and submarine slumps using broadband seismic networks. Geophysical Research Letters, 37(22), n/a-n/a. doi:10.1029/2010gl044685

Norris, R. D. (1994). Seismicity of rockfalls and avalanches at 3 Cascade Range volcanos – Implications for seismic detection of hazardous mass movements. Bulletin of the Seismological Society of America, 84(6), 1925-1939.

Schneider, D., Bartelt, P., Caplan-Auerbach, J., Christen, M., Huggel, C., and McArdell, B. W. (2010). Insights into rock-ice avalanche dynamics by combined analysis of seismic recordings and a numerical avalanche model. Journal of Geophysical Research, 115(F4). Doi:10.1029/2010jf001734.

Surinach, E., Vilajosana, I., Khazaradze, G., Biescas, B., Furdada, G., and Vilaplana, J. M. (2005). Seismic detection and characterization of landslides and other mass movements. Natural Hazards and Earth System Sciences, 5, 791-798.

Suwa, H., Mizuno, T., and Ishii, T. (2010). Prediction of a landslide and analysis of slide motion with reference to the 2004 Ohto slide in Nara, Japan. Geomorphology, 124(3-4), 157-163. Doi:10.1016/j.geomorph.2010.05.003.

**7. P4 L19: how can you consider the lag time between rainfall / soil saturation… and failure?**

R: Thanks for the valuable comments. This study used seismic signals to find out landslide occurrence time, and undoubtedly this time information could help to study the infiltration of rainfall or the relationship between soil saturation and landslides. However, this study currently focused on constructing and compare the rainfall thresholds for landslide warning, the traditionally statistical methods to estimate rainfall threshold were chosen. Time lag between rainfall history and soil saturation process was not considered in the study.

**8. P4 L29: I think the chosen method can be shortly developed here.**

R: Thanks for t your suggestion. The detailed content of locating method will be added to the text as follows:

> "…. Locations are estimated with a cross-correlation method that could maximizes tremor signal coherency among seismic stations. The criteria of station chosen was based on geographic distribution and tremor signal-to-noise ratios. The interpreted signals would be created envelope function to process cross-correlation analyzed from different station pairs. We obtain centroid location estimates by cross-correlating all station pairs and performing the Monte Carlo grids searching method (Wech and Creager, 2008). While traditional methods seek the source location that minimizes the horizontal time difference between predicted travel time and peak lag time, the method seek to minimize the vertical correlation distance between the peak correlation value and the predicted correlation value."

Reference:

Wech, A. G., and Creager, K. C. (2008). Automated detection and location of Cascadia tremor. Geophysical Research Letters, 35(20).

**9. P4 L35: there is only Xiaolin landslide in this figure. So you focus on 2009 events? do you compared all the landslides detected by remote sensing (fig.1 is it landslides detected by seismic signal of remote sensing: to clarify) (how many by remote sensing?) with the seismic signal? an example of signal related to a smaller event than Xiaolin would be interesting (fig.3). Do the SSLs have a significant signal also?**

R: Thanks for the comments. In the figure 2c, six matched landslides (including Xiaolin) were visible. Each orange circle means a successfully-paired landslide. The outline of each paired landslide will be added to the figure. Besides, the signal and spectrogram of a 2005 large landslide smaller than Xiaolin will be provided and added to modified manuscript as below:

Theoretically, the seismic signal induced by a small landslide could be found if there is a seismic station very close to it. In practice, the seismic signals generated by small landslides do not contain enough energy to be recorded by remote seismic stations. Therefore, we do not try to examine small landslides by seismicity approaches.

[Figure]

10. **P5 L3: Now you are studying events between 2005-2014? It is a little bit confusing. How many typhoon events? don't you consider previous smaller rainfall events that could affect the mechanical properties of the slopes?**

R: In the study, totally seventeen typhoon events and two heavy rainfall events occurring in the period of 2005-2014 were chosen to examine the seismic records and identify landslide-induced signals. The list of selected events will be provided in Table S2.

In the study, seven-days antecedent rainfall was considered as a rainfall parameter. The effect of antecedent rainfall should decay with time. Therefore, decay rate of antecedent rainfall with day was used to estimate effective rainfall. According to the decay rate, 0.7, the effect of antecedent rainfall with counting days longer than 7 days is slight. In the study, we adopt seven-days antecedent rainfall to estimate effective rainfall.

Table S2

|   | Event | Date (year/month/date) |
|---|-------|------------------------|
| 1 | Haitang | 2005/07/16-07/20 |
| 2 | Talim | 2005/08/30-09/01 |
| 3 | 0609 Rain | 2005/06/09 |
| 4 | Bilis | 2005/07/12-07/15 |

| 5 | 0604 Rain | 2006/06/04 |
| 6 | Kalmaegi | 2006/07/16-07/18 |
| 7 | Fung-Wong | 2008/07/26-07/29 |
| 8 | Sinlaku | 2008/09/11-09/16 |
| 9 | Morakot | 2009/08/05-08/10 |
| 10 | Fanapi | 2010/09/17-09/20 |
| 11 | Megi | 2010/10/21-10/23 |
| 12 | Nanmadol | 2011/08/27-08/31 |
| 13 | Talim | 2012/06/19-06/21 |
| 14 | Saola | 2012/07/31-08/03 |
| 15 | Tembin | 2012/08/21-08/25 |
| 16 | Soulik | 2013/07/11-07/13 |
| 17 | Trami | 2013/08/20-08/22 |
| 18 | Matmo | 2014/07/21-07/23 |
| 19 | Fung-Wong | 2014/09/19-09/22 |

**11. P5 L6: How can you consider the topographic, orographic effects?**

R: We appreciate the comments. We tested the rainfall data used in the study to validate the influence of distance and topographic effect on rainfall distribution. The effect of rain gauge distribution over the accuracy of rainfall has been assessed using gauge observation in a 35 km × 50 km region of south Taiwan (Fig. S1). The amounts of daily rainfall during 2009 Typhoon Morakot (8/6-8/11) recorded at 19 rain gauge stations were selected to validate the accuracy of rainfall. The influence of topography on rainfall variability has been analyzed in the same 35 km × 50 km region of south Taiwan. The highest station elevation is 1792 m a.s.l. at C1V270, and the lowest station elevation is 105 m a.s.l. at C10830. The standard deviation of station elevation is 561 m. The values of standard deviation of daily rainfall at the 19 stations were calculated, and less than 13% except a high standard deviation, 45%, on sixth August (average daily rainfall less than 2 mm). The results demonstrated that high and even extreme rainfall are less influenced by elevation, while low and medium rainfall events are significantly influenced by elevation variation, with most of the rainfall appearing on high elevations. Similar results have also been reported by some previous studies (Sanchez-Moreno et al., 2014; Ge et al., 2017). Because the study only considered the rainfall events with total cumulated rainfall greater than 500 m, the elevation effect was ignored as selecting rain station.

[Figure]

Fig. S1. The distribution of rain gauge stations and the location of the central point of the testing area for validating the influence of the distance between rain gauge and a given point.

Reference:

Sanchez-Moreno, J.F., Mannaerts, C.M., and Jetten, V. (2014) Influence of topography on rainfall variability in Santiago Island, Cape Verde. International Journal of Climatology, 34, 1081-1097.

Ge, G., Shi, Z., Yang, X., Hao, Y., Guo, H., Kossi, F., Xin, Z., Wei, W., Zhang, Z., Zhang, X., Liu, Y., and Liu, J. (2017) Analysis of Precipitation Extremes in the

Qinghai-Tibetan Plateau, China: Spatio-Temporal Characteristics and Topography Effects. Atmosphere, 8(7), 127, doi:10.3390/atmos8070127.

**12. P5 L10: 100km² is already large catchment.**

R: The effect of station distance has been tested to variation of rainfall. The errors of daily rainfall between the central point and the nearest rain gauge station (01V040) were smaller than 10 % (0.5%-10% at different date). Besides, the correlation coefficients would keep at 90% as a distance between the central point and rain gauge stations less than 20 km, and even keep at 98% as a distance less than 10 km (Fig. S2). Therefore, in the study, an upper limit of basin area smaller than 100 km² (10 km × 10 km was adopted to avoid a significant decrease of the accuracy of rainfall. Because the density of rainfall stations in mountainous area would significantly decreases, the number of usable rainfall stations may be limited. The size of catchment area of 100km² is the upper limit for choose rainfall station. In practice, we chose the closest rainfall station.

[Figure]

Fig. S2. Variation of correlation coefficient

**13. P5 L2: rain event = typhoon?**

R: Yes. The sentence will be revised.

**14. P6 L4: 193 small landslides for which period?**

R: The 193 small landslides were investigated by SWBC during 2006-2013. The illustration on the small landslides will be improved in the later version of manuscript.

**15. P6 L15: EQ1 cohesion here is only considered for a discontinuity (C = 0)? Or for the specific material?**

R: We thanks reviewer's recommendation. Well development of detachment plane (e.g., sliding surface between sedimentary layers, connected joints, weathered foliation, etc.) have been widely considered as one important geological condition to induce a large landslide. Therefore, in the study, the C' of the detachment plane is simply assumed as the value of zero to behave the critical situation of slope stability. Cohesion in equation (1) is not considered for a specific material. The illustration of C' will be modified in the text.

> "…Well development of detachment plane (e.g., sliding surface between sedimentary layers, connected joints, weathered foliation, etc.) have been widely considered as the geological conditions to occur a large landslide (Agliardi et al., 2001; Tsou et al., 2011)."

Reference:
Tsou, C. Y., Feng, Z. Y., & Chigira, M. (2011). Catastrophic landslide induced by typhoon Morakot, Shiaolin, Taiwan. Geomorphology, 127(3-4), 166-178.
Agliardi, F., Crosta, G., & Zanchi, A. (2001). Structural constraints on deep-seated slope deformation kinematics. Engineering Geology, 59(1-2), 83-102.

**16. P7 L9/22: lower slopes VS steeper slopes… upslope VS downslope? Can you explain it? Is it a regressive erosion of the slope?**

R: Thanks for the important reviewing. The sentence should be revised as below:

> "…due to the fact that during the extremely heavy rainfall of Typhoon Morakot in 2009, more than 2,000 mm precipitated in four days, causing a large number of landslides, and exhausting a lot of unstable slopes. Consequently, landslides transferred to occur on steeper slopes in the following years."

**17. P7 L9/24 & 26: landslides for 2009 event? Or the detection of 62 landslides grounded on seismic signal among 686 inventoried landslides? What is the landslide seismic magnitude?**

**R:** In this section, the topographic analysis was for 686 large landslides during 2005-2014. The paragraph will be modified to avoid confusion. We did not calculate landslide

seismic magnitude in the study due to the lack of a standard method for estimating landslide seismic magnitude so far.

**18. P8 L4: what about SSL?**

R: Most of the small landslides have strong instantaneous rainfall intensity. This means that a short duration and heavy rainfall can easily trigger small landslides.

**Discussion:**

**The discussion is interesting because it puts the results in perspective. Nevertheless, some points have to be clarify.**

**19. 5.1. The authors highlight the fact that critical rainfall to trigger landslides has decreased since 2010 (500mm to 300mm) according to the results fig. 7. How many events the threshold is based on? The figure 7 is not so evident.**

**To explain these results, the authors question the Morakot typhoon. Was it an exceptional hydro-climatic event? The other solution is that instabilities induced by the 2009 typhoon are responsible of recent landslides. This idea should be developed here, and maybe associated to a map of the landslides scars (delineation of the departure areas) and differentiated according to the periods of the triggering…**

R: Thanks for comments. The authors agree the conceptual view of reviewer. The discussion needs more solid information and field investigation. Therefore, the part about the influence of rock types and an extreme event will be removed and replaced with in-deep comparison of different rainfall threshold.

**20. 5.2. The authors mention the fact that landslides occurred several types of rocks with different geotechnical behaviors, but the chosen geotechnical parameters (table 1) are identical. Why?**

R: The main research purpose of this study was to establish a rainfall warning threshold which is applicable for large landslides, so a relatively simple but effective method was adopted. In this method, Keefer (1987) assumes that there is a potential sliding surface for these landslides, and the depth of the large-scale landslides are often deep to the strata. Therefore, although the movements of the soil material are not completely the same, under this assumption, it can still reach a considerable good effect.

In order to improve the Qc threshold, the critical volume of water, $Q_c$, for each large landslide was estimated based on its slope gradient and depth (estimated by the equation: $Z = 26.14A^{0.4}$; Z: depth, m; A: disturbed area, $m^2$). Following the equation (4), the drainage rate, $I_0$, for each landslide can be calculated. For the 62 detected landslide, the cumulative probability of 5% of $Q_c$ and $I_0$ values was taken as the critical values in the

mixing physically- and statistically-based threshold. The critical value of $I_0$ was 1.5, and the critical Qc was 430.2. The paragraph will be modified as follows:

> "The critical volume of water, $Q_C$, on sliding surface for each large landslide was estimated based on its slope gradient, depth (estimated by the equation: $Z = 26.14A^{0.4}$; Z: depth in m; A: disturbed area in m$^2$), and the geological material parameters of the study area (Table 1). The $Q_C$ value was inserted into $Q_C = (I - I_0) \cdot D$ to obtain an $I_0$ value for each large landslide. For the 62 detected landslide, the cumulative probability of 5% of Qc and $I_0$ values was taken as the critical values. The critical value of $I_0$ was 1.5, and the critical $Q_C$ was 430.2, which is more suitable for LSLs than for SSLs, and the threshold curve was rewritten as $(I - 1.5) \cdot D = 430.2$."

**21. Effective rainfall, and rainfall duration thresholds according to the rock types are not clear in the figure 8, could another statistical analysis put the conclusion of the authors in obvious fact?**

R: Thank you for your suggestion. The figure will be removed.

**Figure:**

**22. Fig. 2. Add legend for the detected landslide: Is it detected by seismic signal analysis? Is the point, the centroid of the landslide? Why not the delineation of the landslide body? Dates of the both satellite images here.**

R: Thank you for your suggestion. The figure will be modified based on suggestions.

**23. Fig. 3. Location of the detected landslides in 2009? Is there other spectrogram for previous landslides? Or after 2009? or associated to another landslide triggered in 2009: X spectrogram for 1 landslide. The star is the location defined with which seismic station?**

R: This events, the Xioulin landslide, is one of the most tragic event during Typhoon Morakot, 2009. The other 61 detected landslides also have the triangle-shape characteristic patterns in their spectrograms. The star is the location defined with all the stations which could detect the signals from the Xioulin landslide. We will provide two examples of spectrograms of one landslide in 2005 and one landslide after 2009 in supplementary materials.

[Figure]

[Figure]

**24. Fig. 4. Maybe with the topography visible on the map?**

R: Thank you for your suggestion. We had tried added topography in the fig. 4a. However, so many information made visually chaotic.

**25. Fig. 7. A) Is there only 1 event for the lowest limit?**

R: Yes. The figure will be removed in modified version.

Table S1

| | Date and time (UTC) | Longitude | Latitude | Disturbed area (km$^2$) | Elev. of Landslide (m) | Rain Station | Distance (km) | Elev. of Rain station (m) | Reactive Landslide | Image Date |
|---|---|---|---|---|---|---|---|---|---|---|
| 1 | 2005/07/18 19:42 | 120.74 | 22.80 | 0.13 | 1388.6 | C1R120 | 5.3 | 820 | N | 2005/07/25 |
| 2 | 2005/07/20 18:15 | 120.75 | 22.74 | 0.13 | 813.7 | C1R130 | 0.9 | 1040 | N | 2005/07/25 |
| 3 | 2005/07/20 21:55 | 120.82 | 22.88 | 0.12 | 1535.0 | 01P260 | 10.8 | 458 | N | 2005/07/25 |
| 4 | 2005/07/21 06:33 | 120.72 | 22.85 | 0.18 | 950.1 | C0R100 | 3.9 | 1006 | R | 2006/07/29 |
| 5 | 2006/06/09 16:53 | 121.33 | 24.29 | 0.11 | 2304.6 | C1H860 | 24.1 | 1840 | R | 2006/07/19 |
| 6 | 2008/07/18 21:30 | 121.01 | 23.82 | 0.10 | 1093.8 | C1I040 | 1.6 | 1693 | R | 2008/11/20 |
| 7 | 2008/07/18 23:55 | 120.66 | 23.15 | 0.12 | 749.0 | C1V230 | 6.0 | 760 | N | 2008/11/25 |
| 8 | 2008/09/15 02:45 | 121.38 | 24.35 | 0.14 | 2236.4 | 41U090 | 5.7 | 1930 | R | 2008/12/03 |
| 9 | 2008/09/17 18:50 | 121.00 | 24.10 | 0.89 | 1104.3 | 01F100 | 2.6 | 1600 | N | 2008/11/15 |
| 10 | 2009/08/08 00:04 | 120.72 | 22.57 | 0.39 | 1207.1 | C1R240 | 10.9 | 74 | R | 2010/04/10 |
| 11 | 2009/08/08 00:35 | 120.73 | 22.49 | 0.12 | 950.2 | 01Q350 | 6.2 | 700 | N | 2010/04/10 |
| 12 | 2009/08/08 01:20 | 120.77 | 23.49 | 0.14 | 1411.2 | H1M240 | 5.2 | 1850 | N | 2010/02/23 |
| 13 | 2009/08/08 02:20 | 120.85 | 22.98 | 0.11 | 2167.4 | 01P260 | 15.2 | 458 | R | 2010/04/10 |
| 14 | 2009/08/08 03:55 | 120.75 | 23.08 | 0.33 | 923.5 | C1V270 | 5.4 | 1792 | R | 2010/02/23 |
| 15 | 2009/08/08 05:35 | 120.83 | 23.52 | 0.50 | 1903.4 | C0H9A0 | 2.4 | 1595 | N | 2010/02/23 |
| 16 | 2009/08/08 06:25 | 120.82 | 23.06 | 0.39 | 1726.3 | 01V040 | 20.8 | 265 | N | 2010/02/23 |
| 17 | 2009/08/08 06:28 | 120.67 | 23.01 | 0.15 | 517.2 | 01V040 | 4.0 | 265 | N | 2010/02/23 |
| 18 | 2009/08/08 07:15 | 120.70 | 23.01 | 0.23 | 903.9 | C1V300 | 1.5 | 1637 | N | 2010/02/23 |
| 19 | 2009/08/08 07:35 | 120.70 | 22.75 | 0.49 | 647.8 | C1R120 | 1.0 | 820 | R | 2010/02/23 |
| 20 | 2009/08/08 08:10 | 120.81 | 23.00 | 0.19 | 2007.5 | C1V300 | 10.1 | 1637 | R | 2010/02/23 |
| 21 | 2009/08/08 08:20 | 120.91 | 23.33 | 0.41 | 1703.8 | 01V070 | 6.3 | 2230 | N | 2010/01/11 |
| 22 | 2009/08/08 09:01 | 120.79 | 22.61 | 0.62 | 1222.0 | 01Q910 | 13.4 | 1158 | R | 2010/04/10 |

| 23 | 2009/08/08 10:40 | 120.86 | 22.80 | 0.16 | 1424.2 | 01Q910 | 12.8 | 1158 | R | 2010/04/10 |
|----|------------------|--------|-------|------|--------|--------|------|------|---|------------|
| 24 | 2009/08/08 11:35 | 120.95 | 23.33 | 0.22 | 2029.6 | C1V170 | 15.1 | 3340 | N | 2010/01/11 |
| 25 | 2009/08/08 13:56 | 120.66 | 22.96 | 0.11 | 407.3 | 01V040 | 5.0 | 265 | N | 2010/02/23 |
| 26 | 2009/08/08 16:15 | 120.88 | 23.18 | 0.14 | 2162.1 | C1V220 | 7.4 | 1781 | R | 2010/01/11 |
| 27 | 2009/08/08 17:05 | 120.71 | 22.49 | 0.94 | 1168.5 | C1R240 | 9.3 | 74 | N | 2010/04/10 |
| 28 | 2009/08/08 17:21 | 120.90 | 23.07 | 0.28 | 2606.3 | C1V270 | 9.8 | 1792 | R | 2010/04/10 |
| 29 | 2009/08/08 17:53 | 120.91 | 23.08 | 0.19 | 2459.5 | C1V270 | 10.7 | 1792 | R | 2010/04/10 |
| 30 | 2009/08/08 18:11 | 120.79 | 23.51 | 1.12 | 1763.8 | 467530 | 2.8 | 2413.4 | N | 2010/02/23 |
| 31 | 2009/08/08 18:16 | 120.83 | 22.63 | 0.72 | 1055.7 | 01Q910 | 13.5 | 1158 | R | 2010/04/10 |
| 32 | 2009/08/08 18:19 | 120.72 | 22.70 | 0.56 | 603.4 | 01Q910 | 5.3 | 1158 | R | 2010/03/06 |
| 33 | 2009/08/08 18:28 | 120.66 | 22.95 | 0.12 | 554.4 | 01V040 | 5.8 | 265 | R | 2010/02/23 |
| 34 | 2009/08/08 19:19 | 120.71 | 22.67 | 0.64 | 705.7 | 01Q910 | 7.6 | 1158 | R | 2010/03/06 |
| 35 | 2009/08/08 20:15 | 120.73 | 22.59 | 0.73 | 1509.7 | C1R240 | 12.8 | 74 | N | 2010/04/10 |
| 36 | 2009/08/08 20:27 | 120.92 | 23.40 | 0.12 | 2278.6 | C1V460 | 4.5 | 1949 | N | 2010/01/11 |
| 37 | 2009/08/08 21:11 | 120.90 | 23.46 | 0.15 | 1904.4 | C1V460 | 2.3 | 1949 | R | 2010/02/23 |
| 38 | 2009/08/08 21:30 | 120.92 | 23.49 | 0.12 | 2450.4 | C1V460 | 6.4 | 1949 | N | 2010/02/23 |
| 39 | 2009/08/08 21:42 | 120.91 | 23.10 | 0.25 | 2274.1 | C1V460 | 37.3 | 1949 | R | 2010/04/10 |
| 40 | 2009/08/08 22:16 | 120.66 | 23.17 | 2.50 | 681.3 | C1R880 | 6.4 | 223 | R | 2010/02/23 |
| 41 | 2009/08/08 22:52 | 120.90 | 23.54 | 0.12 | 1936.8 | C1I340 | 4.5 | 897 | R | 2010/02/23 |
| 42 | 2009/08/08 23:02 | 120.60 | 23.03 | 0.13 | 747.9 | C0V250 | 5.2 | 298 | N | 2010/04/10 |
| 43 | 2009/08/08 23:14 | 120.75 | 23.29 | 0.56 | 1525.1 | C1V200 | 7.7 | 860 | R | 2010/02/23 |
| 44 | 2009/08/08 23:15 | 120.77 | 22.63 | 0.15 | 2309.0 | 01Q250 | 8.2 | 950 | R | 2010/04/10 |
| 45 | 2009/08/08 23:41 | 120.84 | 22.63 | 0.12 | 825.0 | 01Q910 | 13.9 | 1158 | R | 2010/04/10 |
| 46 | 2009/08/09 00:34 | 120.77 | 23.22 | 2.24 | 1352.5 | C1V210 | 4.0 | 700 | R | 2010/02/23 |
| 47 | 2009/08/09 02:52 | 120.77 | 23.23 | 0.81 | 1559.3 | C1V210 | 4.1 | 700 | R | 2010/02/23 |
| 48 | 2009/08/09 03:55 | 120.72 | 22.60 | 0.63 | 923.5 | 01Q250 | 2.5 | 950 | N | 2010/04/10 |
| 49 | 2009/08/09 09:37 | 120.81 | 22.56 | 2.31 | 1144.3 | 01Q350 | 14.3 | 250 | N | 2010/04/10 |

| 50 | 2009/08/09 11:00 | 120.77 | 22.82 | 0.13 | 1669.4 | C1R120 | 9.0 | 820 | R | 2010/04/10 |
| 51 | 2009/08/10 03:54 | 120.80 | 23.25 | 0.20 | 1227.6 | C1V210 | 2.8 | 700 | R | 2010/02/23 |
| 52 | 2009/08/10 04:22 | 120.76 | 23.31 | 1.52 | 1387.2 | C1V160 | 6.3 | 1040 | R | 2010/02/23 |
| 53 | 2010/09/19 23:24 | 120.73 | 22.85 | 0.15 | 1135.0 | 01Q910 | 13.9 | 1158 | R | 2011/04/16 |
| 54 | 2011/08/30 07:10 | 120.93 | 22.86 | 0.12 | 849.8 | 01Q350 | 44.9 | 1275 | R | 2012/02/27 |
| 55 | 2011/08/30 09:13 | 121.18 | 23.69 | 0.11 | 1811.5 | C1T940 | 19.6 | 1570 | R | 2012/02/27 |
| 56 | 2011/08/31 09:37 | 120.98 | 23.33 | 0.11 | 2714.0 | 01V070 | 8.5 | 2230 | R | 2012/02/27 |
| 57 | 2012/08/01 18:40 | 121.42 | 24.58 | 0.12 | 1512.0 | 01U050 | 8.1 | 400 | R | 2013/07/11 |
| 58 | 2012/08/02 10:00 | 121.85 | 24.52 | 0.12 | 83.3 | C0U710 | 33.3 | 1810 | N | 2013/06/28 |
| 59 | 2012/08/02 19:00 | 120.95 | 23.74 | 0.25 | 1677.9 | C1I310 | 6.6 | 1001 | N | 2013/06/03 |
| 60 | 2012/08/03 01:02 | 121.38 | 24.36 | 0.19 | 2356.6 | 41U090 | 4.7 | 1930 | N | 2013/07/11 |
| 61 | 2013/07/13 14:27 | 120.89 | 23.02 | 0.40 | 2604.8 | C1V270 | 10.1 | 1792 | R | 2014/07/13 |
| 62 | 2013/08/22 19:05 | 121.07 | 23.38 | 0.18 | 2114.6 | C1I140 | 41.2 | 1700 | R | 2014/07/13 |

---

## Author Response (AR2)

**Reply to review report of**
**Evaluating critical rainfall conditions for large-scale landslides by detecting event times from seismic records**

Hsien-Li Kuo, Guan-Wei Lin*, Chi-Wen Chen, Hitoshi Saito, Ching-Weei Lin, Hongey Chen, Wei-An Chao

* Correspondence should be addressed to Guan-Wei Lin; gwlin@mail.ncku.edu.tw

This file includes three sections:
1. Reply to the comments of editor
2. Reply to the comments of reviewer #1
3. Reply to the comments of reviewer #2
4. Certificate of Editing
5. The marked-up manuscript

**1. Reply to the comments of editor**

**We have now received two reviews and two very detailed answers by the authors, along with some modified sections of the manuscript + some possible Annexes. All these support the improvement of the manuscript. We encourage the authors to submit their final revised version of the manuscript assuming also:**

**- a careful technical review of the grammar (some mix of passive/active voices are used - please homogeneize) and figures (some typos errors in some of them) for instantce Guezzeti instead of Guzzetti on figure 6, and so on)**

**- a more in depth discussion on the generosity of the approach used for establishing the EW thresholds in other cases ...**

R: The authors deeply appreciate the editor's reviewing and providing valuable suggestions to improve the manuscript. The authors have improved the English writing of the manuscript through a language editing service to make sure that the article is free of grammatical, spelling, and other common errors.

The authors have made a more in-depth discussion on the application of rainfall threshold to Typhoon Soudelor of 2015. All thresholds proposed in the study have been tested in the case study. The illustration has been made in the section 5.1 as follows:

"To verify the usability of the rainfall thresholds proposed in this study, data from Typhoon Soudelor of 2015 were used to demonstrate the early warning performance. One of the most powerful storms on record, Typhoon Soudelor made landfall in Taiwan on August 7, 2015. It generated 1400 mm of rainfall in northeastern Taiwan and almost 1000 mm of rainfall in the southern mountainous area of Taiwan (Wei, 2017; Su et al., 2016). After seismic signal analysis, the time of a large landslide (named the Putanpunas Landslide) in southern Taiwan, 2015/8/8 18:59:50 (UTC), was obtained (Fig. 7). The seismic signals generated by the Putanpunas Landslide were also detected by Chao et al. (2017). The seismic signals generated by this large landslide were identified from six BATS stations, and the distance error was less than 6 km. The rainfall records of rain gauge station C1V190, which was situated in the same watershed and 14.6 km away from the large landslide, were collected for rainfall analysis. At rain gauge station C1V190, it dropped a cumulated rainfall of 546 mm and had a maximum rainfall intensity of 39 mm/h on August 8 (Fig. 8). The rainfall event began at 22:00 August 7 and lasted for 26 hours, and the Putanpunas Landslide initiated at the 22nd hour. This landslide occurred when the rainfall intensity was on the decline.

Once the rainfall conditions at a given rainfall station exceed the rainfall threshold for triggering landslides, the slopes located within the region of the rainfall station will have high potential for failure. When this threshold is reached, landslide warnings can be issued. Based on the statistically-based I-D threshold for small landslides, a small-landslide warning would have been issued at the sixth hour of the rainfall event (Fig. 8), sixteen hours before the Putanpunas Landslide. This premature warning could have been declared a false alarm, and people might have returned to the affected area. Therefore, it is essential to establish different thresholds for landslides of different scales. The I-Rt threshold (i.e., $Rt \cdot I = 5,640$) would have led to a large-landslide warning at the ninth hour of the rainfall event (i.e., thirteen hours before the Putanpunas Landslide occurred), and the statistically-based I-D threshold for large landslides would have yielded a landslide warning at the same time. These warnings would also have been premature. In contrast, a warning based on the Rt-D threshold (i.e., $Rt \cdot D = 12,773$) would have been issued three hours after the time of the Putanpunas Landslide. However, applying the rainfall records and the critical height of water model (i.e., $(I-1.5) \cdot D = 430.2$) would have led to a landslide warning at 16:00 on August 8, three hours before the time of the Putanpunas Landslide. This warning would have allowed sufficient time for evacuation and had low probability of being declared a false alarm. Compared to the statistically-based I-D threshold, the I-Rt threshold, and the Rt-D threshold, the critical height of water model had a better early-warning performance for the 2015 Putanpunas Landslide."

**Kuo et al., present a landslide catalogue in Taiwan, obtained by remote sensing, from which they extract 62 large landslides that can be accurately timed thanks to seismic detection, and compared to local rainfall gaging data. Then they assess which type of rainfall threshold could be derived for this dataset, including a threshold guided by physical considerations, and compare it to a dataset of smaller landslides in Taiwan. The paper ends with a rather unconvincing or unclear discussion on potential variabiliy of the thresholds and on issues sith seismic detection.**

**Overall, the authors present an interesting, novel dataset (although relatively modest) and do a series of classic (rainfall threshold) and less classic (physically based threshold) analysis that can be worth publishing, but the discussion and some of the analysis need to be improved before that.**

R: The authors very much appreciate the reviewer's valuable time, comments and suggestions.

(1) **Major comment**

**1. Timing is an issue but rainfall estimation as well. Notably because rain gage may be far from the landslides and not experiencing similar rainfall especially due to orographic effects. The author explain they only associate landslide with rainfall measured within 100km². I think this is a good start but in the analysis it would be good to indicate (by a color coding?) the horizontal distance from the landslide, as well as to discuss difference in elevation between station and landslide median elevation for example. This would allow the authors to discuss uncertainty and the degree of reliability of rainfall estimates for the landslides.**

R: The authors appreciate the reviewer's constructive suggestion. The spatial information (distance and elevation) of each used rain gauge station has been added to supplementary materials as Table S2.

The effect of rain gauge distribution over the accuracy of rainfall has been assessed using gauge observation in a 35 km × 50 km region of south Taiwan (Fig. S2). The amounts of daily rainfall during 2009 Typhoon Morakot (8/6-8/11) recorded at 19 rain gauge stations were selected to validate the accuracy of rainfall. At first, the amounts of daily rainfall were interpolated to 01V040 station using IDW methods. The errors between measurements and interpolated data were smaller than 15 %. It indicates IDW method can be used to interpolate rainfall to a selected location in our study area.

Secondly, the amounts of daily rainfall at the central point of the 35 km × 50 km region were estimated. The errors of daily rainfall between the central point and the nearest rain gauge station (01V040) were smaller than 10 % (0.5%-10% at different date).

Besides, the correlation coefficients would keep at 90% as a distance between the central point and rain gauge stations less than 20 km, and even keep at 98% as a distance less than 10 km (Fig. S3). Therefore, in the study, an upper limit of basin area smaller than 100 km$^2$ (10 km × 10 km was adopted to avoid a significant decrease of the accuracy of rainfall.

The influence of topography on rainfall variability has been analyzed in the same 35 km × 50 km region of south Taiwan. The highest station elevation is 1792 m a.s.l. at C1V270, and the lowest station elevation is 105 m a.s.l. at C10830. The standard deviation of station elevation is 561 m. The values of standard deviation of daily rainfall at the 19 stations were calculated, and less than 13% except a high standard deviation, 45%, on August 6 (average daily rainfall less than 2 mm). The results demonstrated that high and even extreme rainfall are less influenced by elevation, while low and medium rainfall events are significantly influenced by elevation variation, with most of the rainfall appearing on high elevations. Similar results have also been reported by some previous studies (Sanchez-Moreno et al., 2014; Ge et al., 2017). Because the study only considered the rainfall events with total cumulated rainfall greater than 500 m, the elevation effect was ignored as selecting rain station. The above illustration has been attached to the supplementary material S3.

[Figure]

Fig. S2

[Figure]

Fig. S3

Reference

Mishra, A.K. (2013) Effect of rain gauge density over the accuracy of rainfall: a case study over Bangalore, India. SpringerPlus, 2, 311.

Sanchez-Moreno, J.F., Mannaerts, C.M., and Jetten, V. (2014) Influence of topography on rainfall variability in Santiago Island, Cape Verde. International Journal of Climatology, 34, 1081-1097.

Ge, G., Shi, Z., Yang, X., Hao, Y., Guo, H., Kossi, F., Xin, Z., Wei, W., Zhang, Z., Zhang, X., Liu, Y., and Liu, J. (2017) Analysis of Precipitation Extremes in the Qinghai-Tibetan Plateau, China: Spatio-Temporal Characteristics and Topography Effects. Atmosphere, 8(7), 127, doi:10.3390/atmos8070127.

**2. I think the attempt of the authors to define a threshold based on physical considerations is worth, but insufficient in the present form: the assumption and limit of the model lack validation/discussion, and the practical utility/validity of the model compared to pure empirical ones is poorly demonstrated. I give detailed proposition to test and refine the model, but in any case a more quantitative comparison of the validity of the different threshold seems important if the author want to underline the physical model**

**has a path forward. I think also this part may benefit from being put in perspective compared to other work on physically based threshold. For example:**

**Salciarini and Tamagni 2013, Physically based rainfall thresholds for shallow landslide initiation at regional scales**

**Papa et al., 2013, Derivation of critical rainfall thresholds for shallow landslides as a tool for debris flow early warning systems**

**Alvioli et al., 2014, scaling properties of rainfall induced landslides predicted by a physically based model.**

**R:** The authors appreciate the reviewer's suggestions and agree that the comparison of physically-based and statistically-based thresholds is needed. The study focused on rainfall conditions for triggering landslides in a wide (national scale) study area, a purely physical model may be not suitable. We would like to call it a mixed physically- and statistically-based model. The rainfall threshold using a mixed physically- and statistically-based model in the study will be compare with others using physically-based models. The relative discussion has been added to the text as below.

[revised manuscript text omitted]

**3. I think the discussion needs to be revised significantly. The authors seek to discuss effects on critical threshold that cannot really be assessed with the data they have, while several points are not really discussed: For example 1/ uncertainty on rainfall parameters, 2/ the added value of seismic dating of landslide and its limit (size of landslide distance from stations (currently section 5.3 needs significant clarification) , 3/ The value of the critical rainfall volume : how better compare with other, how to determine or constrain I0 etc**

**R:** The authors appreciate the reviewer's constructive comment. The section of discussion has been revised significantly. The revision includes:

1) The authors agree that uncertainty on rainfall parameters will influence on the distribution of statistically-based rainfall data. In order to constrain the indeterminate variation of rainfall threshold analyses, a consistent process of calculating rainfall data with a standard of station selection has to be constructed. In the study, we tested the accuracy of rainfall data and used a consistent calculation method for rainfall parameters carefully. Therefore, the variation of rainfall parameters (*I, D*, and *Rt*) could be under control. The detailed validation of rainfall parameters has been added to the section S3 of the supplementary material.

2) The detailed information (position, time, elevation, disturbed area, used rainfall station, activity) of each detected landslide has been added to the supplementary material as Table S2.

3) The critical height of water (Qc) was estimated using the physically-based model proposed by Keefer et al. (1987). Subsequently, the threshold equation, (*I-*

$I_0$)×$D$=Qc, was adopted to fixed the lower boundary of rainfall data in the *I-D* plot. The value of $I_0$ was estimated using the same statistically-based method with *I-Rt* threshold. The value of 1.5 was obtained as the exceeding probability of 5%. We would like to call it a mixed physically- and statistically-based model. The mixed model could recover the limitation while we just used a purely physically-based model or a purely statistically-based model. The comparison of the critical height of water model with other studies has been added in Figure 6, and table S3. The modified illustration has been added to the test as below:

> "In this study, the $Q_C$ threshold for a large landslide was estimated based on a mixture of physically- and statistically-based methods. Unlike other physically-based *I-D* thresholds, which are commonly constructed based on artificial rainfall information for shallow landslides (Salciarini et al., 2012; Chen et al., 2013c; Napolitano et al., 2016) (Table S3), the $Q_C$ threshold proposed in this study seemed to be higher and more suitable for large landslides (Fig. 6d)."

**4. Last, I strongly suggest the authors to define variable names for antecedent rainfall (e.g. Ra), cumulated rainfall (e.g. Rc) to later compare with Rt (Rt = Rc + Ra) and to be consistent in text and figure when they talk about rainfall amount.**

**R:** Thanks for the suggestion. The variable names have been modified according to the suggestions.

**(2) Line by Line comments:**

1. **P2 L 5: LSL / SSL : this is heavy and makes the draft harder to read. Why not simply use small and large landslide and indicating the boundary is at 0.1km² ?**

**R:** Thanks for the suggestion. The origin term, large-scale landslide and small-scale landslide, have both replaced with "large landslide" and "small landslide", respectively.

2. **P2 L21: State in the text how was estimated the occurrence time. Based on peak rainfall correct? In Fig 1 Caption you say that in general peak rainfall intensity is used. This may go int the main text, with one or two references. Indeed, simple groundwater modelling (e.g. Wilson and Wieczorek, 1995) could estimate soil moisture based on the rainfall data and find a maximal pore pressure after the peak rainfall. Other simple modelling approach or assumption may give different estimation times.**

**R:** Thanks for the suggestions. The authors agree that more and more useful approaches have been developed to get the exact time information of landslide initiation. However, the approaches all depended on in-situ monitoring or other assumptions. So far, the most common and convenient way to assess a factor of rainfall intensity is still based on the peak rainfall intensity. The statement on peak rainfall intensity has been added to text with some references (i.g. Chen et al., 2005; Wei et al., 2006; Staley et al., 2013; Yu et al., 2013; Xue et al., 2016). The study uses the time interval between the timing with peak rainfall intensity and exact landslide timing to explain the misjudgment results of rainfall analysis (Chen et al., 2005). The reference have been added to text as below:

> "…In general, if the exact time of a landslide is unknown, the time point with the maximum hourly rainfall will be conjectured as the time of the landslide (Chen et al., 2005; Wei et al., 2006; Staley et al., 2013; Yu et al., 2013; Xue et al., 2016).……"

**I understand you want a single average threshold to compare to a population. Nevertheless, you can make an almost individual prediction of each large landslide (with Depth and Slope) and compare it to uniquely constrain rainfall information, thanks to your seismic dating. I think it would be worth checking the validity of the model this way, and potentially refining the drainage model that seems critical to really obtain a physically based threshold.**

**R:** Thanks for valuable suggestions. The authors have tried the suggested approaches to recalculate Qc and to estimate the rainfall threshold. The revised rainfall threshold is $(I - 1.5) D = 430.2$. The relative paragraph and Fig. 6 has been modified in the later version of manuscript.

The modified text is as follows:

"The critical height of water, $Q_C$, on a sliding surface for each large landslide was estimated based on its slope gradient, its depth (estimated by the equation Z = $26.14A^{0.4}$; Z: depth in m; A: disturbed area in $m^2$), and the geological material parameters of the study area (Table 1). The $Q_C$ value was inserted into $Q_C = (I - I_0) \cdot D$ to obtain an $I_0$ value for each large landslide. For the 62 detected landslides, the cumulative probability of 5% of the $Q_C$ and $I_0$ values was taken as the critical value. The critical value of $I_0$ was 1.5, the critical $Q_C$ was 430.2, which is more suitable for large than for small landslides, and the threshold curve was rewritten as $(I - 1.5) \cdot D = 430.2$."

27. **P9 L 23 -25: Is this curve allowing to better predict the LSL compared to the other plots in Fig 5 (Especially I – Rt or I-D?) Same question for the separation from SSL/LSL. The authors should provide some statistics confirming that this model is better than a Rt -I for example. Log Log plot is absolutely necessary for all plot. Further, the very low drainage found by the authors, mean their threshold is almost ID ~452 or R~452. And indeed a vertical line in the I -Rt graph at about 500 may be as good...**

**R:** Thank you for your suggestion. The log-log figure will be modified based on suggestions. And a vertical line has been added in the *I-Rt* graph at *Rt* of 500m as below:

[Figure]

Figure 6

**28. P10 L14: If so you should observe a larger fraction of the LSL in 2010-2014 neighbored a 2005-2009 landslide, compare to LSL in 2005-2009 being the reactivation of older landslides. Given the small dataset (62?), I encourage the authors to check each LSL and report the proportion of reactivation before and after 2009. Then they can support and discuss this hypothesis**

**R:** Thank you for the valuable comments. The information on reactivity of 2010-2014 landslides will be provided in table S2. Once a 2010-2014 landslide was on the location of a 2005-2009 landslide, the landslides was classified as a reactivated landslide (marked as "R" in Table S2).

**29. P10 Section 5.1 and 5.2 Strange writing: the authors oscillate between presenting new result about shift between threshold for different subset and then concluding that they are insignificant. Based on Fig 7 and 8 I do think the**

**dataset of the author is insufficient to discuss these two topics and I would strongly suggest the author to remove these two sections (or just mention rapidly that sub dividing the dataset does not give clear difference and send Fig 7 and 8 in Supplement.) and give more space to discussing other points, like their critical rainfall model, or the uncertainties on rainfall.**

**R:** Thanks for comments. The authors agree the conceptual view of reviewer. The discussion needs more solid information and field investigation. Therefore, the part about the influence of rock types and an extreme event has been removed and replaced with in-deep comparison of different rainfall threshold.

**30. P11 section 5.3: maybe interesting but Fig 9 is too confusing. So I suspect text and Fig 9 should be clarified a bit.**

**R:** To determine the limits of large landslide detection distance as a function of event volume, we selected the farthest seismic station at which each event was detectable. An event was deemed detectable when we had selected the station for the distance-dependency analysis. The remaining results are shown on a plot of distance versus disturbed area (Fig. 9), where we can observe an upper detection limit described by equation 5. If, for a given event, a station plots in the lower right area below the dashed line (equation (5)), the seismic signal should be detectable. The detection limit also depends on the station signal quality; if the noise level is high, the signal may be obscured, even though a station farther away with a lower noise level will still record it clearly. Similar studies had been reported by Dammeier et al. (2011) and Chen et al. (2013).

**31. P11 Eq 5: to discuss validity and limits of EQ 5 it should be made clearer how (empirically?) and with which dataset/environment this relationship was obtained.**

R: The authors appreciate reviewer's recommendation. In the study, the lower boundary of detection was determined empirically based on two lowest values of the farthest distance of detection (i.g. 31.0 km and 37.6 km) having the disturbed areas of $1.6 \times 10^5$ and $1.2 \times 10^5$ m$^2$ . Dammeier et al. (2011) used a similar way to get their equation of the lower boundary of detection. The modification has been added to text as follows:

"…The boundary of detection was determined empirically based on the two lowest values of the farthest distance of detection (i.e., 31.0 km and 37.6 km) of landslides having disturbed areas of $1.6 \times 10^5$ and $1.2 \times 10^5$ m$^2$. For a given large landslide, if a station is located below the upper detection limit, the seismic signal should be detectable. However, not all the stations located in detectable regions

recorded clear large landslide-induced seismic signals. One possible reason for this lack of detection is that the environmental background noise affected the signal to noise ratio of the seismic records during heavy rainfall events. Therefore, the detection limit may also depend on the signal quality at each station."

**32. Fig 3: closest station is MASB (in the caption) or SGSB (in the map) ? It means 90% of the landslide and seismic signal**

**R:** We deeply thanks for careful inspection. The closest station should be SGSB. The mistake has been revised in Figure 3 of the later version of manuscript.

**33. Fig 5: The last panel is not very clear: Cumulated rainfall is the total rainfall in the triggering storm. Antecedent rainfall has no reason to be compared directly with landslide occurrence, but only when summed with the cumulated rainfall. So why not show Rt the total effective rainfall together with Rc the cumulative rainfall (Given that Rt>= Rc it should be easy to visualize).**

**R:** Thanks for the constructive suggestion. The last panel in Figure 5 has been revised to display Rt and Rc. The revised Figure 5 is as below:

[Figure]

Figure 5

**34. Fig 6: Log Log scale is needed on all panel. Right now we do not see clearly the position of the different data points.**

**R:** Thanks for the constructive suggestion. All sub-figures in Figure 6 have been transferred to log-log scale as follows:

[Figure]

Figure 6

**35. Fig 7 and 8: I do not believe any of the subset can be significantly distinguished. What is driving the (small) difference in threshold curve is only 1 or 2 points out of each subset (that seems to be15-25 points). These low points shift the threshold while the bulk of each population do not seem different in any way. I am convinced this can only be due to chance and not to a shift of the whole population. I am even surprised that the curves are so low because if they are the 5% exceedance probability ~1 point should be left out in subset of ~20...**

**R:** We appreciate the valuable comments, and decided to remove this section. Meanwhile, we added a section to illustrate the validation of the rainfall thresholds mentioned in the study and other previous studies.

**36. Fig 9: I really tried, but did not understand it... I got that the line, is an empirical estimation of the distance at which station should be able to detect a landslide of a given size. What are the points? The 62 LSL? If yes, why are they all above the line? Does that mean only distant station detect the slides? I can believe for some but not the whole dataset, and this seems contradictory with Fig 3**

**R:** This detection limit line is estimated empirically based on distribution of data which represented the farthest distance that landslide signal was detected. This result indicate that the distance between a station and a landside below this line must be detected, but if it exceeds this line, it would probably be missed.

To determine the limits of large landslide detection distance as a function of event volume, we selected the farthest seismic station at which each event was detectable. An event was deemed detectable when we had selected the station for the distance-dependency analysis. The remaining results are shown on a plot of distance versus disturbed area (Fig. 9), where we can observe an upper detection limit described by equation 5. If, for a given event, a station plots in the lower right area below the dashed line (equation (5)), the seismic signal should be detectable. The detection limit also depends on the station signal quality; if the noise level is high, the signal may be obscured, even though a station farther away with a lower noise level will still record it clearly.

In total, 62 data points in Fig. 9. Each data point represents the distance between landslide location and the farthest detectable station as well as landslide-disturbed area. In the order words, it indicate that landslide signals can be detectable as the distance between landslide and seismic station shorter than the value of data. Therefore, to determine a lower boundary of these data can demarcate an effectively detectable region. The illustration of section 5.2 has been modified as follows:

> "The number of large landslides detected from seismic records, 62, comprised only nine percent of the total large landslides in 2005–2014 in Taiwan. This low percentage indicates that the vast majority of large landslides were not well identified from seismic records. If this limitation can be surmounted, more time information on large landslide occurrences can be used to develop rainfall thresholds. The average interstation spacing of the Broadband Array in Taiwan for Seismology is around 30 km. A higher density of seismic stations would improve

the detection function. In addition, to determine the limitation of large landslide detection distance as a function of large landslide-disturbed area, the most distant seismic station where large landslide signals were visible was selected. Some previous studies have applied similar approaches to probe the detection limit (Dammeier et al., 2011; Chen et al., 2013a). The relationship between the maximum distance of detection and the large landslide-disturbed area shows a limitation of the detection distance due to the large landslide's magnitude (Fig. 9). In Figure 9, each data point represents the distance between a landslide location and the most distant seismic station detecting it, as well as the landslide-disturbed area. In other words, when the distance between a seismic station and a landslide that has the same given landslide-disturbed area as the data is shorter than the value of the data, seismic signals induced by the landslide can be interpreted from the records of the seismic station. Therefore, a lower boundary of these data can be determined to demarcate an effective detectable region. As a large landslide's area increases, the maximum distance between the large landslide location and seismic detection increases. A detection limit can be described by

$$\log(\text{distance}) = 0.5069 \times \log(\text{area}) - 1.3443, \tag{5}$$

The boundary of detection was determined empirically based on the two lowest values of the farthest distance of detection (i.e., 31.0 km and 37.6 km) of landslides having disturbed areas of $1.6 \times 10^5$ and $1.2 \times 10^5$ m$^2$. For a given large landslide, if a station is located below the upper detection limit, the seismic signal should be detectable. However, not all the stations located in detectable regions recorded clear large landslide-induced seismic signals. One possible reason for this lack of detection is that the environmental background noise affected the signal to noise ratio of the seismic records during heavy rainfall events. Therefore, the detection limit may also depend on the signal quality at each station."

**37. References not used in the manuscript**
**-- Wilson and Wieczorek 1995, Rainfall thresholds for the initiation of debris flows at La Honda, California**
**-- Iverson, 2000, Landslide triggering by rain infiltration**
**-- Van Asch et al., 1999, A view on some hydrological triggering systems in landslides**
**Larsen et al., 2010, Landslide erosion controlled by hillslope material**
**R:** Thanks for reviewing. The reference list has been overhauled completely before resubmitted.

In the original manuscript, Typhoon Morakot was mentioned many times because it was one of the most tragic event in Taiwan in the past 20 years (more than 20,000 landslides, and more than four hundred large landslides with the disturbed area larger than 0.1 km$^2$), and therefore many good examples can be shown. In the modified version, the examples of other events have been provided in the supplementary material.

Table S1

|   | Event | Date (year/month/date) |
|---|-------|------------------------|
| 1 | Haitang | 2005/07/16-07/20 |
| 2 | Talim | 2005/08/30-09/01 |
| 3 | 0609 Rain | 2006/06/09 |
| 4 | Bilis | 2006/07/12-07/15 |
| 5 | 0604 Rain | 2007/06/04 |
| 6 | Kalmaegi | 2008/07/16-07/18 |
| 7 | Fung-Wong | 2008/07/26-07/29 |
| 8 | Sinlaku | 2008/09/11-09/16 |
| 9 | Morakot | 2009/08/05-08/10 |
| 10 | Fanapi | 2010/09/17-09/20 |
| 11 | Megi | 2010/10/21-10/23 |
| 12 | Nanmadol | 2011/08/27-08/31 |
| 13 | Talim | 2012/06/19-06/21 |
| 14 | Saola | 2012/07/31-08/03 |
| 15 | Tembin | 2012/08/21-08/25 |
| 16 | Soulik | 2013/07/11-07/13 |
| 17 | Trami | 2013/08/20-08/22 |
| 18 | Matmo | 2014/07/21-07/23 |
| 19 | Fung-Wong | 2014/09/19-09/22 |

The modified text is as follows:

"…In this study, a total of nineteen rainstorm events having cumulated rainfall exceeding 500 mm (seventeen typhoon-induced events and two heavy rainfall events) in the years 2005–2014 were selected,.…"

**2. P3 L8: Date of the images? Number? Mapping only for the 2009 event.**

R: Thanks for comments. The date and number of used SPOT-4 satellite images have been listed in Table S2. Landslide mapping was conducted for nineteen rainstorm events (seventeen typhoon-induced events and two heavy rainfall events).

**3. P3 L23: Why 0.1km² Is it the limit of the automatic detection based on SPOT images? How many landslides were detected?**

**R:** Based on the definition and characteristic of deep-seated gravitational slope deformation (DSGSD) and description of large-scale landslides (Lin et al., 2013a; Lin et al., 2013b), a large-scale landslide should have three characteristics, including 1) a depth larger than 10 m, 2) a volume greater than 1000,000 $m^3$, and 3) a speedy

movement velocity. In practice, it is difficult to get these three characteristics without in-situ investigation and geodetic survey. Therefore, we chose the disturbed area of 100,000 m$^2$ (0.1 km$^2$, volume/depth) as the indicator to sort large-scale landslides from other types of slope failure. Landslide interpretation with satellite imagery is the fastest way to classify large-scale landslides. Actually, more than three hundred seismic signals having the seismic characteristics of landslide-induced ground motions, however, only 62 signals having clear landslide-signal signatures were detected by at least three seismic stations. Therefore, we just could locate the possible locations of these 62 signals and paired the locating points with landslides. Although the successful detection and locating rate may be less than 20 % in the period of 2005-2014, we believe that the 62 landslides still provide many valuable time information.

**R:** The sentence maybe not clear. The revision is as below:

"…To reduce the uncertainty caused by manual identification, events with obvious triangular signatures in the spectrograms (e.g. Fig. S1) were used to examine rainfall statistics in this study."

The in-deep description of the characteristics of landslide-induced seismic signals has been added to text as follows:

"…The seismic wave generated by a landslide can be attributed to the shear force and loading on the ground surface as the mass moves downslope. Many studies have shown that the source mechanism of a landslide is highly complicated, and that the seismic waves of landslides mainly consist of surface waves and shear waves. Consequently, it is difficult to distinguish $P$ and $S$ waves from station records (Lin et al., 2010; Suwa et al., 2010; Dammeier et al., 2011; Feng, 2011; Hibert et al., 2014). The onset of a landslide seismic signal is generally abrupt. The seismic amplitude gradually rises above the ambient noise

level to the peak amplitude, exhibiting a cigar-shaped envelope. After the peak amplitude, most landslide-generated seismic signals have relatively long decay times, averaging about 70% of the total signal duration (Norris, 1994; La Rocca et al., 2004; Suriñach et al., 2005; Deparis et al., 2008; Schneider et al., 2010; Dammeier et al., 2011; Allstadt, 2013). In the frequency domain, landslide-induced seismic energy is mainly distributed below 10 Hz, and the signature in a spectrogram is triangular due to an increase in high-frequency constituents over time (Suriñach et al., 2005; Dammeier et al., 2011). The triangular signature in the spectrogram is the distinctive property that distinguishes landslide-induced signals from those of earthquakes and other ambient noise"

**R:** Thanks for the comments. In the figure 2c, five matched landslides (including Xiaolin) were visible. Each orange circle means a successfully-paired landslide. The outline of each paired landslide will be added to the figure. Theoretically, the seismic signal induced by a small landslide could be found if there is a seismic station very close to it. In practice, the seismic signals generated by small landslides do not contain enough energy to be recorded by remote seismic stations. Therefore, we do not try to examine small landslides by seismicity approaches. Besides, the signal and spectrogram of a 2005 large landslide smaller than Xiaolin will be provided and added to supplementary material as below:

[Figure]

Fig. S1

**10. P5 L3: Now you are studying events between 2005-2014? It is a little bit**

**confusing. How many typhoon events? don't you consider previous smaller rainfall events that could affect the mechanical properties of the slopes?**

R: In the study, totally seventeen typhoon events and two heavy rainfall events occurring in the period of 2005-2014 were chosen to examine the seismic records and identify landslide-induced signals. The list of selected events has been provided in Table S1.

In the study, seven-days antecedent rainfall was considered as a rainfall parameter. The effect of antecedent rainfall should decay with time. Therefore, decay rate of antecedent rainfall with day was used to estimate effective rainfall. According to the decay rate, 0.7, the effect of antecedent rainfall with counting days longer than 7 days is slight. In the study, we adopt seven-days antecedent rainfall to estimate effective rainfall.

Table S1

|    | Event | Date (year/month/date) |
|----|-------|------------------------|
| 1  | Haitang | 2005/07/16-07/20 |
| 2  | Talim | 2005/08/30-09/01 |
| 3  | 0609 Rain | 2005/06/09 |
| 4  | Bilis | 2005/07/12-07/15 |
| 5  | 0604 Rain | 2006/06/04 |
| 6  | Kalmaegi | 2006/07/16-07/18 |
| 7  | Fung-Wong | 2008/07/26-07/29 |
| 8  | Sinlaku | 2008/09/11-09/16 |
| 9  | Morakot | 2009/08/05-08/10 |
| 10 | Fanapi | 2010/09/17-09/20 |
| 11 | Megi | 2010/10/21-10/23 |
| 12 | Nanmadol | 2011/08/27-08/31 |
| 13 | Talim | 2012/06/19-06/21 |
| 14 | Saola | 2012/07/31-08/03 |
| 15 | Tembin | 2012/08/21-08/25 |
| 16 | Soulik | 2013/07/11-07/13 |
| 17 | Trami | 2013/08/20-08/22 |
| 18 | Matmo | 2014/07/21-07/23 |
| 19 | Fung-Wong | 2014/09/19-09/22 |

**11. P5 L6: How can you consider the topographic, orographic effects?**

R: We appreciate the comments. We tested the rainfall data used in the study to validate the influence of distance and topographic effect on rainfall distribution. The effect of rain gauge distribution over the accuracy of rainfall has been assessed using gauge observation in a 35 km × 50 km region of south Taiwan (Fig. S2). The amounts of daily rainfall during 2009 Typhoon Morakot (8/6-8/11) recorded at 19 rain gauge stations were selected to validate the accuracy of rainfall. The influence of topography on rainfall variability has been analyzed in the same 35 km × 50 km region of south Taiwan. The highest station elevation is 1792 m a.s.l. at C1V270, and the lowest station elevation is 105 m a.s.l. at C10830. The standard deviation of station elevation is 561 m. The values of standard deviation of daily rainfall at the 19 stations were calculated, and less than 13% except a high standard deviation, 45%, on sixth August (average daily rainfall less than 2 mm). The results demonstrated that high and even extreme rainfall are less influenced by elevation, while low and medium rainfall events are significantly influenced by elevation variation, with most of the rainfall appearing on high elevations. Similar results have also been reported by some previous studies (Sanchez-Moreno et al., 2014; Ge et al., 2017). Because the study only considered the rainfall events with total cumulated rainfall greater than 500 m, the elevation effect was ignored as selecting rain station.

[Figure]

Fig. S2

R: The effect of station distance has been tested to variation of rainfall. The errors of daily rainfall between the central point and the nearest rain gauge station (01V040) were smaller than 10 % (0.5%-10% at different date). Besides, the correlation coefficients would keep at 90% as a distance between the central point and rain gauge stations less than 20 km, and even keep at 98% as a distance less than 10 km (Fig. S3). Therefore, in the study, an upper limit of basin area smaller than 100 km² (10 km × 10 km was adopted to avoid a significant decrease of the accuracy of rainfall. Because the density of rainfall stations in mountainous area would significantly decreases, the number of usable rainfall stations may be limited. The size of catchment area of 100km² is the upper limit for choose rainfall station. In practice, we chose the closest rainfall station.

[Figure]

Fig. S3

**13. P5 L2: rain event = typhoon?**

R: Yes. The sentence has been revised as below.

"In the study, hourly rainfall data were collected from the records of rain gauge stations (Fig. 2a). The major rainfall events analysed in the study were typhoon events. The distribution of precipitation during typhoon events is usually closely

related to the typhoon track and the position of the windward slope, also as known as the orographic effect....."

**14. P6 L4: 193 small landslides for which period?**

**R:** The 193 small landslides were investigated by SWBC during 2006-2014. The illustration on the small landslides has been improved in the later version of manuscript as below.

"In addition to the time information of large landslides, the time information of 193 small landslides, such as shallow landslides and debris flows, from the years 2006–2014 was collected from the annual reports of debris flows investigated by the Soil and Water Conservation Bureau (SWCB) of Taiwan. For these landslides, no information was extracted from seismic records. Most of the 193 small landslides caused disasters and loss of life and property. In some cases, in-situ steel cables or closed-circuit television recorded the time information. This information was applied to the rainfall data analysis and then used to compare the rainfall conditions of the large landslides."

**15. P6 L15: EQ1 cohesion here is only considered for a discontinuity (C = 0)? Or for the specific material?**

**R:** We thanks reviewer's recommendation. Well development of detachment plane (e.g., sliding surface between sedimentary layers, connected joints, weathered foliation, etc.) have been widely considered as one important geological condition to induce a large landslide. Therefore, in the study, the C' of the detachment plane is simply assumed as the value of zero to behave the critical situation of slope stability. Cohesion in equation (1) is not considered for a specific material. The illustration of C' has been modified in the text as below.

"...Good development of a detachment plane (e.g., the sliding surface between sedimentary layers, connected joints, and weathered foliation) has been widely considered as the geological condition under which a large landslide occurs (Agliardi et al., 2001; Tsou et al., 2011). Therefore, in this study, the *c'* of the detachment plane was simply assumed to be zero to represent the critical situation of slope stability."

**R:** The main research purpose of this study was to establish a rainfall warning threshold which is applicable for large landslides, so a relatively simple but effective method was adopted. In this method, Keefer (1987) assumes that there is a potential sliding surface for these landslides, and the depth of the large-scale landslides are often deep to the strata. Therefore, although the movements of the soil material are not completely the same, under this assumption, it can still reach a considerable good effect.

In order to improve the Qc threshold, the critical volume of water, $Q_c$, for each large landslide was estimated based on its slope gradient and depth (estimated by the equation: $Z = 26.14A^{0.4}$; Z: depth, m; A: disturbed area, $m^2$). Following the equation (4), the drainage rate, $I_0$, for each landslide can be calculated. For the 62 detected landslide, the cumulative probability of 5% of $Q_c$ and $I_0$ values was taken as the critical values in the mixing physically- and statistically-based threshold. The critical value of $I_0$ was 1.5, and the critical Qc was 430.2. The paragraph has been modified as follows:

> "The critical height of water, $Q_C$, on a sliding surface for each large landslide was estimated based on its slope gradient, its depth (estimated by the equation $Z = 26.14A^{0.4}$; Z: depth in m; A: disturbed area in $m^2$), and the geological material parameters of the study area (Table 1). The $Q_C$ value was inserted into $Q_C = (I - I_0) \cdot D$ to obtain an $I_0$ value for each large landslide. For the 62 detected landslides, the cumulative probability of 5% of the $Q_C$ and $I_0$ values was taken as the critical value. The critical value of $I_0$ was 1.5, the critical $Q_C$ was 430.2, which is more suitable for large than for small landslides, and the threshold curve was rewritten as $(I - 1.5) \cdot D = 430.2$."

**21. Effective rainfall, and rainfall duration thresholds according to the rock types are not clear in the figure 8, could another statistical analysis put the conclusion of the authors in obvious fact?**

**R:** Thank you for your suggestion. The figure has been removed.

**Figure:**

**22. Fig. 2. Add legend for the detected landslide: Is it detected by seismic signal analysis? Is the point, the centroid of the landslide? Why not the delineation of the landslide body? Dates of the both satellite images here.**

**R:** Thank you for your suggestion. The figure has been modified based on suggestions.

[Figure]

Figure 2

**23. Fig. 3. Location of the detected landslides in 2009? Is there other spectrogram for previous landslides? Or after 2009? or associated to another landslide triggered in 2009: X spectrogram for 1 landslide. The star is the location defined with which seismic station?**

**R:** This events, the Xioulin landslide, is one of the most tragic event during Typhoon Morakot, 2009. The other 61 detected landslides also have the triangle-shape characteristic patterns in their spectrograms. The star is the location defined with all the stations which could detect the signals from the Xioulin landslide. We provided one example of spectrograms of a landslide in 2005 in supplementary materials (Fig. S1) and one example of a 2015 landslide in Figure 7.

[Figure]

Fig. S1

[Figure]

Figure 7

**24. Fig. 4. Maybe with the topography visible on the map?**

**R:** Thank you for your suggestion. We had tried added topography in the fig. 4a. However, so many information made visually chaotic.

**25. Fig. 7. A) Is there only 1 event for the lowest limit?**

**R:** Yes. The figure will be removed in modified version.

**Certificate of Editing**

**This is to certify that the academic paper**

Evaluating critical rainfall conditions for large-scale landslides by detecting event times from seismic records

**by the authors**

Hsien-Li Kuo[1], Guan-Wei Lin[1,*], Chi-Wen Chen[2], Hitoshi Saito[3], Ching-Weei Lin[1], Hongey Chen[2,4], Wei-An Chao[5]

[1] Department of Earth Sciences, National Cheng Kung University, No. 1, University Road, Tainan City, 70101, Taiwan
[2] National Science and Technology Center for Disaster Reduction, No. 200, Sec. 3, Beixin Road, Xindian District, New Taipei City, 23143, Taiwan
[3] College of Economics, Kanto Gakuin University, 1-50-1 Mutsuura-higashi, Kanazawa-ku, Yokohama, 236-8501, Japan
[4] Department of Geosciences, National Taiwan University, No.1, Section 4, Roosevelt Road, Taipei, 10617, Taiwan
[5] Department of Civil Engineering, National Chiao Tung University, No. 1001, Daxue Rd., Hsinchu, 30010, Taiwan

**has been proofread and edited by a native speaker of English.**

The editor, Mr. John Pearson Ring, is an established freelance science editor with twenty years of experience. He was born and raised in Brunswick, Maine, USA, and earned a Bachelor of Arts degree at Hamilton College in Clinton, New York, USA. He is currently the Head Coordinator at the Language Training and Testing Center, Taipei, Taiwan (https://www.lttc.ntu.edu.tw/), and was formerly a freelance editor for the Academic Writing Education Center at National Taiwan University, Taipei, Taiwan (http://www.awec.ntu.edu.tw/).

**Signed and dated by the editor described above:**

**Signature:** _John P. Ring_

**Date:** _Sep. 10, 2018_

Contact Information:
John P. Ring
Email: johnringintaiwan@yahoo.com
Mobile: (+886) 939 647 451

[revised manuscript text omitted]